# TECHNICAL REPORT

# Pangenome-based genome inference allows efficient and accurate genotyping across a wide spectrum of variant classes

Jana Ebler [1], Peter Ebert [1], Wayne E. Clarke [2], Tobias Rausch [3,4], Peter A. Audano [5], Torsten Houwaart [6], Yafei Mao [5], Jan O. Korbel [3], Evan E. Eichler [5,7], Michael C. Zody [2], Alexander T. Dilthey [6,8,9] and Tobias Marschall [1] ✉

**Typical genotyping workflows map reads to a reference genome before identifying genetic variants. Generating such alignments introduces reference biases and comes with substantial computational burden. Furthermore, short-read lengths limit the ability to characterize repetitive genomic regions, which are particularly challenging for fast *k*-mer-based genotypers. In the present study, we propose a new algorithm, PanGenie, that leverages a haplotype-resolved pangenome reference together with *k*-mer counts from short-read sequencing data to genotype a wide spectrum of genetic variation—a process we refer to as genome inference. Compared with mapping-based approaches, PanGenie is more than 4 times faster at 30-fold coverage and achieves better genotype concordances for almost all variant types and coverages tested. Improvements are especially pronounced for large insertions (≥50 bp) and variants in repetitive regions, enabling the inclusion of these classes of variants in genome-wide association studies. PanGenie efficiently leverages the increasing amount of haplotype-resolved assemblies to unravel the functional impact of previously inaccessible variants while being faster compared with alignment-based workflows.**

Recent, single-molecule, long-read sequencing technologies have enabled breakthroughs in producing de novo haplotype-resolved genome assemblies[1–4]. Major efforts are under way[5] (https://www.genome.gov/news/news-release/NIH-funds-centers-for-advancing-sequence-of-human-genome-reference) to generate hundreds of human genome assemblies, with the intention of deriving a variation-aware pangenome representation that replaces the current linear reference genome, GRCh38. Although long-read technologies are rapidly advancing, advantages of cost and scalability, and the requirement for large study cohorts, will make short reads a more practical approach for the foreseeable future.

Diploid organisms have two copies of each autosomal chromosome, each of which carries genetic variation. The process of determining whether a known variant allele is located on none, one or both of these copies is referred to as genotyping. Variant genotyping is an essential step in genetic studies, enabling population analysis, quantitative trait locus studies or trait association analysis. Large studies have produced comprehensive catalogs of human variation ranging from single-nucleotide polymorphisms (SNPs) and indels (insertions and deletions up to 49 bp in size) to larger structural variants (SVs)[6–9], and many such variants have been linked to diseases and other traits[10–15].

Widely used genotyping methods for sequencing data[16–20] are based on short-read alignments to a reference genome or pangenome graphs, which include possible alternative alleles[21–27].

Graph-based approaches have been shown to improve genotyping accuracy over methods that rely on a linear reference genome. However, aligning sequencing reads is time-consuming even for linear reference genomes, where mapping 30× short-read sequencing data of a single human sample takes around 100 CPU hours. This problem is amplified when transitioning to graph-based pangenome references, where the read-mapping problem is even more computationally expensive.

A much faster alternative is to genotype known variants based on *k*-mers, short sequences of a fixed length *k*, in the raw sequencing reads without alignment to a reference. Counts of reference- and allele-specific *k*-mers allow fast and accurate genotyping of various types of genetic variation[28–33]. However, these methods can struggle in repetitive and duplicated regions of the genome not covered by unique *k*-mers. This is especially problematic for SVs, which are often located in repeat-rich or duplicated regions of the genome[8,34] that are generally difficult to access by short-read sequencing[35].

This problem has been addressed previously by leveraging long-range connectivity information from sequencing reads[36]. In a similar manner, haplotype-resolved assemblies of known samples could improve *k*-mer-based genotyping, especially in difficult-to-access regions of large diploid genomes, but methods for this have so far been lacking. Known haplotypes have been used to construct population-based reference panels to phase small variants (Li–Stephens model)[37] as well as impute missing genotypes[38–41], but accurate reference panels that include SVs are still lacking.

[1]Institute for Medical Biometry and Bioinformatics, Medical Faculty, Heinrich Heine University Düsseldorf, Düsseldorf, Germany. [2]New York Genome Center, New York, NY, USA. [3]European Molecular Biology Laboratory, Genome Biology Unit, Heidelberg, Germany. [4]European Molecular Biology Laboratory, GeneCore, Heidelberg, Germany. [5]Department of Genome Sciences, University of Washington School of Medicine, Seattle, WA, USA. [6]Institute of Medical Microbiology and Hospital Hygiene, Heinrich Heine University Düsseldorf, Düsseldorf, Germany. [7]Howard Hughes Medical Institute, University of Washington, Seattle, WA, USA. [8]Institute of Medical Statistics and Computational Biology, University of Cologne, Cologne, Germany. [9]Cologne Excellence Cluster on Cellular Stress Responses in Aging-Associated Diseases, University of Cologne, Cologne, Germany. ✉e-mail: tobias.marschall@hhu.de

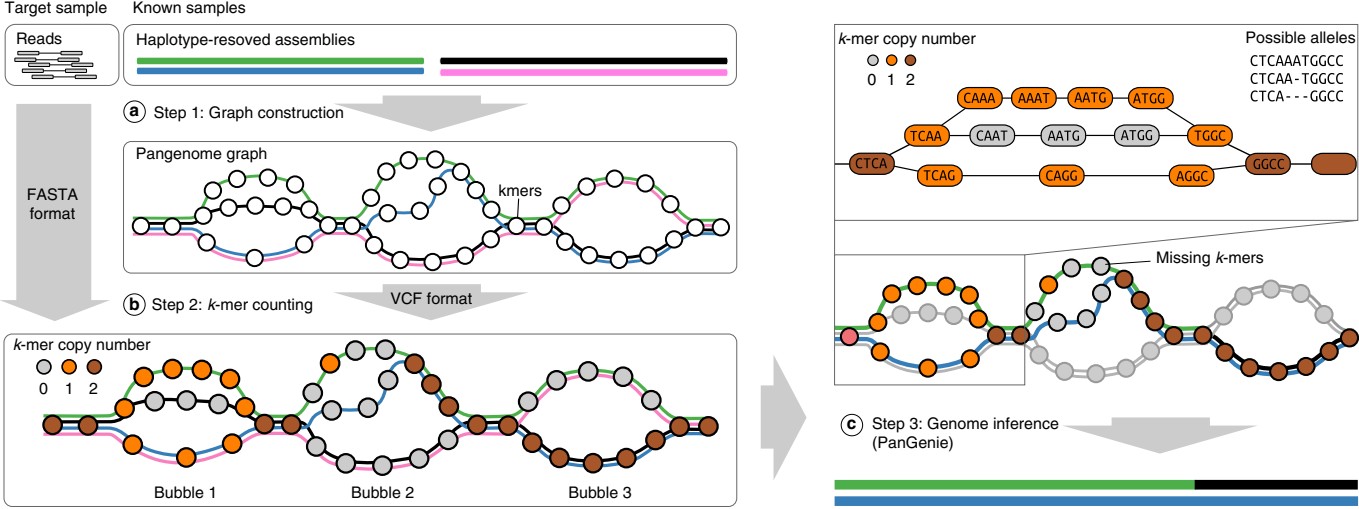

**Fig. 1 | Overview. a**, Step 1: variants are called from haplotype-resolved assemblies of a set of known samples and a pangenome graph is constructed, which represents variants as bubbles and contains one path per haplotype. **b**, Step 2: the *k*-mers (represented by circles) contained in the graph are counted in the short-read sequencing data of the target sample to be genotyped. The color of the nodes indicates copy number estimates for the *k*-mers. **c**, Step 3: PanGenie uses *k*-mer counts and haplotype paths to infer the unknown genome. For the first bubble, *k*-mer counts suggest that the sample probably carries the alleles of the green and blue haplotypes. The second bubble is poorly covered by *k*-mers; however, linkage to adjacent bubbles can be used to infer the two local haplotype paths.

In this report, we describe an algorithm, PanGenie (for Pangenome-based Genome Inference), that makes use of haplotype information from an assembly-derived pangenome representation in combination with read *k*-mer counts to efficiently genotype a wide spectrum of variants. That is, our method can leverage short and longer linkage disequilibrium (LD) structures inherent in the assemblies to infer the genome of a new sample for which only short reads are available. PanGenie bypasses read mapping and is entirely based on *k*-mers, which allows it to rapidly proceed from the input short reads to a final callset including SNPs, indels and SVs, enabling analysis of variants typically not accessible in short-read workflows—including many deletions <1 kb and most insertions ≥50 bp. We applied our method to genotype variants called from haplotype-resolved assemblies of 11 individuals, revealing a substantial advance in terms of runtime, genotyping accuracy and number of accessible variants.

## Results

**Algorithm overview.** We call variants from haplotype-resolved assemblies (see Constructing a pangenome reference) of several samples and construct a pangenome graph in which these variants are represented as bubbles and each haplotype as a path (Fig. 1, step 1). This graph is given as input to PanGenie, together with short-read sequencing data of a new sample to be genotyped. The *k*-mers contained in the graph are counted in the reads and *k*-mers unique to bubble regions are identified (step 2 in Fig. 1; Methods). PanGenie combines two sources of information to genotype bubbles: read *k*-mer counts and the already known haplotype sequences. The distribution of *k*-mer counts along the allele paths of a bubble can provide evidence for the genotype of the sample. Figure 1 (right panel) provides an example: *k*-mers corresponding to the second allele of the first bubble are absent from the reads, indicating that the individual carries the alleles of the green and blue haplotypes. However, bubbles may be poorly covered by *k*-mers or no unique *k*-mers may exist in repetitive regions of the genome. Such positions cannot be reliably genotyped based on the *k*-mer counts alone, but known haplotypes can help to infer genotypes based on neighboring bubbles (Fig. 1).

For genotyping, we integrate information from *k*-mer counts and haplotypes by constructing a hidden Markov model (HMM), which models the unknown genome as a mosaic of the provided haplotypes and reconstructs it based on the read *k*-mer counts observed in the sample's sequencing reads (Methods). Hidden states represent pairs of haplotype paths that can be chosen at each bubble position and emit counts for the unique *k*-mers of the respective region. State transitions between adjacent bubbles correspond to recombination events. Using the forward–backward algorithm, genotype likelihoods are computed for each bubble, from which a genotype is derived.

**Constructing a pangenome reference.** We generated haplotype-resolved assemblies of 14 individuals including 3 mother–father–child trios (Fig. 2a and Methods; samples include: Yoruban trio: NA19238, NA19239, NA19240; Puerto Rican trio: HG00731, HG00732, HG00733; southern Han Chinese trio: HG00512, HG00513, HG00514; and NA12878, HG02818, HG03125, NA24385 and HG03486) and used all 11 unrelated samples to call variants on each haplotype of all autosomes and chromosome X. We computed the transition:transversion (ti:tv) ratio for SNPs and the heterozygous:homozygous (het:hom) ratio as quality control measures[42,43]. Our SNP calls contained around twice as many transitions as transversions (Fig. 2b) resulting in ti:tv ratios between 2.01 and 2.02 for all samples. We obtained het:hom ratios between 1.37 and 2.20 for all our 11 callset samples. These numbers are in line with respective results for African (AFR), American (AMR), Asian (EAS) and European (EUR) individuals reported in previous studies[43,44]. Furthermore, our callset contains comparable numbers of insertions and deletions (Fig. 2c), except for the expected enrichment for insertion alleles for SVs[8]. We show detailed counts of distinct variant alleles for all types in Fig. 2d (first row) and Supplementary Tables 2 and 3. We distinguish small variants (1–19 bp), midsize variants (20–49 bp) and large variants (≥50 bp).

We created an acyclic and directed pangenome graph containing bubbles representing our variant callset (Methods and Extended Data Fig. 1). Sets of overlapping variant alleles are merged into a single bubble representing all alternative sequences observed across

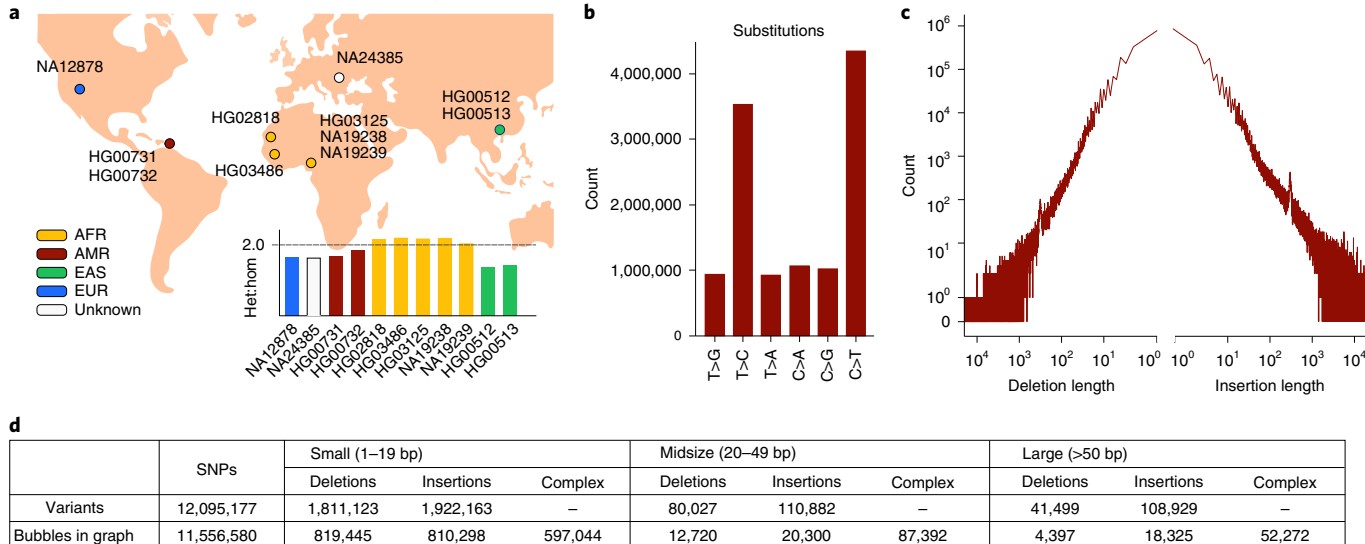

**Fig. 2 | Callset statistics. a**, Overview of the samples for which variants are called from haplotype-resolved assemblies as well as their het:hom ratios. Color corresponds to the population from which the samples originate. **b**, The number of different substitutions reported for all samples. **c**, Length distribution of insertions and deletions across all samples (in basepairs). **d**, Total number of distinct variant alleles detected across all 11 samples (first row), as well as the number of bubbles in the corresponding pangenome graph (second row). We distinguished small (1–19 bp), midsize (20–49 bp) and large (≥50 bp) variants. Biallelic bubbles were classified as SNPs, insertions or deletions; complex corresponds to all remaining bubbles with more than two branches resulting from inserting overlapping variant calls into the graph.

|  | SNPs | Small (1–19 bp) | | | Midsize (20–49 bp) | | | Large (>50 bp) | | |
|---|---|---|---|---|---|---|---|---|---|---|
|  |  | Deletions | Insertions | Complex | Deletions | Insertions | Complex | Deletions | Insertions | Complex |
| Variants | 12,095,177 | 1,811,123 | 1,922,163 | – | 80,027 | 110,882 | – | 41,499 | 108,929 | – |
| Bubbles in graph | 11,556,580 | 819,445 | 810,298 | 597,044 | 12,720 | 20,300 | 87,392 | 4,397 | 18,325 | 52,272 |

the haplotypes (Fig. 2d). The haplotypes themselves are represented as paths through the resulting pangenome. We distinguish biallelic from complex bubbles. The latter corresponds to bubbles with more than two branches and the former to all bubbles with two branches (reference and alternative sequence). Based on the type of bubbles, we define genomic regions as 'biallelic' or 'complex' (Methods and Extended Data Fig. 1).

**Comparison to existing genotyping methods.** We conducted a 'leave-one-out experiment' (Methods and Extended Data Fig. 2) based on Illumina reads from the Genome in a Bottle (GIAB) consortium[45] and 1000 Genomes Project high-coverage data[46]. In the same way as described above, we created a callset containing variants detected from haplotype-resolved assemblies of a subset of ten samples and re-genotyped these variants using Illumina data of the remaining sample. Variants called from the assemblies of the left-out sample are used as the ground truth for evaluation. We ran this experiment twice, leaving out samples NA12878 and NA24385 for evaluation, respectively. In addition to running PanGenie, we ran BayesTyper[32] (*k*-mer based), Platypus[19], GATK HaplotypeCaller[16], GraphTyper[22], Paragraph[25] and Giraffe[27] (all mapping based) to re-genotype the same set of variants (Methods and Extended Data Fig. 2). We ran our experiments on coverage levels 30×, 20×, 10× and 5×.

Not all tools are designed to handle all types of variants. Therefore, we ran GATK only on SNPs, small and midsize variants and Paragraph only on midsize and large variants. GraphTyper and Giraffe were run on large variants only.

Results for NA12878 (Fig. 3 and Extended Data Figs. 3–8) and NA24385 (Supplementary Figs. 4–9) are similar, showcasing consistency of results across samples. To analyze genotyping performance, we introduced the weighted genotype concordance (wGC) which puts equal emphasis on the ability to detect all three possible genotypes (Supplementary Note). As an alternative view on the performance of the individual methods, we offer precision, recall and *F* score, all in an unadjusted version and an adjusted version that

does not penalize methods for 'missing' variants that are undetectable because they are not in the input set (Supplementary Note). Furthermore, we stratify our analyses by considering variants outside and inside short-tandem repeats (STRs) and variable-number tandem repeats (VNTRs)[47]. We annotated variants according to their repeat status and observed that between 68% and 72% of midsize (20–49 bp) and large variants (≥50 bp) are repeat associated, respectively (Supplementary Table 4). We consider two configurations for PanGenie: 'high-gq' filtering, where we use only genotypes reported with high quality scores and treat all other variants as not genotyped, and 'all', where we consider all reported genotypes regardless of their quality.

For biallelic SNPs in nonrepetitive regions, all methods reach excellent levels of genotype concordance (Extended Data Fig. 3) and *F* scores (Extended Data Fig. 7), with all *F* scores >0.95 at coverage 30×. For biallelic SNPs in repetitive regions, PanGenie still achieves an *F* score of 0.85, whereas the second-best tool GATK reaches only 0.75 (Extended Data Fig. 8). In repetitive regions, BayesTyper has the largest fraction of untyped SNPs of all tools, resulting in lowest recall of 0.6 for biallelic SNPs and 0.17 for SNPs inside of complex bubbles (Extended Data Fig. 6).

For small insertions and deletions, PanGenie ('all') outperforms the mapping-based approaches, in particular in STR/VNTR regions (wGC of 90.4% for insertions and 92.8% for deletions; Extended Data Fig. 4), where the best mapping-based tools (GATK) achieved a wGC of 83% and 86.9% for biallelic insertions and deletions, respectively, at coverage 30×. BayesTyper and PanGenie using 'high-gq' filtering achieved the highest wGCs, both >99% for nonrepetitve (Extended Data Fig. 3) and >97% for repetitive regions (Extended Data Fig. 4). For both tools, these good wGCs came at the expense of relatively few genotyped variants, with PanGenie being able to genotype slightly more. We also evaluated our results for SNPs, small and midsize variants using the GIAB high-confidence small variant callset[48] as a ground truth (Supplementary Fig. 11).

Performance differences were largest for midsize and large variants (Fig. 3). PanGenie clearly outperforms the mapping-based

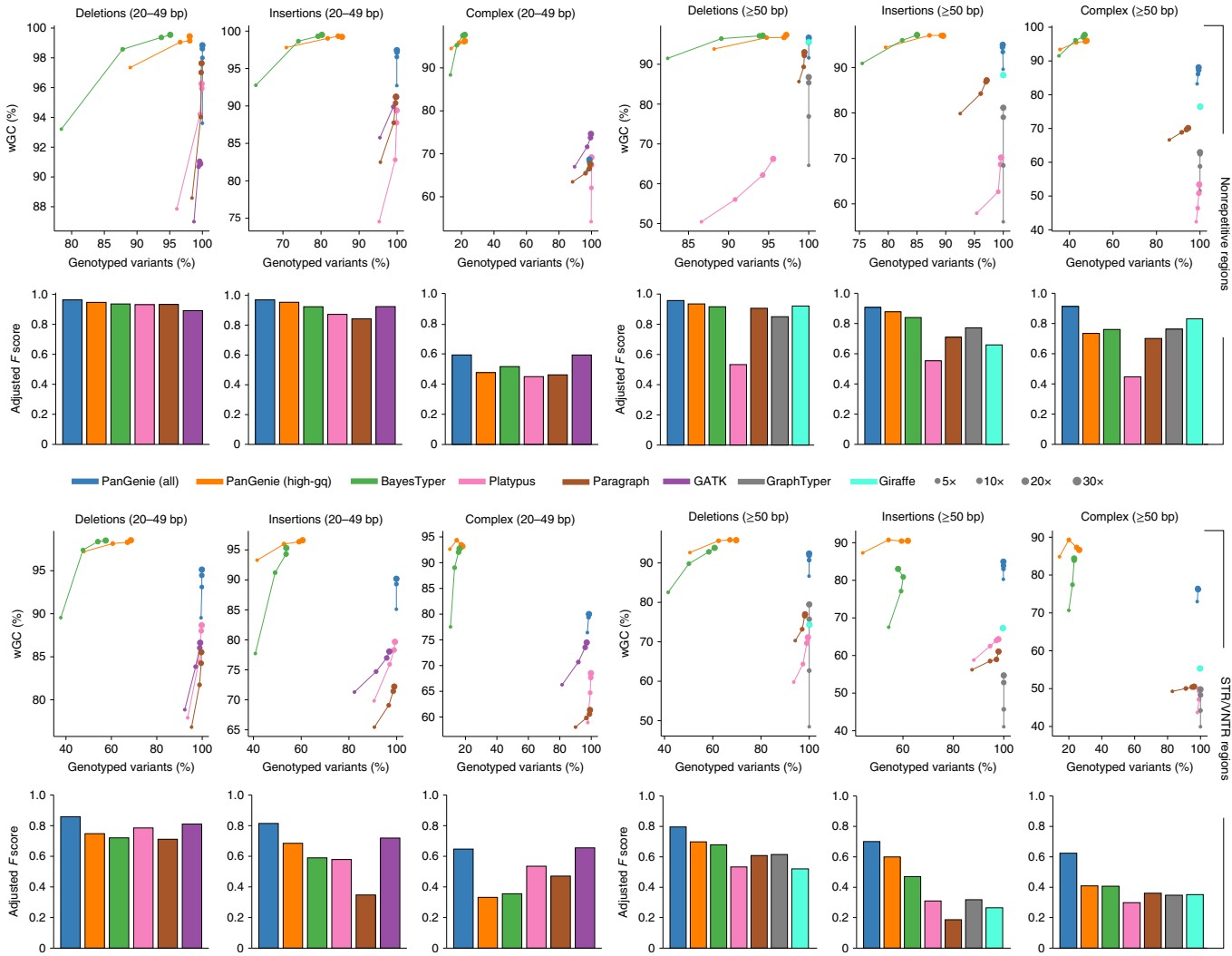

**Fig. 3 | Leave-one-out experiment.** The wGC at different coverages for sample NA12878 and $F$ scores for coverage 30× in nonrepetitive (top) and STR/VNTR regions (bottom). We ran PanGenie, BayesTyper, Paragraph, Platypus, GATK, GraphTyper and Giraffe to re-genotype all callset variants. Besides not applying any filter on the reported genotype qualities ('all'), we additionally report genotyping statistics for PanGenie when using 'high-gq' filtering (genotype quality ≥200). Insertions and deletions include all respective variants in biallelic regions of the genome, whereas complex contains all variant alleles falling into regions with complex bubbles in the pangenome graph representation.

approaches, especially in repeat regions. Here, PanGenie ('all') reaches wGCs for large SVs of 85%, 92% and 76% for biallelic insertions, biallelic deletions and variants in complex multiallelic regions, respectively, at coverage 30×. This is in contrast with the performance of the best mapping-based tool, achieving only 64%, 79% and 51%, respectively. BayesTyper reached high wGCs, but left 42%, 39% and 77% of these variants untyped, respectively. Using 'high-gq' filtering, PanGenie can reach concordances similar or superior to BayesTyper, while still being able to type much larger fractions of variants (Fig. 3). PanGenie's genotyping performance for large SVs in repetitive regions is underscored also by the $F$ score (Fig. 3): for large biallelic insertions, for example, PanGenie ('all') shows an $F$ score of 0.7 whereas all other tools reach $F$ scores <0.5. We additionally used the SVs contained in the syndip benchmark set[49] to evaluate genotyping performance. Although the absolute results tend to be slightly worse for all tools, PanGenie again produced the most accurate genotype predictions and outperformed the other tools (Supplementary Fig. 12).

*Runtimes.* For each method, we measured the time required to produce genotypes given variants and raw, unaligned sequencing reads

(Supplementary Table 5). The $k$-mer-based methods PanGenie and BayesTyper were much faster compared with the remaining, mapping-based methods that were combined with BWA[50] for read mapping. PanGenie was fastest on all coverages, being between 3.97× and 4.6× faster than the fastest mapping-based approach at 30×.

**Accuracy in the major histocompatibility complex.** To evaluate the accuracy of all 14 haplotype-resolved assemblies in the human leukocyte antigen (HLA) region, we used HLA*ASM[51] to determine assembly HLA types (Supplementary Table 6). HLA*ASM successfully processed 27 of 28 input assemblies and identified perfect (edit distance 0) HLA-G group matches[52] for all classic HLA loci (*HLA-A, -B, -C, -DQA1, -DQB1* and *-DRB1*) in all processed input assemblies with one exception (*HLA-DRB1* in NA19238), which was resolved by manual curation with minimap2[53]. To verify the accuracy of the assembly HLA types, we integrated publicly available HLA genotype data for samples from the 1000 Genomes Project[54–56] for *HLA-A, -B, -C, -DQB1* and *-DRB1*, intersected these with the assembly-implied HLA types, and found perfect agreement in all evaluated cases (9 samples and 85 individual genotype comparisons; Supplementary Table 6).

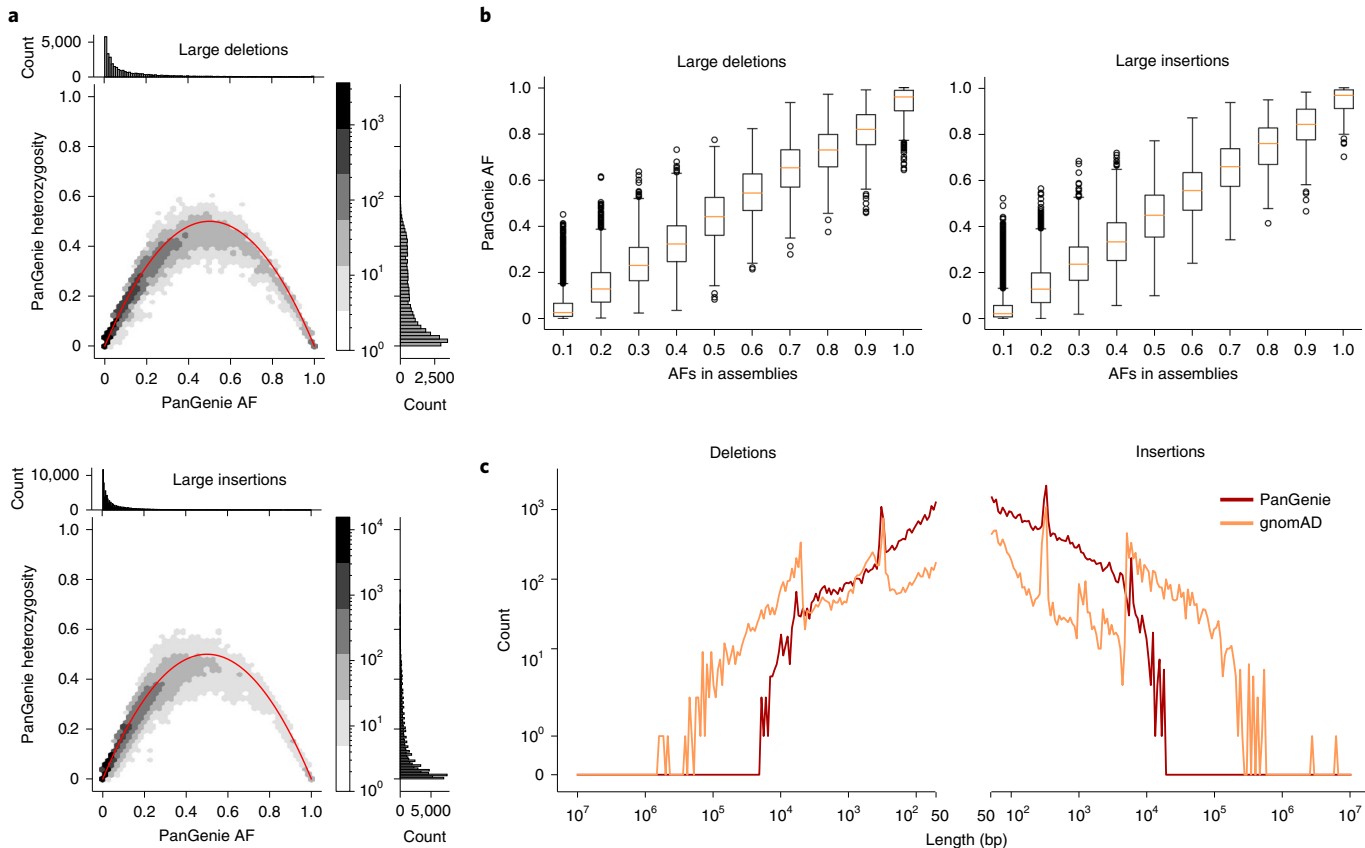

**Fig. 4 | Genotyping large cohorts. a**, The hexbin plots show the relationship between AFs and heterozygosities of the PanGenie genotypes for all 200 unrelated samples from the 1000 Genomes Project. The barplots show the one-dimensional distributions of both features (top: AF, right: heterozygosity). All large insertions (≥50 bp, $n = 84,836$) and deletions (≥50 bp, $n = 34,290$) contained in our lenient set were taken into account. **b**, Comparison of AFs computed from the PanGenie genotypes for 200 samples and the corresponding AFs observed in the 11 assembly samples from which variants were called. As in **a**, we consider all large insertions (≥50 bp, $n = 84,836$) and deletions (≥50 bp, $n = 34,290$) contained in our lenient set. In the boxplots, lower and upper limits of the box represent the lower and upper quartiles (Q1 and Q3); the median is marked in yellow. Lower and upper whiskers are defined as $Q1 − 1.5 (Q3−Q1)$ and $Q3 + 1.5 (Q3−Q1)$, respectively, and outliers are marked by dots. **c**, Length distribution of the number of common insertions and deletions (AF ≥ 5%) contained in the PanGenie lenient callset and gnomAD.

We additionally evaluated PanGenie's genotyping performance in the HLA region based on a 'leave-one-out' experiment for samples HG00731, NA12878 and NA24385, and observed high levels of wGCs across commonly studied HLA genes. Although the average wGC across all three samples was lowest for *HLA-DRB1* and *-C4* (58% and 79%, respectively in biallelic regions), it was between 98% and 100% for *HLA-C*, *-DPA1*, *-DPB1* and *-DRA* in biallelic regions, and between 93% and 100% for all variants in complex regions (Extended Data Fig. 9).

**Genotyping larger cohorts.** The low runtime of PanGenie makes it well suited to genotype larger cohorts. As an example use case, we applied it to a set of 300 samples consisting of 100 randomly selected trios from the 1000 Genomes Project using high-coverage data[46]. We used our pangenome graph containing all 2 × 11 haplotypes to compute genotypes for all detected variants. Similar to the approach introduced previously[4], we employed Mendelian consistency of the genotyped trios and the genotype quality reported by PanGenie to compute an integrated score for genotyping reliability of each variant. To this end, we defined different filters for a positive set with the most reliable (termed 'strict' set) and a negative set with the most unreliable variants. Using a machine-learning approach trained on these two subsets, we computed scores for all remaining variants, reflecting how confident we were about their genotyp-

ing, and used those to derive a 'lenient' set of variants containing 78% and 83% of all insertion SVs and deletion SVs, respectively (Supplementary Note and Supplementary Table 8). To confirm that the lenient set still offers very good genotyping performance, we analyzed allele frequencies and heterozygosities observed from the predicted genotypes for all variants in the lenient set and observed a relationship close to what is expected from the Hardy–Weinberg equilibrium (HWE; Fig. 4a and Methods). When testing for HWE, 90.7% of SV alleles inside of repeats, and 90.9% outside of repeats, showed no significant deviation. Furthermore, observed allele frequencies (AFs) across all 200 unrelated samples are in excellent agreement with coarse-grained AF estimates obtained from the 22 haplotype assemblies of our 11 input samples (Fig. 4b). Note that neither of these two measures, HWE and agreement in estimated AFs, has been used when defining the lenient set and therefore serves as independent evidence for PanGenie's performance. PanGenie on average only took about 30 single-core CPU hours per sample.

Our callset contains 209 of 250 medically relevant SVs reported by GIAB[57]. We observed that 174 medically relevant SVs were contained in our lenient set, of which 119 were part of our strictly filtered set. We show the score distribution for these variants as well as AFs and heterozygosities observed across all 200 unrelated samples for the lenient set in Extended Data Fig. 10.

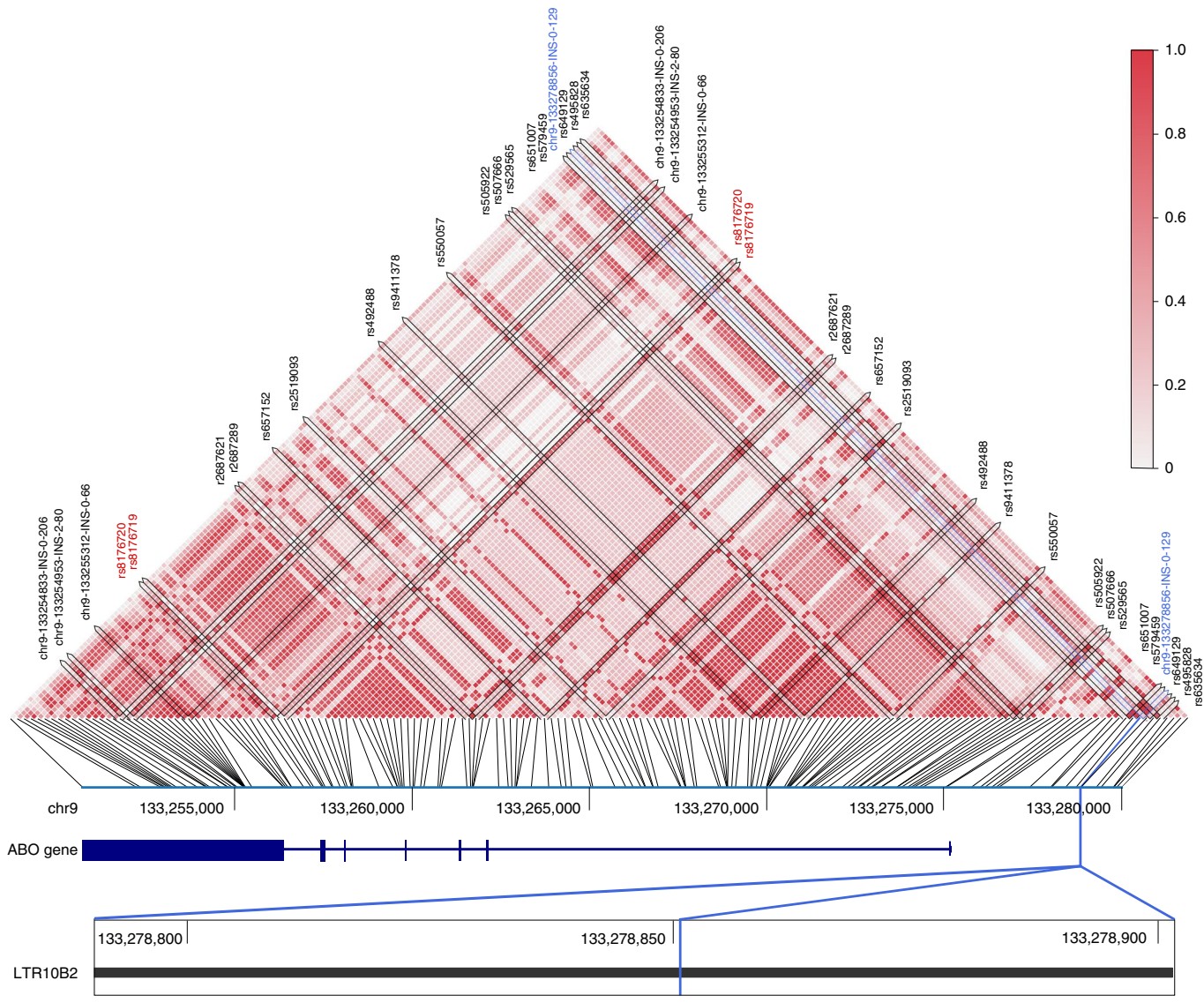

**Fig. 5 | LD analysis.** We calculated the LD for GWAS variants and SVs that were part of our assembly-based callset. We detected an insertion (marked in blue) close to the *ABO* gene which was in LD with six GWAS SNPs. The plots show all callset variants in this region; GWAS variants are annotated with their name. Those variants colored in red correspond to blood-type markers.

**Comparison to gnomAD.** We compared the 119,126 SV alleles genotypable by PanGenie (lenient set) with the SVs that are part of the Genome Aggregation Database (gnomAD)[9]; gnomAD contains SVs collected across 14,891 genomes from different populations[9]. Requiring a reciprocal overlap of at least 50% or a start, end and variant length deviation of <200 bp, we found that both callsets had 34,468 variants in common (Supplementary Note), whereas 84,658 (71%) of our SV alleles were not contained in gnomAD. This finding is consistent with previous observations that short-read-based SV detection misses most SVs[35]. Of those 84,658 SVs, around 80% were located in STR/VNTR regions. Furthermore, 43% of these 84,658 variants were common variants with AF ≥ 0.05 across all genotyped samples. The length distribution of common insertions and deletions (Fig. 4c) demonstrates the ability of PanGenie to genotype variants in regions inaccessible by callers based on short-read data alone, and shows its particular impact when genotyping insertions and shorter deletions.

**LD analysis.** Based on the genotypes obtained across all 200 unrelated samples (Genotyping larger cohorts), we performed an LD analysis (Methods). We selected all SNPs from our callset that were contained at least five times in the NHGRI-EBI GWAS (genome-wide association studies) catalog[58]. For each resulting variant, we calculated LD, comparing it with all our callset variants within a window of 1 Mb.

For 147 of 3,404 disease-associated SNPs from NHGRI-EBI, we found nearby structural variants that were in LD ($r^2 \geq 0.8$; see Supplementary Table 9 for all hits with $r^2 \geq 0.9$). An insertion of length 129 bp located at position 133,278,856 on chromosome 9, close to the *ABO* gene, looked particularly interesting (Fig. 5). It is in LD with six GWAS variants (rs2519093, rs495828, rs507666, rs579459, rs635634 and rs651007) which are related to low-density lipoprotein-cholesterol levels[58]. Of note, neither the GWAS SNPs nor the insertions are in LD with blood-type markers present in our callset (rs8176747 (ref. [59]), rs8176746 (ref. [60]), rs8176743 (ref. [59]), rs8176742 (ref. [61]), rs8176741 (ref. [61]), rs8176740 (ref. [61]), rs7853989 (ref. [61]), rs1053878 (ref. [61]), rs8176720 (ref. [61]) and rs8176719 (ref. [60])). The insertion is located in a long tandem repeat (LTR10B2 for ERV1 endogenous retrovirus). Analysis of the insertion sequence revealed that it contains three exact copies of a 43-bp sequence (Methods and

Supplementary Fig. 19), which appears with copy number 1 in the reference genome. We thus conclude that this insertion is a repeat expansion, leading to four copies of this repeated subsequence. A comparison with nonhuman primate genomes[62,63] shows that the 43-mer occurs as two copies in gorilla (*Gorilla gorilla*), but is a single copy in chimpanzee (*Pan troglodytes*), bonobo (*Pan paniscus*) and the Sumatran orangutan (*Pongo abelii*). This suggests independent expansion events or incomplete lineage sorting in humans and gorillas.

Another interesting association was an intronic insertion of length 322 bp located at position 28,264,365 on chromosome 12, inside the *CCDC91* gene close to a regulatory element reported by ENCODE[64] (Supplementary Fig. 20). It was in LD with two GWAS variants (rs10843151 and rs11049566), which are both linked to body fat[58]. One of these SNPs, rs10843151, is in perfect LD with many other variants in this region, which suggests that it is probably embedded in the same haplotype block. Such perfect LD provides further evidence that PanGenie is accurately genotyping new insertions within short-read sequencing data.

## Discussion

We presented an algorithm, PanGenie, that can leverage the long-range haplotype information inherent to a panel of assembled haplotypes in combination with read *k*-mer counts for genotyping an uncharacterized sample. Although we generated such pangenome reference panels from haplotype-resolved assemblies for the present study, generating these panels was not the main focus of this report and PanGenie is not restricted to panels created in this way. In fact, it can be applied to any acyclic genome graph with fully phased path information.

Traditionally, longer variants are especially difficult to genotype based on short reads only, because such variants are often located in repetitive or duplicated regions of the genome, leading to the difficulty of unambiguously aligning the reads. Approaches based on *k*-mers additionally lack connectivity information contained in the reads because they do not use the order of *k*-mers stemming from the same read or read pair. PanGenie overcomes these limitations of short reads because it incorporates long-range haplotype information inherent to the pangenome reference panel that it uses. In comparison to BayesTyper, a graph-based genotyper relying on *k*-mers, PanGenie genotypes a large fraction of variants not typable by the former. For SVs and indels, PanGenie clearly outperforms mapping-based approaches, which require alignments of reads to a reference genome. Compared with Paragraph, a graph-based method relying on such read alignments, PanGenie produces better genotyping results while additionally providing the ability to jointly genotype SNPs, indels and SVs. Our approach was faster than the other methods, especially when comparing with the mapping-based approaches. The fast runtime makes PanGenie well suited for genotyping larger cohorts, providing the basis for population genetic analysis. In the present study, we have presented an application to a cohort of 300 samples that suggests that SVs in LD with disease-associated SNPs may functionally underlie these associations.

We have hence presented a method that is both fast and leverages a haplotype-resolved pangenome reference to enable genotyping of otherwise inaccessible variants. Although we have tested it only on human data so far, PanGenie can be applied to any diploid genome once corresponding panels of high-quality phased assemblies become available for other species. Still, some limitations remain. Although PanGenie improves results over other methods in repetitive regions of the genome, genotyping within these remains challenging. Although biallelic variants are less problematic, more complex cases such as segmental duplications, α-satellite repeats or acrocentric DNA are hard to access because of the lack of unique *k*-mers, but also because such regions are still difficult to assemble.

Once a panel of telomere-to-telomere assemblies becomes available, future experiments can clarify which additional loci are amenable to genotyping with PanGenie.

Our model assumes that the unknown haplotypes of the sample to be genotyped are mosaics of the given panel haplotypes. Therefore, currently it cannot be used to genotype rare variants that are present only in the sample, but in none of the other haplotypes. We believe that there are exciting opportunities to develop methods to discover variation that our approach has not captured because it was not present in the reference panel. For example, one could either filter the reads for as yet 'unexplained' *k*-mers and use those for the discovery of rare variants, or utilize PanGenie's output as a personalized pangenome reference graph to map reads to.

The runtime of our method depends on the number of input haplotypes, because we defined a hidden state for each possible pair of haplotypes that can be selected for each bubble. Therefore, additional engineering would be required to use much larger panels, which could be approached similarly to how statistical phasing packages prune the solution space and/or proceed iteratively[65–67]. Such techniques could also pave the way toward a version of PanGenie for polyploid genomes, which would be prohibitively slow when implemented without such additional optimization.

In summary, we have presented a method that, in combination with high-quality phased reference assemblies, offers a powerful approach for genotyping and association studies, on ever-larger cohorts, for all variant types—including those currently understudied due to technical limitations.

## Online content

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

## Methods

**Sequencing data.** We used publicly available sequencing data from the GIAB consortium[45], 1000 Genomes Project high-coverage data[46] and Human Genome Structural Variation Consortium (HGSVC)[4]. All datasets include only samples consented for public dissemination of the full genomes.

**Statistics and reproducibility.** For generating the assemblies, we used all 14 samples for which PacBio HiFi-data were available. For variant calling, the three children (HG00733, HG00514 and NA19240) were used for quality control and were not included in the final callsets/graphs, because they do not provide any additional information for genotyping. Code and pipelines to reproduce our analysis are available on *Zenodo*[68,69].

**Variant calling and pangenome construction.** *Assemblies.* Fully phased assemblies for 14 samples (HG00731, HG00732, HG00733, HG00512, HG00513, HG00514, NA19238, NA19239, NA19240, NA12878, HG02818, HG03125, NA24385 and HG03486) were generated using a development version of the PGAS pipeline[2,4] (parameter settings v.13). Compared with the previous PGAS production release (v.12 used in the HGSVC project[4]), this PGAS development update included a new version of the SaaRclust package[70] (v.6cb8c96), controlled for adapter contamination in the input HiFi reads (reimplementation of the process published at https://github.com/sheinasim/HiFiAdapterFilt), and employed hifiasm[71] v.0.15.2 as default assembler. In direct comparison to the previously used HiFi assembler Peregrine[72], hifiasm substantially reduces the number of sequence collapses, leading to overall more correct assemblies (see the evaluation in Cheng et al.[71]). We provide assembly statistics in Supplementary Table 1.

*Variant calling.* We used haplotype-resolved assemblies of all 14 samples to call variants (Extended Data Fig. 1a). The three child samples (HG00733, HG00514 and NA19240) were used only for quality control and filtering, and thus were not part of our final callset/graph. For each sample, we separately mapped contigs of each haplotype (Supplementary Table 1) to the reference genome (GRCh38). This was done using minimap2 (ref. [53]) (v.2.18) with parameters `-cx asm20 -m 10000 -z 10000,50 -r 50000 --end-bonus=100 -O 5,56 -E 4,1 -B 5 --cs`. In the next step, we called variants on each haplotype of all autosomes and chromosome X using paftools (https://github.com/lh3/minimap2/tree/master/misc) with default parameters. We generated a biallelic, VCF file containing variant calls made across all 11 unrelated samples (Extended Data Fig. 1a). If a region was not covered by any contig alignment in a sample, or the sample had multiple overlapping contig alignments, we set all its genotypes in this region to missing ("./."), because it is unclear what the true genotype alleles are in this case. Furthermore, we removed variants from our callset for which >20% of the samples have missing genotype information. The remaining regions covered 91.8% (2.8 Gbp) of chromosomes 1–22 and chromosome X. Of the 8.2% of regions not covered, 48.3% were gaps in GRCh38 and 24.0% were centromeres.

We computed the Mendelian consistency for the Puerto Rican (HG00731, HG00732, HG00733), Chinese (HG00512, HG00513, HG00514) and Yoruban (NA19238, NA19239, NA19240) trios and observed that 97.9%, 96.8% and 97.6% of all variants were consistent with Mendelian laws, respectively. We removed a variant from our callset if there was a Mendelian conflict in at least one of the three trios. We show the number of variants in our final callset and the intermediate stages of variant calling in the first three columns of Supplementary Table 2.

*Pangenome construction.* Given the filtered variant calls, our goal was to construct an acyclic and directed graph by inserting the variants of all haplotypes into the linear reference genome. Variants produce bubbles in the graph with branches that define the corresponding alleles. The input haplotypes can be represented as paths through the resulting pangenome. When constructing the graph, we represent sets of variants overlapping across haplotypes as a single bubble, with potentially multiple branches reflecting all the allele sequences observed in the haplotypes in the respective genomic region (Extended Data Fig. 1b). The total number of bubbles in the resulting graph is presented in the last column of Supplementary Table 2. We represent the pangenome in terms of a fully phased, multisample VCF file that contains one entry for each bubble in the graph (Extended Data Fig. 1b). At each site, the number of branches of the bubble is limited by the number of input haplotype sequences and the genotypes of each sample define two paths through this graph, corresponding to the respective haplotypes. We keep track of which individual input variants contribute to each bubble in the graph, so that we can convert our pangenome graph representation back to the set of input variants. In this way, we can convert genotypes computed by a genotyper for all these bubbles to genotypes for each individual callset variants.

**PanGenie's genotyping algorithm.** We define a hidden Markov model that can be used to compute the two most likely haplotype sequences of a given sample based on known haplotype paths and the sample reads. The new haplotype sequences are combinations of the existing paths through the graph and are computed based on the copy numbers of unique *k*-mers observed in the sequencing reads provided for the sample to be genotyped.

*Identifying unique k-mers.* Sets of bubbles that are less than the *k*-mer size apart (we use $k = 31$) are combined and treated as a single bubble. The alleles corresponding to such a combined bubble are defined by the haplotype paths in the respective region. For each bubble position $v$, we determine a set of *k*-mers, $kmers_v$, that uniquely characterize the region. This is done by counting all *k*-mers along haplotype paths in the pangenome graph using Jellyfish[73] (v.2.2.10), and then determining a set of *k*-mers for each bubble that occurs at most once within a single allele sequence and are not found anywhere outside the variant bubble. We additionally counted all *k*-mers of the graph in the sequencing reads. This allows us to compute the mean *k*-mer coverage of the data, which we use later to compute emission probabilities (see Observable states).

*Hidden states and transitions.* We assume being given $N$ haplotype paths $H_i$, $i = 1, ..., N$, through the graph. Furthermore, for each bubble $v$, $v = 1, ..., M$, we are given a vector of *k*-mers, $kmers_v$, that uniquely characterize the alleles of a bubble. We assume some (arbitrary) order of the elements in $kmers_v$ and refer to the $i$th *k*-mer as $kmers_v[i]$. In addition, we are given sequencing data of the sample to be genotyped and corresponding *k*-mer counts for all *k*-mers in $kmers_v$. For each bubble $v$, we define a set of hidden states $\eta_v = \{H_{v,i,j} | i, j \leq N\}$ which contain a state for each possible pair of the $N$ given haplotype paths in the graph. Each such state $H_{v,i,j}$ induces an assignment of copy numbers to all *k*-mers in $kmers_v$. We define a vector $\mathbf{a}_{v,i,j}$ such that the $k$th position contains the copy number assigned to the $k$th *k*-mer in $kmers_v$:

$$
\mathbf{a}_{v,i,j}[k] = \begin{cases} 0 & kmers_v[k] \notin H_i \cup H_j \\ 1 & kmers_v[k] \in H_i \backslash H_j \\ 1 & kmers_v[k] \in H_j \backslash H_i \\ 2 & kmers_v[k] \in H_i \cap H_j. \end{cases} \quad \forall k = 1, ..., |kmers_v|
$$

The idea here is that we expect to see copy number 2 for all *k*-mers occurring on both haplotype paths. In case only one of the haplotypes contains a *k*-mer, its copy number must be 1 and *k*-mers that do not appear in any of the two paths must have copy number 0. From each state $H_{v,i,j}$ in $\eta_v$ that corresponds to bubble position $v$, there is a transition to each state corresponding to the next position, $v + 1$. In addition, there is a start state, from which there is a transition to each state of the first bubble, and an end state, to which there is a transition from each state that corresponds to the last bubble.

*Transition probabilities.* Transition probabilities are computed following the Li–Stephens model[37]. Given a recombination rate $r$, the effective population size $N_e$ and the distance $x$ (in basepairs) between two ascending bubbles $v - 1$ and $v$ we define:

$$
d = x \times \frac{1}{1,000,000} \times 4rN_e.
$$

We compute the Li–Stephens transition probabilities as:

$$
p_r = \left(1 - \exp\left(-\frac{d}{N}\right)\right) \times \frac{1}{N}
$$

$$
q_r = \exp\left(-\frac{d}{N}\right) + p_r.
$$

Finally, the transition probability from state $H_{v-1,k,l}$ to state $H_{v,i,j}$ is computed as shown below:

$$
P\left(H_{v,i,j} | H_{v-1,k,l}\right) = \begin{cases} q_r \times q_r & i = k, j = l \\ q_r \times p_r & i = k, j \neq l \\ q_r \times p_r & i \neq k, j = l \\ p_r \times p_r & i \neq k, j \neq l. \end{cases}
$$

*Observable states.* Each hidden state $H_{v,i,j}$ in $\eta_v$ outputs a count for each *k*-mer in $kmers_v$. Let $\mathbf{O}_v$ be a vector of length $|kmers_v|$ for bubble $v$ such that $\mathbf{O}_v[k]$ contains the observed *k*-mer count of the $k$th *k*-mer in the sequencing reads. To define the emission probabilities, we first need to model the distribution of *k*-mer counts for each copy number, $P\left(\mathbf{O}_v[k] | a_{v,i,j}[k] = c\right)$, $c = 0, 1, 2$. For copy number 2, we use a Poisson distribution with mean $\lambda$ which we set to the mean *k*-mer coverage that we compute from the *k*-mer counts of all graph *k*-mers. Similarly, we approximate the *k*-mer count distribution for copy number 1 in terms of a Poisson distribution with mean $\frac{\lambda}{2}$. For copy number 0, we need to model the erroneous *k*-mers that arise from sequencing errors. This is done using a geometric distribution, the parameter $p$ of which we choose based on the mean *k*-mer coverage. Finally, we compute the

emission probability for a given state and given observed read $k$-mer counts as shown below, making the assumption that the $k$-mer counts are independent:

$$P\left(\mathbf{O}_v|H_{v,i,j}\right) = \prod_{l=1}^{|kmers_v|} P\left(\mathbf{O}_v\left[l\right]|\mathbf{a}_{v,i,j}\left[l\right]\right).$$

*Genotypes and haplotypes.* In this model, genotypes correspond to pairs of given haplotype paths at each bubble position. Genotype likelihoods can be computed using the forward–backward algorithm.

*Forward–backward algorithm.* The initial distribution of our HMM is such that we assign probability 1 to the start state and 0 to all others. Forward probabilities $\alpha_v()$ are computed in the following way:

$$\alpha_0\left(\text{start}\right) = 1.$$

For states corresponding to bubbles $v = 1, …, M$, the forward probabilities are computed as shown below. The set of observed $k$-mer counts at position $v$ is given by $\mathbf{O}_v$:

$$\alpha_v\left(H_{v,i,j}\right) =$$
$$\sum_{H_{v-1,s,t}\in\eta_{v-1}} \alpha_{v-1}\left(H_{v-1,s,t}\right) \times P\left(H_{v,i,j}|H_{v-1,s,t}\right) \times P\left(\mathbf{O}_v|H_{v,i,j}\right) \forall i, j.$$

The transition probabilities are computed as described above, except for transitions from the start state to all states in the first column, which we assume to have uniform probabilities. Backward probabilities are computed in a similar manner. We set:

$$\beta_M\left(\text{end}\right) = 1.$$

For $v = 1, …, M - 1$, we compute them as:

$$\beta_v\left(H_{v,i,j}\right) =$$
$$\sum_{H_{v+1,s,t}\in\eta_{v+1}} \beta_{v+1}\left(H_{v+1,s,t}\right) \times P\left(H_{v+1,s,t}|H_{v,i,j}\right) \times P\left(\mathbf{O}_{v+1}|H_{v+1,s,t}\right) \forall i, j.$$

Finally, posterior probabilities for the states can be computed:

$$P\left(H_{v,i,j}|\mathbf{O}_1, \mathbf{O}_2, …, \mathbf{O}_M\right) = \frac{\alpha_v\left(H_{v,i,j}\right) \times \beta_v\left(H_{v,i,j}\right)}{\sum_{h\in\eta_v} \alpha_v\left(h\right) \beta_v\left(h\right)}.$$

Several states at a bubble position $v$ can correspond to the same genotype, because different paths can cover the same allele. Also, the alleles in a genotype are unordered, therefore states $H_{v,i,j}$ and $H_{v,j,i}$ always lead to the same genotype. To compute genotype likelihoods, we sum up the posterior probabilities for all states that correspond to the same genotype. In this way, we can compute genotype likelihoods for all genotypes at a bubble position, based on which a genotype prediction can be made.

**Comparison to existing genotyping methods.** We conducted a 'leave-one-out' experiment to mimic a realistic scenario in which we genotyped variants detected from haplotype-resolved assemblies of a set of known samples in a new, unknown sample. We collected variants called across all but one sample and used them as input for genotyping the left-out sample (we refer to this set as known variants in the following). We used the set of variants called from the assemblies of the left-out sample for evaluation (evaluation variants). We ran this experiment twice, removing samples NA12878 and NA24385, respectively. As input for PanGenie (commit 1f3d2d2 (ref. [68])), BayesTyper (v.v1.5) and Paragraph (v.2a), we constructed a pangenome graph representation based on the known variants in the same way as described in Constructing a pangenome reference. We kept track of which variant alleles each resulting bubble consists of, so that genotypes derived for all bubbles can later be converted back to the original variant representation. For the other genotypers tested (GATK 4.1.3.0, Platypus 0.8.1, GraphTyper 2.7.1 and Giraffe v.1.30.0), we directly used the set of known variants as input, without generating the graph representation first, because we observed that these tools could better handle variants represented in this way. As a result of running all genotypers, we had one VCF file per tool containing genotypes for all our known variants. We used the evaluation variants to evaluate the genotype predictions of all tools. Extended Data Fig. 2 provides an illustration of the leave-one-out experiment.

Note that re-genotyping a set of known variants in a new sample is different from variant detection. Variants present in the new sample that have not been seen in the callset samples can thus not be genotyped because genotypers can genotype only variants that they have seen before. We provide the number of unique variants of each panel sample in Supplementary Table 3. Most genotyping metrics (weighted genotyping concordance, adjusted precision/recall) explained in detail in Supplementary Note exclude these variants.

Besides re-genotyping our callset variants, we additionally ran GATK and Platypus in discovery mode to detect and genotype their own variants. We evaluated the results by computing precision/recall based on our ground-truth variants (Supplementary Figs. 10 and 11).

*Evaluation regions.* Some genomic regions are more difficult to genotype than others, such as SVs that tend to be located in repetitive and more complex regions of the genome. Therefore, we looked at variants located inside and outside of STR/VNTR regions which we obtained from the UCSC genome browser (Simple Repeats Track for GRCh38)[47]. In addition, we classified the genome into 'complex' and 'biallelic' regions based on the bubble structure of our pangenome graph: all variants located inside of complex bubbles, that is, bubbles with more than two branches, fell into the first category, and the remaining regions into the second. Consider Extended Data Fig. 1 for an example: the first and third bubbles are complex, thus all variants contained inside these bubbles fall into the category 'complex'. The second bubble is biallelic and therefore the corresponding SNP variant is considered 'biallelic'.

For our 'leave-one-out' experiment for sample NA12878, we show the number of variants falling into the different categories in Fig. 3, Extended Data Figs. 3–8 and Supplementary Table 4. It can be observed that most complex bubbles are located inside STR/VNTR regions (Supplementary Table 4). In addition, more than half of all midsize and large variants are located in these repetitive regions.

**Genotyping larger cohorts.** We randomly selected 100 trios (20 of each superpopulation: AFR, AMR, EAS, EUR, South Asian (SAS)) that are part of the 1000 Genomes Project and genotyped all our variant calls across these 300 samples. We used our pangenome graph representation containing all 11 assembly samples as an input for PanGenie, genotyped all bubbles and later converted the resulting genotypes back to obtain genotypes for the individual callset variants. Our callset might contain variants that are difficult to genotype correctly. To identify a high-quality subset of variants that we could reliably genotype, we defined different filters based on the predicted genotypes that we list below. One metric used for defining filters is the Mendelian consistency. We computed the Mendelian consistency for each variant by counting the number of trios for which the predicted genotypes are consistent with Mendelian laws. We considered only trios with at least two different genotypes, that is, we excluded a trio if all three genotypes were 0/0, 0/1 or 1/1. This resulted in a more strict definition of Mendelian consistency (Supplementary Fig. 14). In addition to genotyping all 300 trio samples, we also genotyped all 11 panel samples using the full input panel. Genotyping samples that are also in the panel helped us to find cases where panel haplotypes and reads disagreed and thus was another useful filter criterion. We defined filters as follows: (1) ac0-fail: a variant fails this filter if it was genotyped with AF 0.0 across all samples; (2) mendel-fail: a variant fails this filter if the fraction of Mendelian consistent trios was <90% (our definition of Mendelian consistency excludes all trios with all 0/0, all 0/1 or all 1/1 genotypes and only considers such with at least two different genotypes); (3) gq-fail: a variant failed this filter if it was genotyped with a genotype quality <200 in >5 samples; (4) self-fail: in addition to the 100 trios, we also genotyped the 11 panel samples; a variant failed this filter if the genotype concordance across all panel samples was <90%; and (5) non-ref-fail: the variant was genotyped as 0/0 across all panel samples.

For all combinations of filters, we show the number of large deletions and large insertions in each category in Supplementary Fig. 15. To define a strict, high-quality set of variants, we selected all that passed all five filters (Supplementary Table 7).

For quality control, we analyzed allele frequencies and the fraction of heterozygous genotypes for all variants contained in our unfiltered and strict sets (Supplementary Figs. 16 and 17). In addition, we used VCFTools[74] (v.0.1.16) to test the genotype predictions of all variants typed with an AF > 0.0 for conformance with the HWE and corrected for multiple hypothesis testing by applying the Benjamini–Hochberg correction[75] ($\alpha = 0.05$).

In addition to defining a strict set, we constructed a more lenient set for our SV calls ($\geq 50\,$bp) using a machine-learning approach based on support vector regression. We used the strict set as a positive set and defined a negative set consisting of all variants that were typed with an AF > 0.0 and failed at least three filters. For large insertions, the negative set contained 2,611 variants, and for large deletions 1,125. The model then predicted scores between −1 (worst) and 1 (best) for all variants that were in neither the positive nor the negative set. We show the distribution of scores for our variant calls in Supplementary Fig. 18. The lenient set was then constructed by adding all variants with a score >−0.5 to our strict SV set (Supplementary Table 8 and Supplementary Fig. 18).

**LD analysis.** We performed an LD analysis based on the genotypes we obtained across all 200 unrelated samples. We used `gatk4` (ref. [16]) (v.4.1.9.0) to annotate the calls with variant IDs from dbSNP (build 154)[76]. We selected variants that are contained in the NHGRI-EBU GWAS catalog[58] and used `plink`[77] (v.190b618) to determine SVs that are in LD with the GWAS variants ($r^2 \geq 0.8$). For comparison with other nonhuman primates, human genomic sequence (GRCh38; chr9:133278657-133279020) corresponding to 50 bp flanking the annotated

LTR10B2 VNTR was used to retrieve the corresponding orthologous sequence from primate genomes[62] or HiFi PacBio sequence data from nonhuman primates[63]. Multiple sequence alignments were constructed using MAFFT and manually inspected for VNTR copy number.

**Reporting Summary.** Further information on research design is available in the Nature Research Reporting Summary linked to this article.

## Data availability
Illumina short reads for NA24385 were downloaded from: https://ftp-trace.ncbi. nlm.nih.gov/ReferenceSamples/giab/data/AshkenazimTrio/HG002_NA24385_ son/NIST_Illumina_2x250bps/reads. For 1000 Genomes samples, Illumina short reads were downloaded from the National Center for Biotechnology Information's Search Read Archive (BioProject, accession no. PRJEB31736). For syndip, reads were downloaded from ftp://ftp.sra.ebi.ac.uk/vol1/fastq/ERR134/006/ERR1341796. The GIAB small variant benchmark was downloaded from ftp://ftp-trace.ncbi.nlm. nih.gov/giab/ftp/release/NA12878_HG001/NISTv3.3.2. GIAB medically relevant SVs were obtained from ftp://ftp-trace.ncbi.nlm.nih.gov/ReferenceSamples/giab/ data/AshkenazimTrio/analysis/NIST_HG002_medical_genes_SV_benchmark_ v0.01. The syndip benchmark variants were downloaded from https://github. com/lh3/CHM-eval/releases (v.20180222). GnomAD variants were downloaded from https://gnomad.broadinstitute.org/downloads (v.2). Haplotype-resolved assemblies, variant calls and genotypes produced in the present study are available from: https://doi.org/10.5281/zenodo.5607680 (ref. [78]). For generating haplotype-resolved assemblies, we used sequencing data published in ref. [4].

## Code availability
The implementation of PanGenie is available at: https://github.com/eblerjana/ pangenie. Code to reproduce the data and rerun the analysis is available at: https://bitbucket.org/jana_ebler/genotyping-experiments/src/master. The versions used for the experiments in this report are additionally available at https://doi.org/ 10.5281/zenodo.5767765 and https://doi.org/10.5281/zenodo.5864867.

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

## Acknowledgements
We thank Z. Iqbal for fruitful discussions on an earlier prototype of PanGenie. We thank D. Porubsky for his contributions to the PGAS pipeline. Computational infrastructure and support were provided by the Centre for Information and Media Technology at Heinrich Heine University Düsseldorf. This work was supported in part by grants from the National Institutes of Health (grant nos. 5U24HG007497 and 1U01HG010973 to T.M., E.E.E. and J.O.K., and R01HG002385 and HG010169 to E.E.E.) and the German Federal Ministry for Research and Education (grant nos. BMBF 031L0184 to A.T.D, J.O.K. and T.M., and BMBF 031L0181A to J.O.K.). This work was supported by the BMBF-funded de.NBI Cloud within the German Network for Bioinformatics Infrastructure (de.NBI) (nos. 031A532B, 031A533A, 031A533B, 031A534A, 031A535A, 031A537A, 031A537B, 031A537C, 031A537D and 031A538A). The funders had no role in study design, data collection and analysis, decision to publish or preparation of the manuscript.

## Author contributions
J.E. and T.M. developed the algorithms and designed the study. J.E. implemented PanGenie and the pangenome graph construction. P.E. generated the assemblies. T.H. and A.T.D provided ideas for graph construction and preliminary versions of the graph. J.E. and T.M. designed the experiments. J.O.K., P.A.A., E.E.E., M.C.Z., T.R., A.T.D., W.E.C. and T.H. contributed ideas and suggestions. J.E. performed all experiments. W.E.C. helped with the Paragraph evaluation. Y.M., P.A.A. and E.E.E. provided the results for nonhuman primates discussed in the LD analysis. T.H and A.T.D. evaluated the assemblies in the HLA region. J.E. and T.M. wrote a draft of the paper and all authors contributed edits and comments. All authors approved the final manuscript.

## Competing interests
The authors declare no competing interests.

## Additional information
**Extended data** Extended data are available for this paper at https://doi.org/10.1038/ s41588-022-01043-w.

**Correspondence and requests for materials** should be addressed to Tobias Marschall.

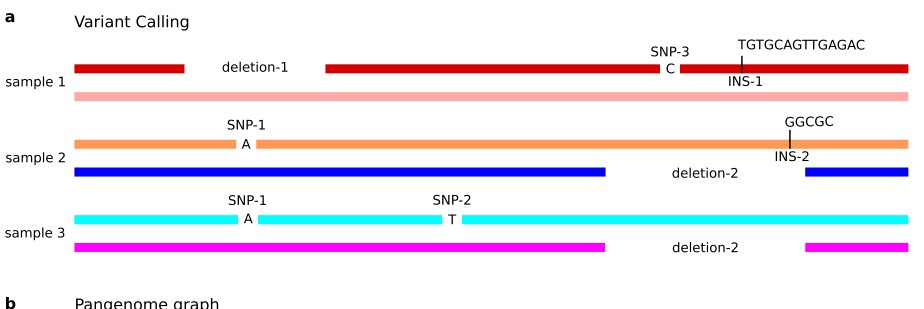

**a**  Variant Calling

| | ALT | sample 1 | sample 2 | sample 3 |
|---|---|---|---|---|
| **variant 1** | deletion-1 | 1|0 | 0|0 | 0|0 |
| **variant 2** | SNP-1 | 0|0 | 1|0 | 1|0 |
| **variant 3** | SNP-2 | 0|0 | 0|0 | 1|0 |
| **variant 4** | deletion-2 | 0|0 | 0|1 | 0|1 |
| **variant 5** | SNP-3 | 1|0 | 0|0 | 0|0 |
| **variant 6** | INS-1 | 1|0 | 0|0 | 0|0 |
| **variant 7** | INS-2 | 0|0 | 1|0 | 0|0 |

**b**  Pangenome graph

| | ALT | sample 1 | sample 2 | sample 3 |
|---|---|---|---|---|
| **bubble 1** | deletion-1, SNP-1 | 1|0 | 2|0 | 2|0 |
| **bubble 2** | SNP-2 | 0|0 | 0|0 | 1|0 |
| **bubble 3** | SNP-3:INS-1, INS-2, deletion-2 | 1|0 | 2|3 | 0|3 |

**Extended Data Fig. 1 | Variant calling and graph construction. a)** Shown are haplotype-resolved assemblies for three samples and corresponding variant calls made relative to a reference genome. On the right, we show how these variants are represented in a VCF file (simplified). The VCF file is biallelic and contains one record per (distinct) variant allele detected across the assemblies. **b)** Shown is the pangenome representation of the variants detected in panel a). Variants are represented as bubble structures. Sets of overlapping variants are merged into a single multi-allelic bubble (see first and last bubble for examples). Each haplotype can be represented as a path through the graph. We represent the pangenome in terms of a VCF file containing a record for each bubble and alleles corresponding to the branches of the bubble (right). We keep track of which callset variants each branch of the bubble was constructed from as illustrated in the VCF representation. In this way, we can later convert genotypes derived for a bubble back to genotypes for each individual variant inside of a bubble. Note that our VCFs contain the actual allele sequences in their 'ALT' column, we replaced them by their IDs in this figure for simplicity.

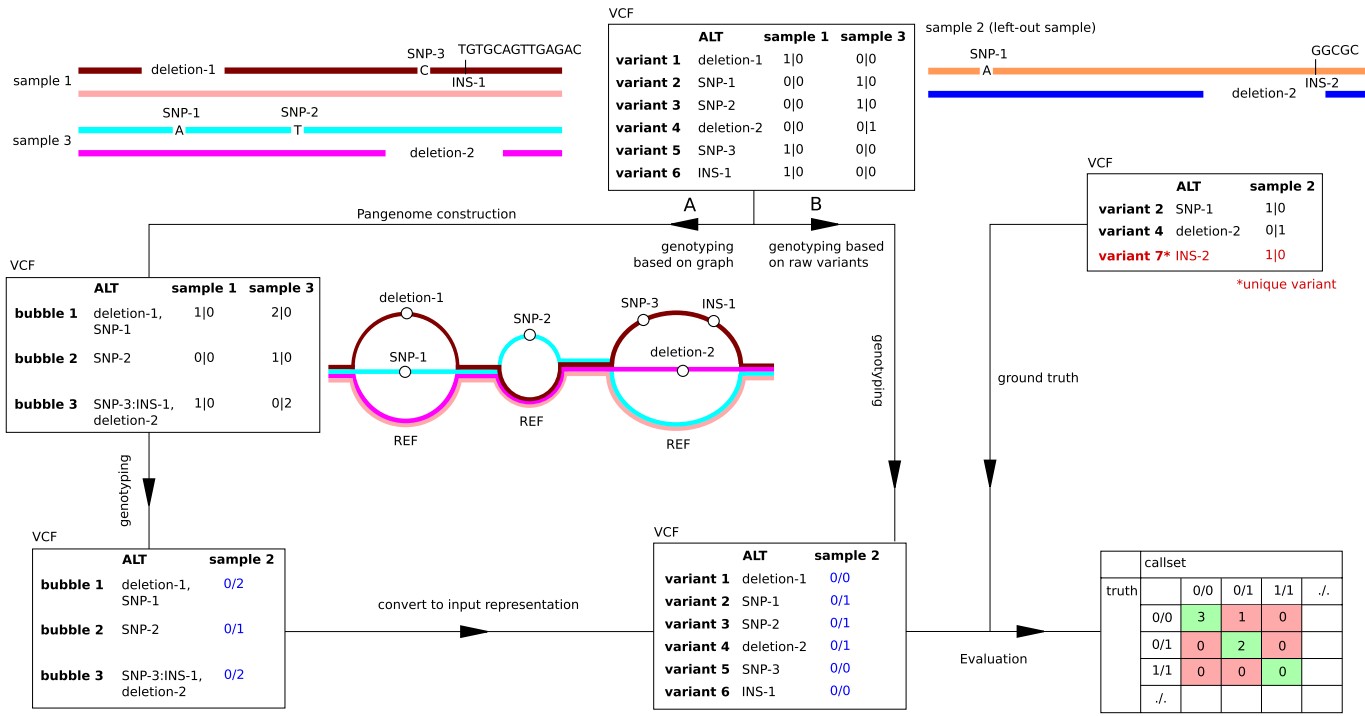

**Extended Data Fig. 2 | Leave one out experiment.** We illustrate the leave-one-out experiment using three samples. Variants are called for all samples based on haplotype-resolved assemblies. For evaluation, we construct a callset containing all variants called in samples 1 and 3, and a truth set containing all variants called in the left out sample (sample 2). The former set of variants is used for genotyping, the latter for evaluation. When running PanGenie, BayesTyper and Platypus, we first convert the variant calls into a pangenome graph representation (stored as VCF) and genotyped the corresponding bubbles (A). We keep track of which bubbles consist of which variant alleles so that genotypes can later be converted back to the original variant representation. For the other tools tested (GATK, Platypus, GraphTyper, Giraffe), we directly used the callset variants as input, without creating the graph (B). The genotypes predicted by each tool are then compared to the variants detected in the left out sample for evaluation. Variants unique to the left out sample cannot be genotyped correctly by any re-genotyping approach (marked in red). We exclude such variants when computing weighted genotype concordances and adjusted precision/recall/Fscore metrics.

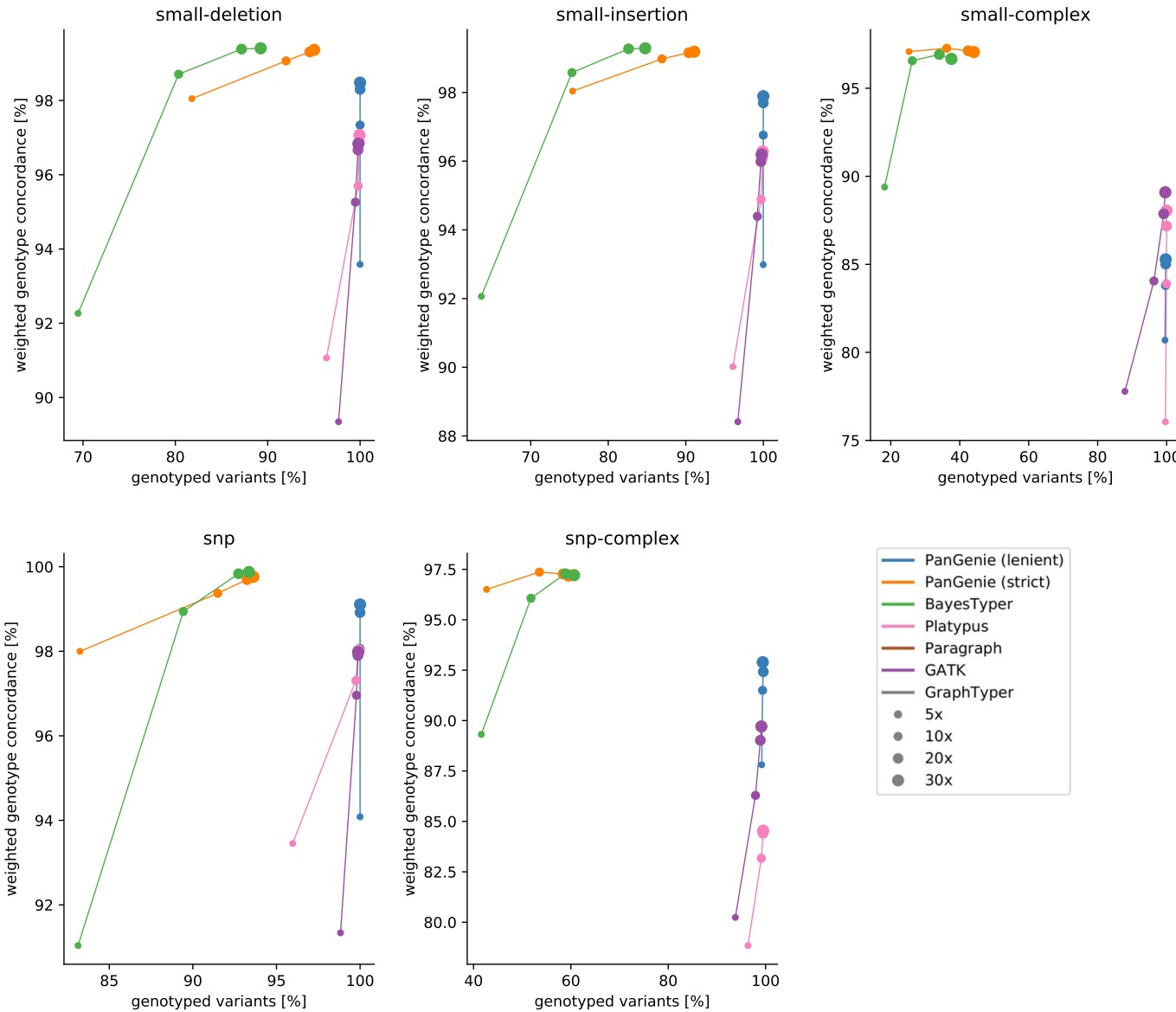

**Extended Data Fig. 3 | Weighted genotype concordance for NA12878 (non-repetitive regions).** Weighted genotype concordance at different coverages for sample NA12878. We ran PanGenie, BayesTyper, Paragraph, Platypus, GATK, GraphTyper and Giraffe in order to re-genotype all callset variants. Besides not applying any filter on the reported genotype qualities ('all'), we additionally report genotyping statistics for PanGenie when using 'high-gq' filtering (genotype quality 200). SNPs, insertions and deletions include all respective variants in biallelic regions of the genome, while *complex* contains all variant alleles falling into regions with complex bubbles in the pangenome graph representation.

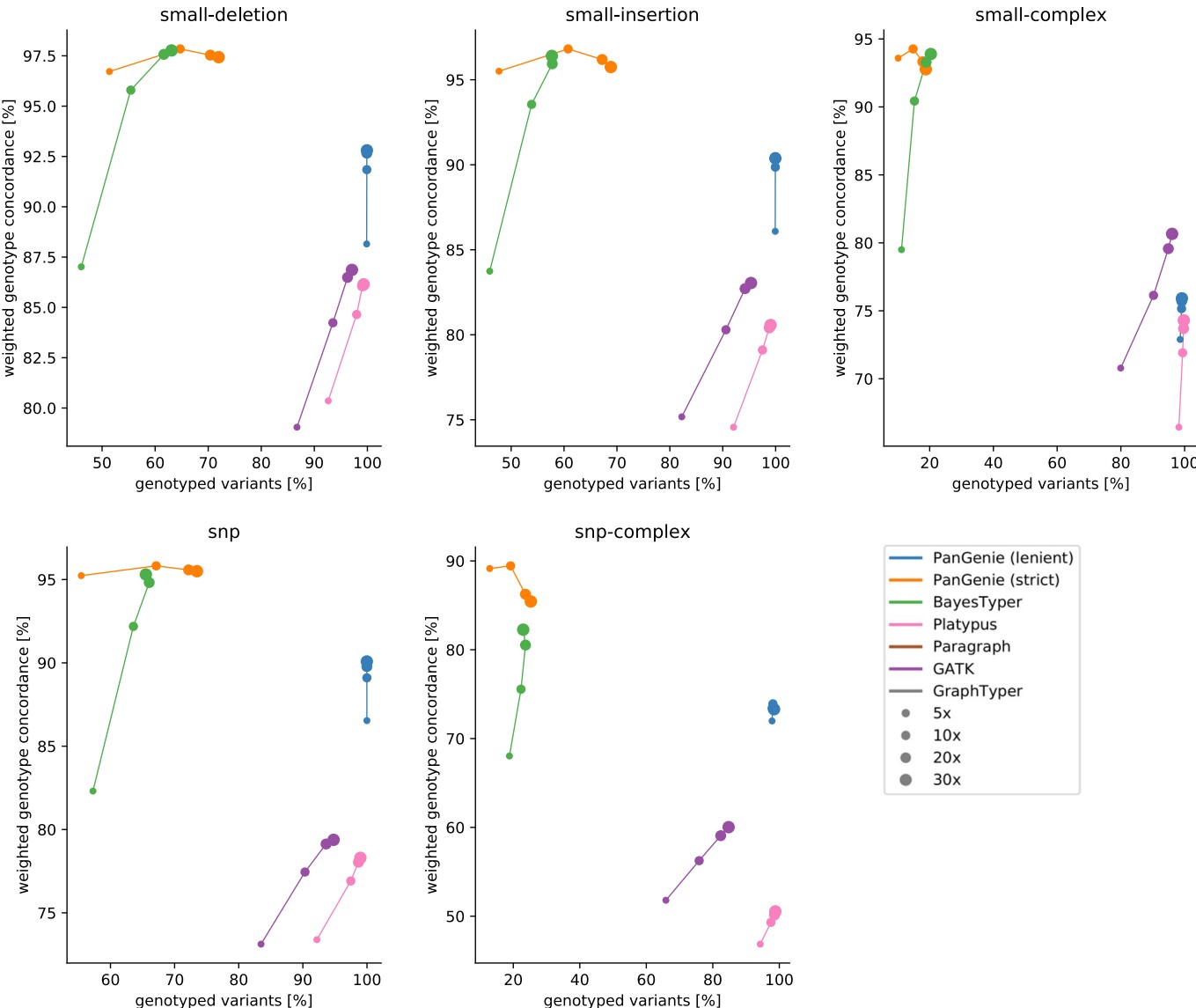

**Extended Data Fig. 4 | Weighted genotype concordance for NA12878 (STR/VNTR regions).** Weighted genotype concordance at different coverages for sample NA12878. We ran PanGenie, BayesTyper, Paragraph, Platypus, GATK, GraphTyper and Giraffe in order to re-genotype all callset variants. Besides not applying any filter on the reported genotype qualities ('all'), we additionally report genotyping statistics for PanGenie when using 'high-gq' filtering (genotype quality 200). SNPs, insertions and deletions include all respective variants in biallelic regions of the genome, while *complex* contains all variant alleles falling into regions with complex bubbles in the pangenome graph representation.

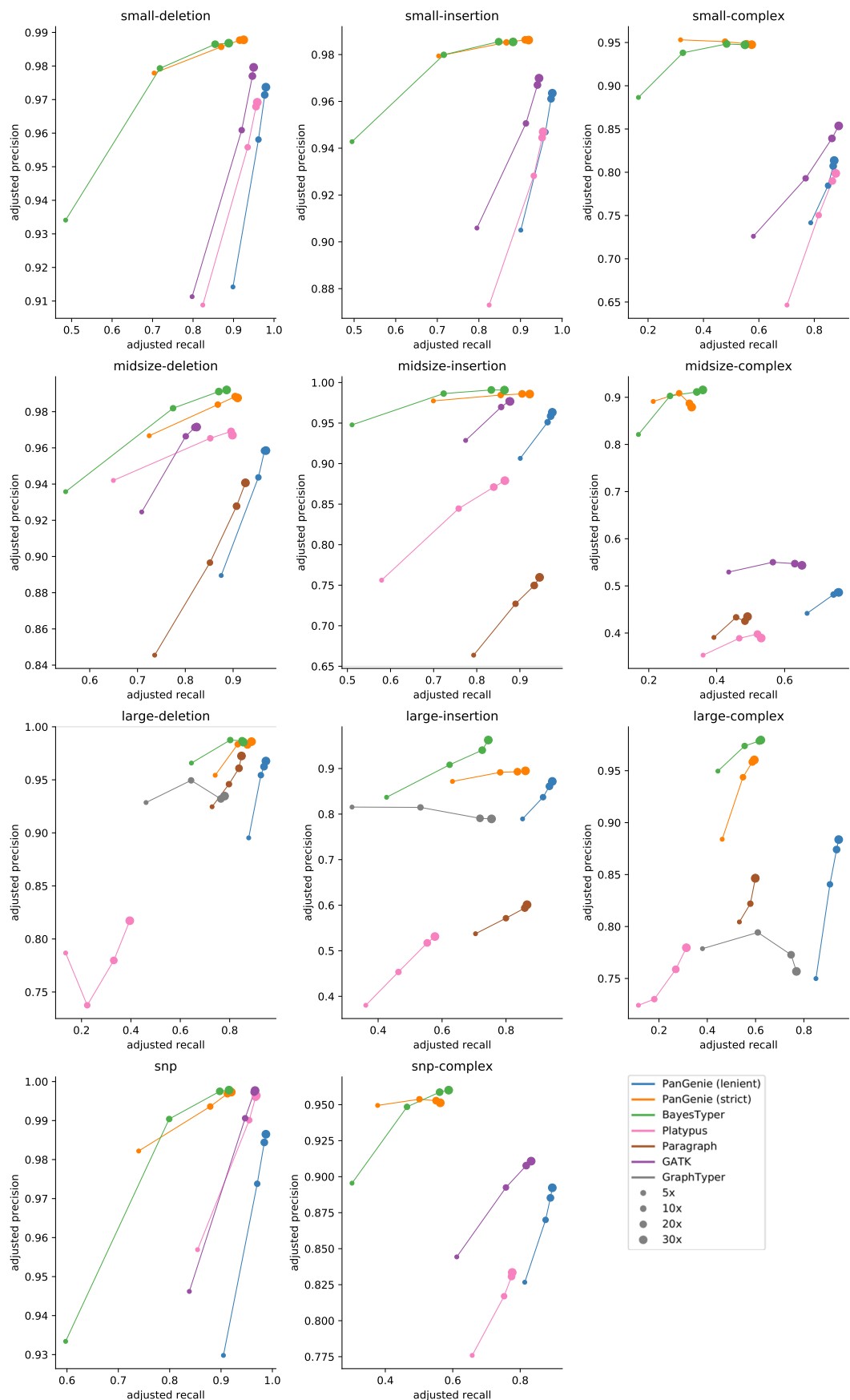

**Extended Data Fig. 5 | See next page for caption.**

**Extended Data Fig. 5 | Adjusted precision/recall for NA12878 (non-repetitive regions).** Adjusted precision/recall at different coverages for sample NA12878. We ran PanGenie, BayesTyper, Paragraph, Platypus, GATK, GraphTyper and Giraffe in order to re-genotype all callset variants. Besides not applying any filter on the reported genotype qualities ('all'), we additionally report genotyping statistics for PanGenie when using 'high-gq' filtering (genotype quality 200). SNPs, insertions and deletions include all respective variants in biallelic regions of the genome, while *complex* contains all variant alleles falling into regions with complex bubbles in the pangenome graph representation.

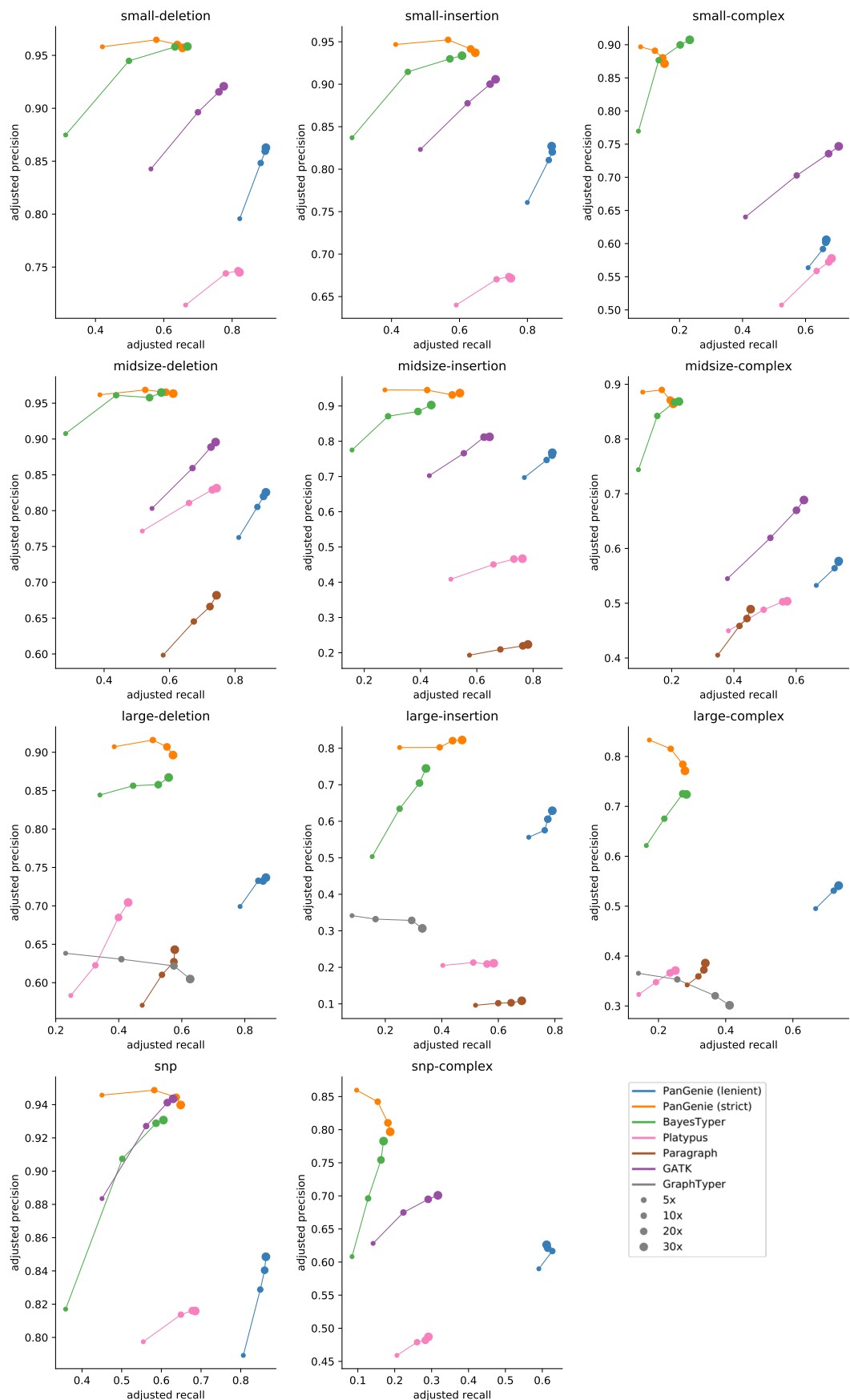

**Extended Data Fig. 6 | See next page for caption.**

**Extended Data Fig. 6 | Adjusted precision/recall for NA12878 (STR/VNTR regions).** Adjusted precision/recall at different coverages for sample NA12878. We ran PanGenie, BayesTyper, Paragraph, Platypus, GATK, GraphTyper and Giraffe in order to re-genotype all callset variants. Besides not applying any filter on the reported genotype qualities ('all'), we additionally report genotyping statistics for PanGenie when using 'high-gq' filtering (genotype quality 200). SNPs, insertions and deletions include all respective variants in biallelic regions of the genome, while *complex* contains all variant alleles falling into regions with complex bubbles in the pangenome graph representation.

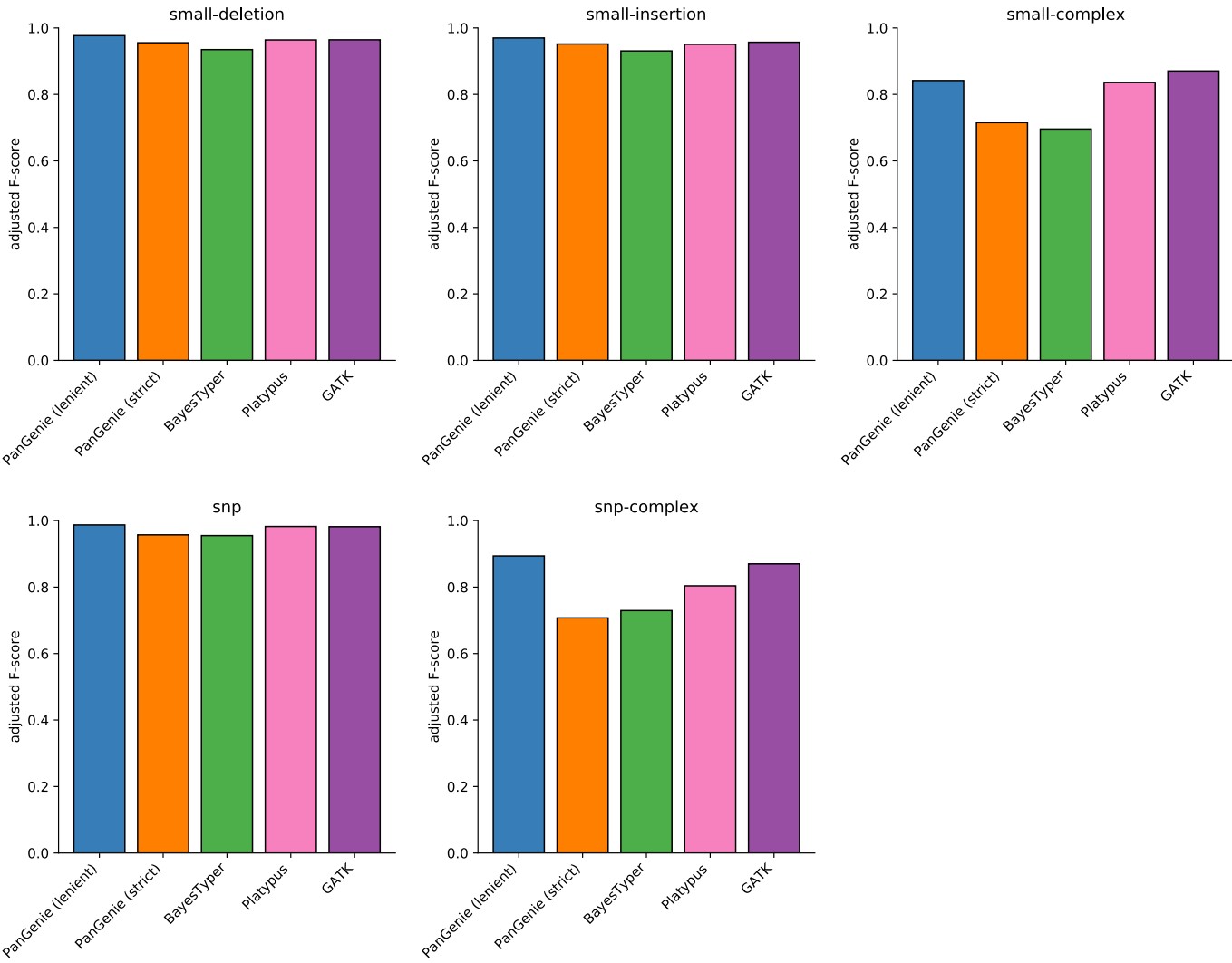

**Extended Data Fig. 7 | Adjusted Fscore for NA12878 (non-repetitive regions).** Adjusted Fscore at coverage 30× for sample NA12878. We ran PanGenie, BayesTyper, Paragraph, Platypus, GATK, GraphTyper and Giraffe in order to re-genotype all callset variants. Besides not applying any filter on the reported genotype qualities ('all'), we additionally report genotyping statistics for PanGenie when using 'high-gq' filtering (genotype quality 200). SNPs, insertions and deletions include all respective variants in biallelic regions of the genome, while *complex* contains all variant alleles falling into regions with complex bubbles in the pangenome graph representation.

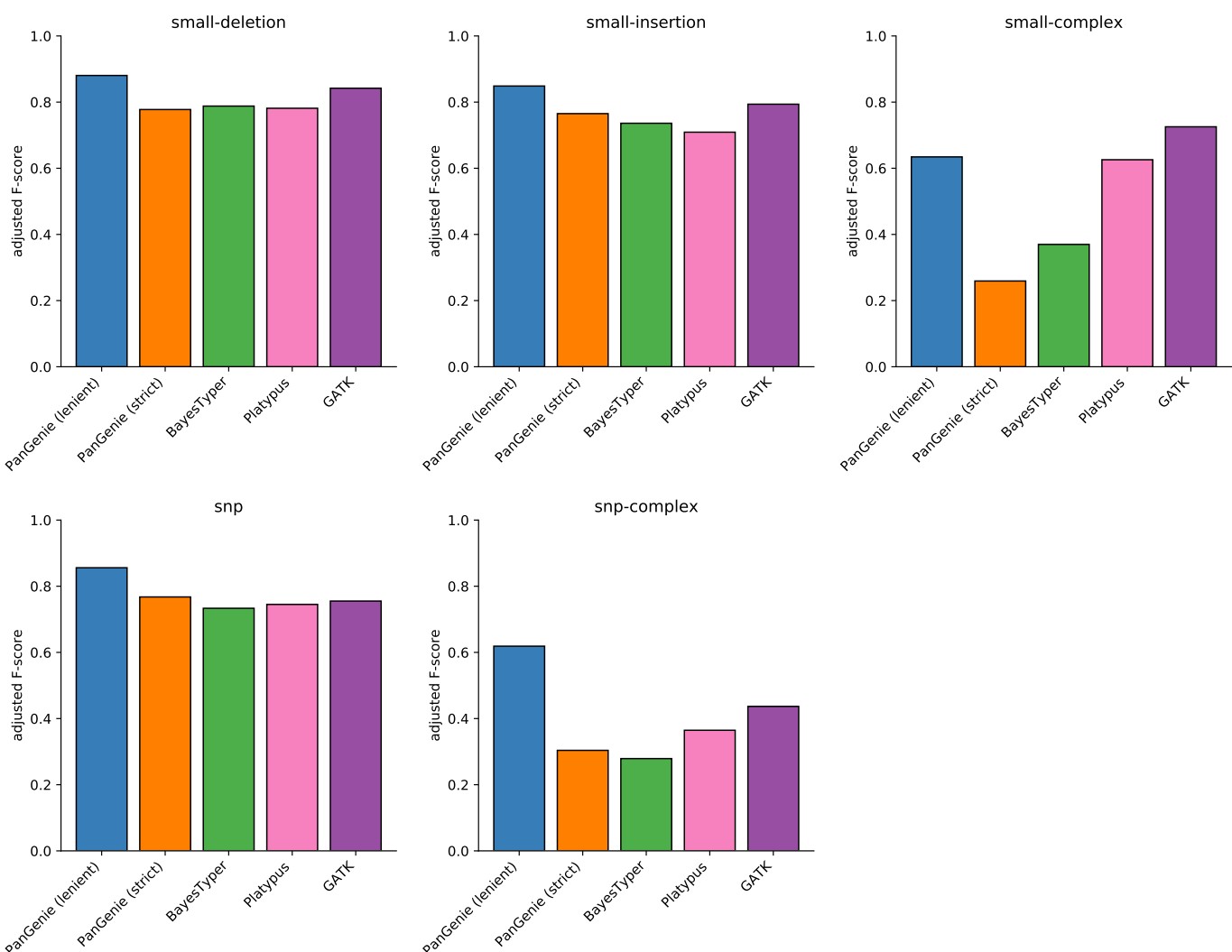

**Extended Data Fig. 8 | Adjusted Fscore for NA12878 (STR/VNTR regions).** Adjusted Fscore at coverage 30× for sample NA12878. We ran PanGenie, BayesTyper, Paragraph, Platypus, GATK, GraphTyper and Giraffe in order to re-genotype all callset variants. Besides not applying any filter on the reported genotype qualities ('all'), we additionally report genotyping statistics for PanGenie when using 'high-gq' filtering (genotype quality 200). SNPs, insertions and deletions include all respective variants in biallelic regions of the genome, while *complex* contains all variant alleles falling into regions with complex bubbles in the pangenome graph representation.

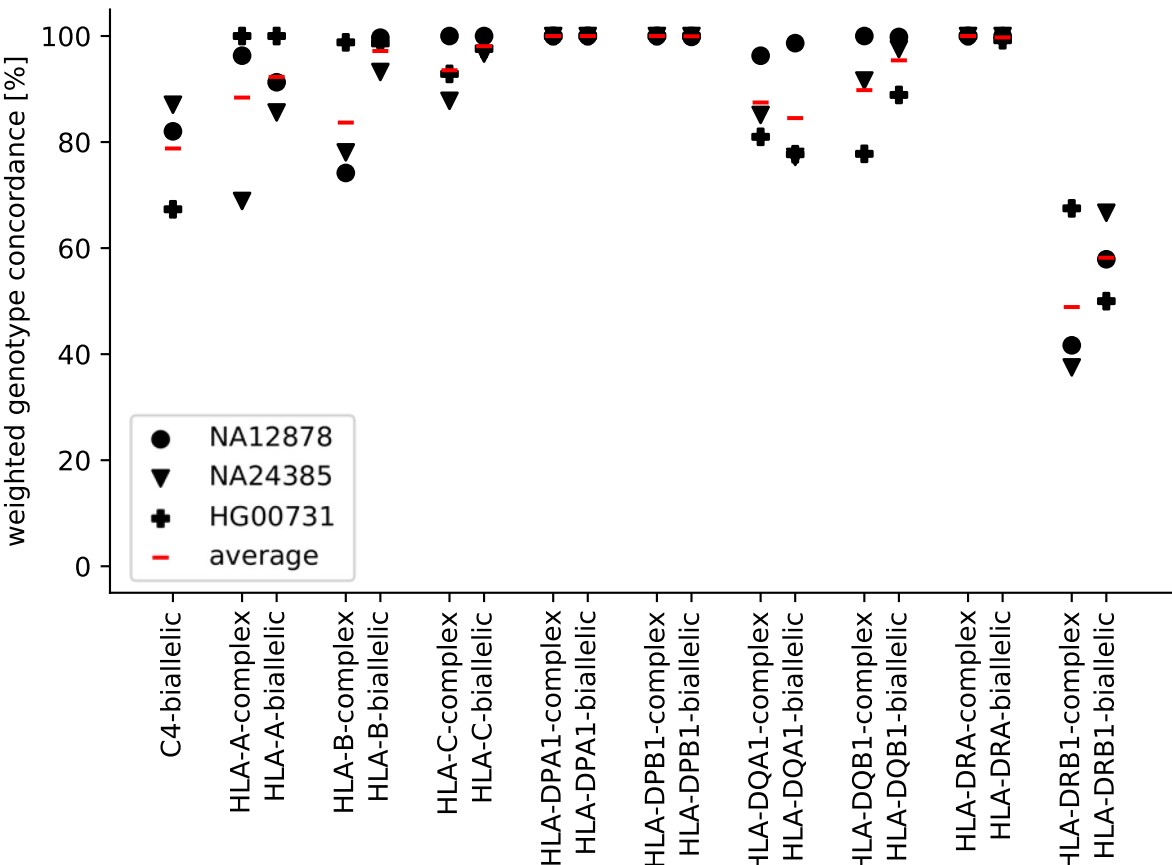

**Extended Data Fig. 9 | HLA genotyping.** Weighted genotype concordances for samples NA12878, NA24385 and HG00731 resulting from a 'leave-one-out' experiment for HLA genes, as well as the average weighted genotype concordance across all three samples (red). For each gene, we separately computed concordances for the simpler, 'biallelic' regions, as well as the more difficult 'complex' regions.

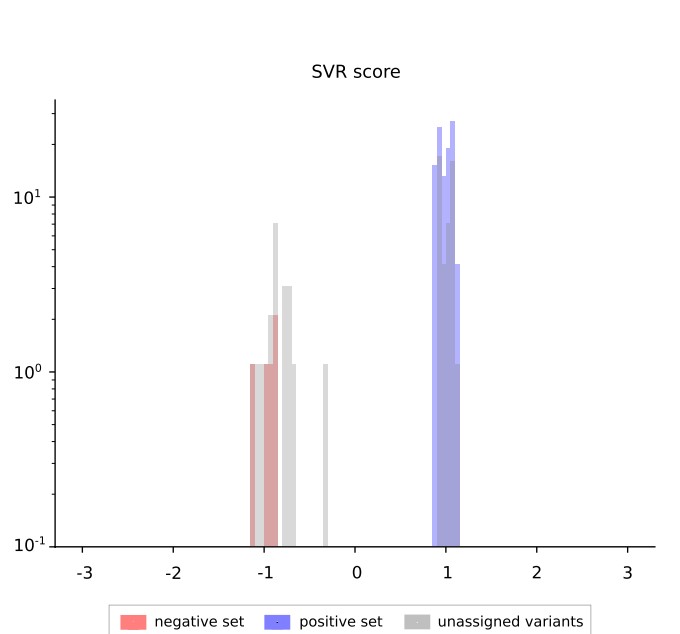
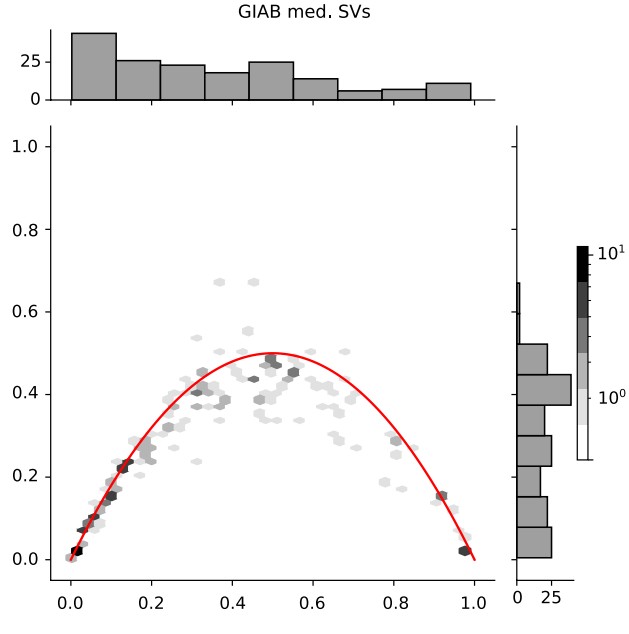

**Extended Data Fig. 10 | GIAB medically relevant SVs in our lenient set.** Distribution of SVR scores for all 209 GIAB medically relevant genes that are part of our variant callset (left), as well as heterozygosities and allele frequencies observed across all 200 unrelated trio samples in our lenient set (right).

# Reporting Summary

Nature Research wishes to improve the reproducibility of the work that we publish. This form provides structure for consistency and transparency in reporting. For further information on Nature Research policies, see our Editorial Policies and the Editorial Policy Checklist.

## Statistics

For all statistical analyses, confirm that the following items are present in the figure legend, table legend, main text, or Methods section.

| n/a | Confirmed | |
|---|---|---|
| ☐ | ☒ | The exact sample size ($n$) for each experimental group/condition, given as a discrete number and unit of measurement |
| ☒ | ☐ | A statement on whether measurements were taken from distinct samples or whether the same sample was measured repeatedly |
| ☒ | ☐ | The statistical test(s) used AND whether they are one- or two-sided *Only common tests should be described solely by name; describe more complex techniques in the Methods section.* |
| ☒ | ☐ | A description of all covariates tested |
| ☒ | ☐ | A description of any assumptions or corrections, such as tests of normality and adjustment for multiple comparisons |
| ☐ | ☒ | A full description of the statistical parameters including central tendency (e.g. means) or other basic estimates (e.g. regression coefficient) AND variation (e.g. standard deviation) or associated estimates of uncertainty (e.g. confidence intervals) |
| ☒ | ☐ | For null hypothesis testing, the test statistic (e.g. $F$, $t$, $r$) with confidence intervals, effect sizes, degrees of freedom and $P$ value noted *Give P values as exact values whenever suitable.* |
| ☒ | ☐ | For Bayesian analysis, information on the choice of priors and Markov chain Monte Carlo settings |
| ☐ | ☒ | For hierarchical and complex designs, identification of the appropriate level for tests and full reporting of outcomes |
| ☒ | ☐ | Estimates of effect sizes (e.g. Cohen's $d$, Pearson's $r$), indicating how they were calculated |

*Our web collection on statistics for biologists contains articles on many of the points above.*

## Software and code

Policy information about availability of computer code

| Data collection | No specific software for data collection was used. |
|---|---|
| Data analysis | Software and workflows developed for this study (PanGenie, Snakemake workflows for evaluation) are available under MIT licence from the respective public repositories (https://github.com/eblerjana/pangenie, (version: commit 1f3d2d2), https://bitbucket.org/jana_ebler/genotyping-experiments/src/master/).  Assemblies were generated using the PGAS pipeline (https://github.com/ptrebert/project-diploid-assembly, parameter settings: v13), hifiasm (v0.15.2) and the SaaRclust package (version #6cb8c96). In our evaluation experiments, we additionally used the following publicly available tools:  minimap2 (https://github.com/lh3/minimap2, version 2.17),  paftools (https://github.com/lh3/minimap2/tree/master/misc),  jellyfish (version 2.2.10), BayesTyper (https://github.com/bioinformatics-centre/BayesTyper, version v1.5),  GATK (version 4.1.3.0),  Paragraph (https://github.com/Illumina/paragraph, version v2a ), Platypus (https://github.com/andyrimmer/Platypus, version 0.8.1), GraphTyper (https://github.com/DecodeGenetics/graphtyper, version 2.7.1), VG/Giraffe (https://github.com/vgteam/vg_snakemake, VG version: v1.30.0), VCFTools (v0.1.16), UCSC  liftOver (https://genome.ucsc.edu/cgi-bin/hgLiftOver), plink (v190b618) and RTG vcfeval (version: v3.9.1). Experiments were run with Snakemake (version: 5.30.1) |

For manuscripts utilizing custom algorithms or software that are central to the research but not yet described in published literature, software must be made available to editors and reviewers. We strongly encourage code deposition in a community repository (e.g. GitHub). See the Nature Research guidelines for submitting code & software for further information.

## Data

Policy information about availability of data

All manuscripts must include a data availability statement. This statement should provide the following information, where applicable:
- Accession codes, unique identifiers, or web links for publicly available datasets
- A list of figures that have associated raw data
- A description of any restrictions on data availability

Illumina short reads for NA24385 were downloaded from: https://ftp-trace.ncbi.nlm.nih.gov/ReferenceSamples/giab/data/AshkenazimTrio/HG002_NA24385_son/NIST_Illumina_2x250bps/reads/. For 1000 Genomes samples, Illumina short reads were downloaded from NCBI SRA (BioProject PRJEB31736). For syndip, reads were downloaded from ftp://ftp.sra.ebi.ac.uk/vol1/fastq/ERR134/006/ERR1341796/. The GIAB small variant benchmark was downloaded from ftp://ftp-trace.ncbi.nlm.nih.gov/giab/ftp/release/NA12878_HG001/NISTv3.3.2/. GIAB medically relevant SVs were obtained from ftp://ftp-trace.ncbi.nlm.nih.gov/ReferenceSamples/giab/data/AshkenazimTrio/analysis/NIST_HG002_medical_genes_SV_benchmark_v0.01/. The syndip benchmark variants were downloaded from https://github.com/lh3/CHM-eval/releases (version 20180222). GnomAD variants were downloaded from https://gnomad.broadinstitute.org/downloads/ (v2). Haplotype-resolved assemblies, variant calls and genotypes produced in this study are available from: 10.5281/zenodo.5607680. For generating haplotype-resolved assemblies, we used sequencing data published in (Ebert et al. 2021).

# Field-specific reporting

Please select the one below that is the best fit for your research. If you are not sure, read the appropriate sections before making your selection.

[x] Life sciences   [ ] Behavioural & social sciences   [ ] Ecological, evolutionary & environmental sciences

For a reference copy of the document with all sections, see nature.com/documents/nr-reporting-summary-flat.pdf

# Life sciences study design

All studies must disclose on these points even when the disclosure is negative.

| | |
|---|---|
| Sample size | Variant calls were made based on all assembly samples (HG00731, HG00732, HG00512, HG00513, NA19238, NA19239, NA12878, HG02818, HG03125, NA24385 and HG03486). HG00733, HG00514 and NA19240 were used for evaluation. |
| Data exclusions | No data was excluded. |
| Replication | Not applicable. No data was generated, all computational analyes can be replicated using the provided pipelines. |
| Randomization | Not applicable. Sample were not assigned to groups. |
| Blinding | Not applicable. All experiments were done computationally and do not involve a human experimenter. |

# Behavioural & social sciences study design

All studies must disclose on these points even when the disclosure is negative.

| | |
|---|---|
| Study description | *Briefly describe the study type including whether data are quantitative, qualitative, or mixed-methods (e.g. qualitative cross-sectional, quantitative experimental, mixed-methods case study).* |
| Research sample | *State the research sample (e.g. Harvard university undergraduates, villagers in rural India) and provide relevant demographic information (e.g. age, sex) and indicate whether the sample is representative. Provide a rationale for the study sample chosen. For studies involving existing datasets, please describe the dataset and source.* |
| Sampling strategy | *Describe the sampling procedure (e.g. random, snowball, stratified, convenience). Describe the statistical methods that were used to predetermine sample size OR if no sample-size calculation was performed, describe how sample sizes were chosen and provide a rationale for why these sample sizes are sufficient. For qualitative data, please indicate whether data saturation was considered, and what criteria were used to decide that no further sampling was needed.* |
| Data collection | *Provide details about the data collection procedure, including the instruments or devices used to record the data (e.g. pen and paper, computer, eye tracker, video or audio equipment) whether anyone was present besides the participant(s) and the researcher, and whether the researcher was blind to experimental condition and/or the study hypothesis during data collection.* |
| Timing | *Indicate the start and stop dates of data collection. If there is a gap between collection periods, state the dates for each sample cohort.* |
| Data exclusions | *If no data were excluded from the analyses, state so OR if data were excluded, provide the exact number of exclusions and the rationale behind them, indicating whether exclusion criteria were pre-established.* |

| Non-participation | *State how many participants dropped out/declined participation and the reason(s) given OR provide response rate OR state that no participants dropped out/declined participation.* |
|---|---|
| Randomization | *If participants were not allocated into experimental groups, state so OR describe how participants were allocated to groups, and if allocation was not random, describe how covariates were controlled.* |

# Ecological, evolutionary & environmental sciences study design

All studies must disclose on these points even when the disclosure is negative.

| Study description | *Briefly describe the study. For quantitative data include treatment factors and interactions, design structure (e.g. factorial, nested, hierarchical), nature and number of experimental units and replicates.* |
|---|---|
| Research sample | *Describe the research sample (e.g. a group of tagged Passer domesticus, all Stenocereus thurberi within Organ Pipe Cactus National Monument), and provide a rationale for the sample choice. When relevant, describe the organism taxa, source, sex, age range and any manipulations. State what population the sample is meant to represent when applicable. For studies involving existing datasets, describe the data and its source.* |
| Sampling strategy | *Note the sampling procedure. Describe the statistical methods that were used to predetermine sample size OR if no sample-size calculation was performed, describe how sample sizes were chosen and provide a rationale for why these sample sizes are sufficient.* |
| Data collection | *Describe the data collection procedure, including who recorded the data and how.* |
| Timing and spatial scale | *Indicate the start and stop dates of data collection, noting the frequency and periodicity of sampling and providing a rationale for these choices. If there is a gap between collection periods, state the dates for each sample cohort. Specify the spatial scale from which the data are taken* |
| Data exclusions | *If no data were excluded from the analyses, state so OR if data were excluded, describe the exclusions and the rationale behind them, indicating whether exclusion criteria were pre-established.* |
| Reproducibility | *Describe the measures taken to verify the reproducibility of experimental findings. For each experiment, note whether any attempts to repeat the experiment failed OR state that all attempts to repeat the experiment were successful.* |
| Randomization | *Describe how samples/organisms/participants were allocated into groups. If allocation was not random, describe how covariates were controlled. If this is not relevant to your study, explain why.* |
| Blinding | *Describe the extent of blinding used during data acquisition and analysis. If blinding was not possible, describe why OR explain why blinding was not relevant to your study.* |

Did the study involve field work? ☐ Yes ☐ No

## Field work, collection and transport

| Field conditions | *Describe the study conditions for field work, providing relevant parameters (e.g. temperature, rainfall).* |
|---|---|
| Location | *State the location of the sampling or experiment, providing relevant parameters (e.g. latitude and longitude, elevation, water depth).* |
| Access & import/export | *Describe the efforts you have made to access habitats and to collect and import/export your samples in a responsible manner and in compliance with local, national and international laws, noting any permits that were obtained (give the name of the issuing authority, the date of issue, and any identifying information).* |
| Disturbance | *Describe any disturbance caused by the study and how it was minimized.* |

# Reporting for specific materials, systems and methods

We require information from authors about some types of materials, experimental systems and methods used in many studies. Here, indicate whether each material, system or method listed is relevant to your study. If you are not sure if a list item applies to your research, read the appropriate section before selecting a response.

## Materials & experimental systems

| n/a | Involved in the study |
|-----|----------------------|
| ☒ ☐ | Antibodies |
| ☒ ☐ | Eukaryotic cell lines |
| ☒ ☐ | Palaeontology and archaeology |
| ☒ ☐ | Animals and other organisms |
| ☒ ☐ | Human research participants |
| ☒ ☐ | Clinical data |
| ☒ ☐ | Dual use research of concern |

## Methods

| n/a | Involved in the study |
|-----|----------------------|
| ☒ ☐ | ChIP-seq |
| ☒ ☐ | Flow cytometry |
| ☒ ☐ | MRI-based neuroimaging |

# Antibodies

| | |
|---|---|
| Antibodies used | Describe all antibodies used in the study; as applicable, provide supplier name, catalog number, clone name, and lot number. |
| Validation | Describe the validation of each primary antibody for the species and application, noting any validation statements on the manufacturer's website, relevant citations, antibody profiles in online databases, or data provided in the manuscript. |

# Eukaryotic cell lines

Policy information about cell lines

| | |
|---|---|
| Cell line source(s) | State the source of each cell line used. |
| Authentication | Describe the authentication procedures for each cell line used OR declare that none of the cell lines used were authenticated. |
| Mycoplasma contamination | Confirm that all cell lines tested negative for mycoplasma contamination OR describe the results of the testing for mycoplasma contamination OR declare that the cell lines were not tested for mycoplasma contamination. |
| Commonly misidentified lines (See ICLAC register) | Name any commonly misidentified cell lines used in the study and provide a rationale for their use. |

# Palaeontology and Archaeology

| | |
|---|---|
| Specimen provenance | Provide provenance information for specimens and describe permits that were obtained for the work (including the name of the issuing authority, the date of issue, and any identifying information). |
| Specimen deposition | Indicate where the specimens have been deposited to permit free access by other researchers. |
| Dating methods | If new dates are provided, describe how they were obtained (e.g. collection, storage, sample pretreatment and measurement), where they were obtained (i.e. lab name), the calibration program and the protocol for quality assurance OR state that no new dates are provided. |

☐ Tick this box to confirm that the raw and calibrated dates are available in the paper or in Supplementary Information.

| | |
|---|---|
| Ethics oversight | Identify the organization(s) that approved or provided guidance on the study protocol, OR state that no ethical approval or guidance was required and explain why not. |

Note that full information on the approval of the study protocol must also be provided in the manuscript.

# Animals and other organisms

Policy information about studies involving animals; ARRIVE guidelines recommended for reporting animal research

| | |
|---|---|
| Laboratory animals | For laboratory animals, report species, strain, sex and age OR state that the study did not involve laboratory animals. |
| Wild animals | Provide details on animals observed in or captured in the field; report species, sex and age where possible. Describe how animals were caught and transported and what happened to captive animals after the study (if killed, explain why and describe method; if released, say where and when) OR state that the study did not involve wild animals. |
| Field-collected samples | For laboratory work with field-collected samples, describe all relevant parameters such as housing, maintenance, temperature, photoperiod and end-of-experiment protocol OR state that the study did not involve samples collected from the field. |
| Ethics oversight | Identify the organization(s) that approved or provided guidance on the study protocol, OR state that no ethical approval or guidance was required and explain why not. |

Note that full information on the approval of the study protocol must also be provided in the manuscript.

# Human research participants

nature research | reporting summary

Policy information about studies involving human research participants

| | |
|---|---|
| Population characteristics | *Describe the covariate-relevant population characteristics of the human research participants (e.g. age, gender, genotypic information, past and current diagnosis and treatment categories). If you filled out the behavioural & social sciences study design questions and have nothing to add here, write "See above."* |
| Recruitment | *Describe how participants were recruited. Outline any potential self-selection bias or other biases that may be present and how these are likely to impact results.* |
| Ethics oversight | *Identify the organization(s) that approved the study protocol.* |

Note that full information on the approval of the study protocol must also be provided in the manuscript.

# Clinical data

Policy information about clinical studies

All manuscripts should comply with the ICMJE guidelines for publication of clinical research and a completed CONSORT checklist must be included with all submissions.

| | |
|---|---|
| Clinical trial registration | *Provide the trial registration number from ClinicalTrials.gov or an equivalent agency.* |
| Study protocol | *Note where the full trial protocol can be accessed OR if not available, explain why.* |
| Data collection | *Describe the settings and locales of data collection, noting the time periods of recruitment and data collection.* |
| Outcomes | *Describe how you pre-defined primary and secondary outcome measures and how you assessed these measures.* |

# Dual use research of concern

Policy information about dual use research of concern

## Hazards

Could the accidental, deliberate or reckless misuse of agents or technologies generated in the work, or the application of information presented in the manuscript, pose a threat to:

| No | Yes | |
|----|-----|---|
| ☐ | ☐ | Public health |
| ☐ | ☐ | National security |
| ☐ | ☐ | Crops and/or livestock |
| ☐ | ☐ | Ecosystems |
| ☐ | ☐ | Any other significant area |

## Experiments of concern

Does the work involve any of these experiments of concern:

| No | Yes | |
|----|-----|---|
| ☐ | ☐ | Demonstrate how to render a vaccine ineffective |
| ☐ | ☐ | Confer resistance to therapeutically useful antibiotics or antiviral agents |
| ☐ | ☐ | Enhance the virulence of a pathogen or render a nonpathogen virulent |
| ☐ | ☐ | Increase transmissibility of a pathogen |
| ☐ | ☐ | Alter the host range of a pathogen |
| ☐ | ☐ | Enable evasion of diagnostic/detection modalities |
| ☐ | ☐ | Enable the weaponization of a biological agent or toxin |
| ☐ | ☐ | Any other potentially harmful combination of experiments and agents |

# ChIP-seq

## Data deposition

☐ Confirm that both raw and final processed data have been deposited in a public database such as GEO.

☐ Confirm that you have deposited or provided access to graph files (e.g. BED files) for the called peaks.

April 2020

| Data access links<br>*May remain private before publication.* | *For "Initial submission" or "Revised version" documents, provide reviewer access links. For your "Final submission" document, provide a link to the deposited data.* |
| Files in database submission | *Provide a list of all files available in the database submission.* |
| Genome browser session<br>(e.g. UCSC) | *Provide a link to an anonymized genome browser session for "Initial submission" and "Revised version" documents only, to enable peer review. Write "no longer applicable" for "Final submission" documents.* |

## Methodology

| Replicates | *Describe the experimental replicates, specifying number, type and replicate agreement.* |
| Sequencing depth | *Describe the sequencing depth for each experiment, providing the total number of reads, uniquely mapped reads, length of reads and whether they were paired- or single-end.* |
| Antibodies | *Describe the antibodies used for the ChIP-seq experiments; as applicable, provide supplier name, catalog number, clone name, and lot number.* |
| Peak calling parameters | *Specify the command line program and parameters used for read mapping and peak calling, including the ChIP, control and index files used.* |
| Data quality | *Describe the methods used to ensure data quality in full detail, including how many peaks are at FDR 5% and above 5-fold enrichment.* |
| Software | *Describe the software used to collect and analyze the ChIP-seq data. For custom code that has been deposited into a community repository, provide accession details.* |

# Flow Cytometry

## Plots

Confirm that:

☐ The axis labels state the marker and fluorochrome used (e.g. CD4-FITC).

☐ The axis scales are clearly visible. Include numbers along axes only for bottom left plot of group (a 'group' is an analysis of identical markers).

☐ All plots are contour plots with outliers or pseudocolor plots.

☐ A numerical value for number of cells or percentage (with statistics) is provided.

## Methodology

| Sample preparation | *Describe the sample preparation, detailing the biological source of the cells and any tissue processing steps used.* |
| Instrument | *Identify the instrument used for data collection, specifying make and model number.* |
| Software | *Describe the software used to collect and analyze the flow cytometry data. For custom code that has been deposited into a community repository, provide accession details.* |
| Cell population abundance | *Describe the abundance of the relevant cell populations within post-sort fractions, providing details on the purity of the samples and how it was determined.* |
| Gating strategy | *Describe the gating strategy used for all relevant experiments, specifying the preliminary FSC/SSC gates of the starting cell population, indicating where boundaries between "positive" and "negative" staining cell populations are defined.* |

☐ Tick this box to confirm that a figure exemplifying the gating strategy is provided in the Supplementary Information.

# Magnetic resonance imaging

## Experimental design

| Design type | *Indicate task or resting state; event-related or block design.* |
| Design specifications | *Specify the number of blocks, trials or experimental units per session and/or subject, and specify the length of each trial or block (if trials are blocked) and interval between trials.* |
| Behavioral performance measures | *State number and/or type of variables recorded (e.g. correct button press, response time) and what statistics were used to establish that the subjects were performing the task as expected (e.g. mean, range, and/or standard deviation across subjects).* |

## Acquisition

**Imaging type(s)**
*Specify: functional, structural, diffusion, perfusion.*

**Field strength**
*Specify in Tesla*

**Sequence & imaging parameters**
*Specify the pulse sequence type (gradient echo, spin echo, etc.), imaging type (EPI, spiral, etc.), field of view, matrix size, slice thickness, orientation and TE/TR/flip angle.*

**Area of acquisition**
*State whether a whole brain scan was used OR define the area of acquisition, describing how the region was determined.*

**Diffusion MRI**  ☐ Used  ☐ Not used

## Preprocessing

**Preprocessing software**
*Provide detail on software version and revision number and on specific parameters (model/functions, brain extraction, segmentation, smoothing kernel size, etc.).*

**Normalization**
*If data were normalized/standardized, describe the approach(es): specify linear or non-linear and define image types used for transformation OR indicate that data were not normalized and explain rationale for lack of normalization.*

**Normalization template**
*Describe the template used for normalization/transformation, specifying subject space or group standardized space (e.g. original Talairach, MNI305, ICBM152) OR indicate that the data were not normalized.*

**Noise and artifact removal**
*Describe your procedure(s) for artifact and structured noise removal, specifying motion parameters, tissue signals and physiological signals (heart rate, respiration).*

**Volume censoring**
*Define your software and/or method and criteria for volume censoring, and state the extent of such censoring.*

## Statistical modeling & inference

**Model type and settings**
*Specify type (mass univariate, multivariate, RSA, predictive, etc.) and describe essential details of the model at the first and second levels (e.g. fixed, random or mixed effects; drift or auto-correlation).*

**Effect(s) tested**
*Define precise effect in terms of the task or stimulus conditions instead of psychological concepts and indicate whether ANOVA or factorial designs were used.*

**Specify type of analysis:**  ☐ Whole brain  ☐ ROI-based  ☐ Both

**Statistic type for inference**
(See Eklund et al. 2016)
*Specify voxel-wise or cluster-wise and report all relevant parameters for cluster-wise methods.*

**Correction**
*Describe the type of correction and how it is obtained for multiple comparisons (e.g. FWE, FDR, permutation or Monte Carlo).*

## Models & analysis

n/a | Involved in the study
☐ | ☐ Functional and/or effective connectivity
☐ | ☐ Graph analysis
☐ | ☐ Multivariate modeling or predictive analysis

**Functional and/or effective connectivity**
*Report the measures of dependence used and the model details (e.g. Pearson correlation, partial correlation, mutual information).*

**Graph analysis**
*Report the dependent variable and connectivity measure, specifying weighted graph or binarized graph, subject- or group-level, and the global and/or node summaries used (e.g. clustering coefficient, efficiency, etc.).*

**Multivariate modeling and predictive analysis**
*Specify independent variables, features extraction and dimension reduction, model, training and evaluation metrics.*

