## [Peer Review File · Nature Genetics]

Peer Review Information

Manuscript Title: Pangenome-based genome inference allows efficient and accurate genotyping across a wide spectrum of variant classes

Corresponding author name(s): Dr. Tobias Marschall

Reviewer Comments & Decisions:

Decision Letter, initial version:

27th Apr 2021

Dear Tobias,

Your Technical Report, "Pangenome-based Genome Inference" has now been seen by 4 referees. You will see from their comments copied below that while they find your work of considerable potential interest, they have raised quite substantial concerns that must be addressed. In light of these comments, we cannot accept the manuscript for publication, but would be very interested in considering a revised version that addresses these serious concerns.

Briefly, all four referees appreciate the technical soundness and interest of PanGenie. However, each also raises important issues that we think should be addressed before we make a decision.

Reviewer #1 provides a thoughtful review; they comment several times issues with the benchmarking (versus other tools and SV calls compared to gnomAD) and provide clear suggestions for improvement.

Reviewer #2 makes a few specific suggestions, specifically on PanGenie's performance based on input data; and how different haplotype graphs may affect interpretation and comparison of analyses.

Reviewer #3 sounds quite positive; both of their specific comments regard PanGenie's performance with respect to complexity, in terms of genomic regions and in the haplotype graph.

Reviewer #4 provides a straightforward report; they also think that PanGenie is interesting but desires much more detail on how it works.

We thought that Reviewer #1 and #3's comments about the benchmarking and complexity were important and should be fully addressed in a revision. We thought that other comments - specifically Reviewer #1's suggestion for a larger-scale analysis on the 1000G data, or looking at specific complex loci; and Reviewer #2 regarding non-human genomes - were also interesting and, if acted upon, would strengthen your study.

We hope you will find the referees' comments useful as you decide how to proceed. If you wish to submit a substantially revised manuscript, please bear in mind that we will be reluctant to approach the referees again in the absence of major revisions.

To guide the scope of the revisions, the editors discuss the referee reports in detail within the team, including with the chief editor, with a view to identifying key priorities that should be addressed in revision and sometimes overruling referee requests that are deemed beyond the scope of the current study. We hope that you will find the prioritised set of referee points to be useful when revising your study. Please do not hesitate to get in touch if you would like to discuss these issues further.

If you choose to revise your manuscript taking into account all reviewer and editor comments, please highlight all changes in the manuscript text file. At this stage we will need you to upload a copy of the manuscript in MS Word .docx or similar editable format.

*2) If you have not done so already please begin to revise your manuscript so that it conforms to our Technical Report format instructions, available here. Refer also to any guidelines provided in this letter.

*3) Include a revised version of any required Reporting Summary: <https://www.nature.com/documents/nr-reporting-summary.pdf>
It will be available to referees (and, potentially, statisticians) to aid in their evaluation if the manuscript

goes back for peer review.

[REDACTED]

If you wish to submit a suitably revised manuscript we would hope to receive it within 6 months. If you cannot send it within this time, please let us know. We will be happy to consider your revision so long as nothing similar has been accepted for publication at Nature Genetics or published elsewhere. Should your manuscript be substantially delayed without notifying us in advance and your article is eventually published, the received date would be that of the revised, not the original, version.

Thank you for the opportunity to review your work.

Sincerely,

Michael Fletcher, PhD
Associate Editor, Nature Genetics

ORCID: 0000-0003-1589-7087

Referee expertise: all referees have experience in human genetics and/or computational genetics/bioinformatics.

Reviewers' Comments:

Reviewer #1:

Remarks to the Author:

The paper proposes a new method (PanGenie) to genotype variation using pangenome graphs. The PanGenie implementation is publicly available with an open source license. In the paper, the authors describe how they constructed a graph containing variation found in five publicly available haplotype resolved (phased) human assemblies. Each haplotype is aligned to the reference genome to generate a phased set of input variants. Like in several other previous graph-based methods, the purpose of the graph is to make use of prior variant information to improve genotyping accuracy. PanGenie is also a mapping-free method, variation in the graph is genotyping using only a hash table of k-mer counts which results in a very fast genotyping compared to mapping-based methods. PanGenie does not discover new variants, however, the authors mention that this might be a possibility in the future.

PanGenie supports a wide variety of variants, including SNVs, indels, tandem repeats, structural variants. Therefore, the authors compare to both methods that genotype small variants (SNVs and indels) and those that genotyping larger variants (SVs), but also two (BayesTyper and Platypus) that can genotype all of the same variant types as PanGenie. In one experiment, they remove one of the five individuals and use a graph from the other four to genotype the one left out individual. In this experiment, BayesTyper seems to produce the most accurate genotype calls, however, in the STR/VNTR regions PanGenie outperforms the other tools. The authors highlight their methods performance in these repeat regions. Based on the description of the method, BayesTyper and PanGenie seem to have many similarities, as both tools genotype using pangenome graphs based on k-mer counts so it is no surprise their results are similar (pangenie "strict" set). The most distinct feature between these two methods is that PanGenie incorporates the haplotype information and uses that to infer genotypes.

I thought all of the mapping-based methods are doing surprisingly poor in the benchmarks, as these metrics are not much in line with previous publications. For example, GATK and Platypus, have been repeatedly shown to have at least 99.5+% and 98+% sensitivity and precision with high coverage data (30x+) in non-repeat regions for SNPs and indels, respectively. Both methods are mostly doing poorly in the benchmark of the paper and it is not clear why. My guess is that either the benchmark truth set has many incorrect calls, or that these methods perform suboptimally when "force calling" a set of variants

and would actually perform better if they are allowed to discover their own set of variants. In case of the latter, the authors should also run these tools in their recommended way (with variant discovery). I realize this would mean that the dataset no longer contains the exact same variants, but it should not be an issue if you use published benchmark tools such as “rtg vcfval” and Illumina’s hap.py for comparisons. These tools are designed to handle well datasets that have different variant representations. I think the authors should also add a comparison to an external public truth set using one of those benchmark tools. The truth set could, for example, either be Genome in a Bottle (GIAB, <https://www.nist.gov/programs-projects/genome-bottle>) or syndip (<https://www.nature.com/articles/s41592-018-0054-7/>). These truth set are an extremely useful resource for validating genotyping methods with real data. Genome in a Bottle already has a truth set for NA12878/HG001, which are already part of this study. The syndip truth set VCF is created using the same tools as the authors use for creating the input VCF, so it should also include the STR/VNTR regions that the manuscript highlights.

In the PanGenie graph, the haplotype information is stored and is used to infer genotype calls. PanGenie has the fewest missing genotype calls out of all the benchmarked methods, and my understanding is that it is because they use the haplotype information for inferring a possible genotype call, even when there are no unique k-mers to tag variant alleles. While it is not novel to infer genotypes based on haplotypes in population genetics, I have not seen this used as part of a genotyping method before. A few methods use the allele frequency as a prior but I personally like the authors’ method much better, however, I think the reference panel here is way too small to be effective. Reference panels usually require several hundreds and often thousands of haplotypes for an accurate imputation of the genotype calls. I suspect that many of the genotype calls that have no unique k-mers that inferred genotype calls are widely inaccurate since the call will be determined only from a handful of known haplotypes. The authors seem to be aware of the lack of haplotypes, because they add six haplotype paths into their graph by typing them with PanGenie. However, this does not add any new haplotype path for variants that have no unique k-mers, since they are solely inferred using the other five haplotypes. This also makes the “leave-one-out” experiment not very realistic. Instead of excluding an individual from the extended panel, it would be more realistic to leave out the individual before the other extended panel was created, because his/her haplotype has been used for inferring other calls in the dataset.

As a side note: I believe that genotyping with genome inference would shine in genotyping studies in which there are thousands of publicly available haplotypes. For example, the phased set of 1000 genomes variants, or genotyping known haplotypes of HLA, KIR, immunoglobulin, and other difficult-to-call genes that have hundreds or thousands of known haplotypes publicly available. Of course, I do not insist that the authors would add any such analysis to the manuscript.

The paper also describes genotyping of random 100 individuals from the 1000 Genomes project to show that the methods can scale to larger cohorts. The structural variants of this set is compared against

public gnomad SVs, however, I found the results very surprising because the overlap between common variants ($AF \geq 5\%$) in gnomad and PanGenie is very low (around 23% of PanGenie variants, I did not find in the manuscript the fraction of gnomad SVs which were not found with PanGenie but I would think based on Figure 4 that it is also quite low). The two datasets also have very different allele frequency spectrum (Figure 4a and 4c). The authors interpret the results by suggesting that gnomad might be missing up to 77% of common SVs in their study. However, in my experience the gnomad set is of very high quality, particularly for common SVs so I do not find this very convincing. I believe a much more likely explanation is that PanGenie is overestimating SV frequencies due to the low number of haplotypes used for genome inference. The only quality metric used here is testing for HWE abnormalities but those tests would not detect such issues. These results emphasize even further the need to benchmark against external truth sets that the authors could use to validate their SV genotyping (GIAB SV or syndip, for example).

In summary, I found the inclusion of haplotype information an interesting approach and I believe it could be promising for many applications where a large number of haplotypes is known. However, from the above observations, I have many concerns regarding the validation of the genotyping accuracy in the authors' experiments. The authors should more properly benchmark their method by using published tools and compare against external truth sets to validate the accuracy of their method. Additionally, comparing the datasets to external datasets created with orthogonal methods would increase the confidence in the accuracy even further.

Other minor comments:

- The manuscript compares against k-mer based pangenome method but none of the mapping-based pangenome methods, which are referenced in the introduction (vg, graph typer, GRAF). I would suggest comparing to at least one of those methods.
- Figure 4a labels both plots with "pangenie". While reading, I assumed the lower plot was for gnomad SV since the allele frequencies reach lower numbers in that plot.
- When discussing runtimes, I only found the total runtimes shown in Supplementary Table S4. It would be good to add some further stratification of it, particularly the time for running BWA and the k-mer counting. Since if the input variants were updated, you could subtract those runtimes for estimating the time for re-genotyping.
- In addition to runtimes, memory requirements should be added.

I thank the authors for the good work on the topic. I found the paper to be well written and easy to follow. I hope you find my comments to be constructive.

Reviewer #2:

Remarks to the Author:

In their publication, Ebler et al. present a novel algorithm PanGenie for genotyping of populations based on short reads.

PanGenie builds a pangenome graph with ~two haplotypes from phased SNPs, and then takes short reads from many individuals, counts k-mers in these reads, and looks up k-mers in the haplotype graph to genotype individuals fast. PanGenie outperforms regular short-read based genotyping pipelines because k-mer counting and lookup is faster than short-read alignment. Faster genotyping is needed now as human sequencing projects sequence millions of individuals, with read-alignment being one of the bottlenecks.

The authors show that PanGenie is more accurate in genotyping alleles in repetitive regions, which I feel is a bit like an apples-to-oranges comparison. PanGenie's input is a haplotype graph built by aligning chromosome-scale assemblies with the human reference (in the paper called 'contigs') using minimap2 via the authors' vcf-merging pipeline. The input for the other pipelines (BayesTyper etc.) are BWA-made soft-trimmed short-read alignments. It is no surprise then that repetitive regions look better in PanGenie, as large chromosome-scale alignments easily bridge repeats while soft-trimmed short-read alignments struggle. This should at least be discussed. I wonder how it would look like if the input to PanGenie were BWA-made short-read-based SNPs, it would probably look much worse. The quality of the input SNPs obviously has a huge impact on PanGenie's performance.

The tool is presented as a generic tool when in fact it has only been assessed on human genomes and is unlikely to work with some other species. Plants for example have very different genomes to humans, often with polyploidy and low heterozygosity. This limitation should be clarified in the text.

While some of the shortcomings of the method are discussed, I see a major advantage of read mapping is the ability to detect previously unseen variants, inversions and translocations. I wonder if these limitations will limit the future implementation of this method over read mapping. While the approach is interesting, it isn't totally novel. Shajii et al., 2016 and Sun et al. 2018 should be cited because at its base, PanGenie uses the same idea – genotyping known SNPs is faster when you look up known k-mers instead of wasting time aligning reads (<https://pubmed.ncbi.nlm.nih.gov/27587672/> and <https://academic.oup.com/bioinformatics/article/35/3/415/5056043>). A major question is, given the limitations, would this method become widely adopted, and if so, how would any bias from using different haplotype graphs be accounted for when comparing studies?

What would improve this manuscript:

- Table S4 contains the runtimes for all compared software. I'd like to have added on what kind of

machine these steps were run – number of CPUs, memory etc. I think for completeness' and fairness sake, this table should also include the runtime of the vcf-merging pipeline. After all, that step does most of the heavy lifting, PanGenie itself 'just' checks k-mers for the called variants.

- The last line in the supplementary notes before the References part breaks off early – 'tools can be f'
- Line 343, in the definition of genotype concordance, how is 'wrong predictions' defined here? Is it false positives, so a genotype is called where one shouldn't be called, or is it false negatives, no genotype is called where one should be called, or is it both together, or something else entirely, i.e., genotype B is called where A should be called? Listing in a sentence which one it is probably helpful. If it is false negatives then 'genotype concordance' is the same as 'recall'.
- Line 218, 'we separately mapped contigs of each haplotype', it should say the average or N50 size of these contigs here, they're probably huge

Reviewer #3:

Remarks to the Author:

The manuscript from authors Jana Ebler and colleagues reports a graph-based method that combines pan-genome representations from haplotype-resolved assemblies with k-mer based genotyping.

The method is well designed and algorithmically sound. The utilization of a Hidden Markov Model to link reconstructed genotypes is a great application of the technology in this context. Laudably, the authors carry out detailed and careful comparisons to alternative genotyping methods, and demonstrate that their graph-based method excels at genotyping larger variants (which is exactly what is expected from this method). They also convincingly show that the method scales well for population studies (i.e. with the number of genomes to be genotyped).

I have two related major concerns:

Concern 1: The authors restricted the construction of the pan-genome graph to 87.5% of the genome, where they were able to identify genomic variation across the included 10 haplotypes from the 5 reference individuals. This likely excludes the majority of more complex regions, where the lion's share of the types of longer and more complex variation resides for which genotyping improvements are the most sorely needed. This may severely restrict the usefulness of the method.

Concern 2: It is unclear how the method will scale with the number of reference haplotypes included in the pan-genome reference graph. With an increasing number of haplotypes, the number of regions where imperfect copies of medium-sized and larger variants overlap in complex ways also increases. It is unclear how PanGenie's graph construction method that is conceptually clean for "well-behaved" and spatially distinct bi-allelic variations will perform in such scenarios. Will many of these areas be excluded

(see concern 1 above)?

Overall, a really nice method, but I fear that method will not be able to achieve its full potential impact for genotype improvement without strengthening extending the pan-genome graph construction method to include more than the "best-behaved" parts of the human genome.

Reviewer #4:

Remarks to the Author:

The paper describes a new method for genotype calling from sequence data, that uses haplotype-resolved sequences to build a reference graph. Reads are then compared to the graph and an HMM is used to estimate a likely pair of sequences. The method seems novel, and the comparison to other methods is convincing. However the method itself is poorly described. The paper does not make it clear how variants are called. Especially variants that are not represented in the pangenome graph.

I think the method has 4 main steps?

1. The pangenome graph is constructed. I think this involves only unique kmers to each region?
2. kmers in the graph are counted in the reads of a new sample.
3. The HMM resolves a pair of paths through the graph.
4. How then are variants that are not represented in graph called? The methods section doesn't mention the word deletion once. Tell us how they are called in the Methods.

I think you need to make this a lot clearer. Setting out a series of steps, and having figures that better describe examples would help.

In figure 1 : The dashed red and blue lines - do these colors correspond to the solid colored lines (grey, green, blue and orange) in the left hand panels? It seems to me that the red dashed line might correspond to the solid green line?

I think the figure could be better arranged/annotated to make it clearer that the reads (top left box) are from the sample to be genotyped.

It's not clear where you get the list of 3 possible alleles from in the top right panel. Also the kmer CAGG in the middle of the blue dashed line - does the correspond to the middle kmer (orange circle) on the blue and orange lines of the first bubble in the bottom left panel. It could be clearer how you go about calling a deletion in this example. Maybe you could have an example without a deletion as the main figure, then a supplementary figure where you illustrate what happens when there is a deletion.

Using orange for circles and lines in the bottom left panel isn't ideal. Could you annotate in the figure "bubble 1" , "bubble 2" etc

In the caption you say "The second bubble is poorly covered by k-mers, however, linkage to adjacent variants can be used to infer the two local haplotype paths." In the text (lines 53-54) you say "The second variant is poorly covered by k-mers but the count distributions along the alleles of the first variant indicate that the unknown genome is composed of the green and blue haplotypes." I think you mean second bubble rather than second variant?

On lines 70-74 you describe "our callsets" and it would be better if you could define that more clearly.

Does the method work equally well on WGS and Exome sequencing?

Author Rebuttal to Initial comments

Editor

Comment E.1: Your Technical Report, "Pangenome-based Genome Inference" has now been seen by 4 referees. You will see from their comments copied below that while they find your work of considerable potential interest, they have raised quite substantial concerns that must be addressed. In light of these comments, we cannot accept the manuscript for publication, but would be very interested in considering a revised version that addresses these serious concerns.

Briefly, all four referees appreciate the technical soundness and interest of PanGenie. However, each also raises important issues that we think should be addressed before we make a decision.

Response: We are grateful for this positive assessment and have addressed all points raised.

Comment E.2: Reviewer #1 provides a thoughtful review; they comment several times issues with the benchmarking (versus other tools and SV calls compared to gnomAD) and provide clear suggestions for improvement.

Response: We have substantially extended and re-run all experiments, comprehensively addressing all these concerns. In summary, we made the following changes to the evaluation:

- In the interim, more data has become publicly available, in particular, in the frame of the Human Genome Structural Variation Consortium (HGSCV), we published more long read data sets (Ebert et al., Science, 2021) and a preprint on the 1000 Genomes high coverage data has appeared (Byrska-Bishop, bioRxiv, 2021). These new data are utilized as follows.

- It has become apparent that genome assemblies produced by HifiAsm are superior to Peregrine, which was underlying our previous assemblies and was used by Ebert et al. For this revision, we therefore selected the PacBio Hifi data sets from Ebert et al., updated the assembly pipeline to use HifiAsm instead of Peregrine, and generated 28 new whole-genome haplotype assemblies from 14 diploid samples (11 unrelated samples and 3 children). The new assemblies show a mean N50 value of 25.42 Mbp (previously 20.97 Mbp for Peregrine).
- We called variants for these new phased assemblies, creating a reference panel of 22 unrelated haplotypes (more than doubling the number compared to the 10 haplotypes in our original submission).
- We changed the design of our “leave-out-one” experiment. Instead of removing a sample after the graph was generated, we now build a separate graph based only on variants called across $n-1$ samples, so that the graph was constructed without any information on the left out sample. Variant calls for the left out sample are then made completely separately and used as ground truth for evaluation.
- As suggested, we added more tools for comparison (GraphTyper and VG Giraffe) and additionally, we ran GATK and Platypus in “discovery mode”.

- We welcome the suggestion to better relate our results to previously published benchmarks and metrics. We now offer more different views on our results in terms of different metrics, and particularly used “rtg vcfeval” to compute precision/recall as suggested by Reviewer 1. Additionally, we introduce two new metrics: 1) “weighted genotype concordance”, that weighs concordances for all genotypes equally and hence provides a more balanced view when a majority of variants are absent in a benchmark sample (i.e. genotype 0/0) and 2) “adjusted precision/recall/F-score”, that correct for the fact that some variants cannot be detected by genotyping, simply because they are not part of the input set of variants to be genotyped. We focus the main display items on weighted genotype concordance and adjusted F-score, and provide alternative metrics as supplementary figures. The new results confirm PanGenie’s excellent performance.
- We now include detailed benchmarking results with respect to Genome-in-a-Bottle GIAB small variant ground truth sets. We consider the new GIAB set of challenging medically relevant SVs and establish that 174 of 250 (70%) of these SVs can be reliably genotyped with PanGenie.
- For the initial submission, we had run PanGenie on short-read data of 100 unrelated individuals. Now we extended this analysis by running it on 300 samples from 100 mother-father-child trios. This allows us to assess, besides Hardy-Weinberg equilibrium, also Mendelian consistency. Based on these and other metrics, we determine subsets of variants that can be genotyped with outstanding accuracy (strict) or still at acceptable reliability (lenient).

Comment E.3: Reviewer #2 makes a few specific suggestions, specifically on PanGenie’s performance based on input data; and how different haplotype graphs may affect interpretation and comparison of analyses.

Response: We responded in detail to all points raised by Reviewer 2.

Comment E.4: Reviewer #3 sounds quite positive; both of their specific comments regard PanGenie’s performance with respect to complexity, in terms of genomic regions and in the haplotype graph.

Response: We thank Reviewer 3 for raising these points about complex regions. Indeed our way of constructing the input set of variants was overly stringent and excluded many regions. We have now revised our strategy to handle difficult loci: While before we excluded any region where at least one haplotype assembly did not have an alignment, we now include all regions where at least 80% of the assemblies align. PanGenie is now able to handle such “missing alleles” in a reference panel. We now also include chrX. In summary, these measures lead to the inclusion of 2.8Gbp of sequence in our reference panel (previously 2.5Gbp), corresponding to 91.8% of the included chromosomes (previously 87.5%). Of the remaining regions still inaccessible, 48.3% are gaps in GRCh38 and 24.0% are centromeres.

Comment E.5: Reviewer #4 provides a straightforward report; they also think that PanGenie is interesting but desires much more detail on how it works.

Response: Based on the reviewer's comments, we have improved the description of the PanGenie method and added two new Supplementary Figures (S1 and S2) that provide a detailed illustration of the graph construction and the evaluation process, respectively.

Comment E.6: We thought that Reviewer #1 and #3's comments about the benchmarking and complexity were important and should be fully addressed in a revision. We thought that other comments - specifically Reviewer #1's suggestion for a larger-scale analysis on the 1000G data, or looking at specific complex loci; and Reviewer #2 regarding non-human genomes - were also interesting and, if acted upon, would strengthen your study.

Response: This guidance is much appreciated. We fully addressed all these points, with one exception: After careful consideration, we decided not to extend the analysis to other species. We considered presenting a corresponding analysis on mice, but the use case was less convincing in our view because mouse researchers typically use inbred strains where the full genotypes of an offspring population can be straightforwardly determined. While PanGenie could be of great utility when applied to other (non-inbred) diploid species, the availability of high-quality haplotype resolved assemblies is too limited to present such use cases at the present time. We are very optimistic that such use cases will emerge when such data becomes available and we do not foresee any problems running PanGenie on such data sets. A polyploid version of PanGenie would clearly be very appealing and would have exciting applications in plant genomics, but

implementing and evaluating this would be a major project in itself and is, in our view, beyond the scope of this present work.

Reviewer #1 :

Comment 1.1: The paper proposes a new method (PanGenie) to genotype variation using pangenome graphs. The PanGenie implementation is publicly available with an open source license. In the paper, the authors describe how they constructed a graph containing variation found in five publicly available haplotype resolved (phased) human assemblies. Each haplotype is aligned to the reference genome to generate a phased set of input variants. Like in several other previous graph-based methods, the purpose of the graph is to make use of prior variant information to improve genotyping accuracy. PanGenie is also a mapping-free method, variation in the graph is genotyping using only a hash table of k-mer counts which results in a very fast genotyping compared to mapping-based methods. PanGenie does not discover new variants, however, the authors mention that this might be a possibility in the future.

PanGenie supports a wide variety of variants, including SNVs, indels, tandem repeats, structural variants. Therefore, the authors compare to both methods that genotype small variants (SNVs and indels) and those that genotyping larger variants (SVs), but also two (BayesTyper and Platypus) that can genotype all of the same variant types as PanGenie. In one experiment, they remove one of the five individuals and use a graph from the other four to genotype the one left out individual. In this experiment, BayesTyper seems to produce the most accurate genotype calls, however, in the STR/VNTR regions PanGenie outperforms the other tools. The authors highlight their methods performance in these repeat regions. Based on the description of the method, BayesTyper and PanGenie seem to have many similarities, as both tools genotype using pangenome graphs based on k-mer counts so it is no surprise their results are similar (pangenie “strict” set). The most distinct feature between these two methods is that PanGenie incorporates the haplotype information and uses that to infer genotypes.

Response: We thank the reviewer for the good summary and agree that the use of haplotype information inherent to phased genome assemblies is the most important novelty of PanGenie. We emphasize that incorporating this information was not a trivial task and required us to design and implement a novel algorithmic framework that bridges HMMs established in population genetics to modern k-mer indexing techniques. When setting strict genotype quality cutoffs, i.e. requiring high levels of evidence for each genotype, then indeed variants with direct k-mer evidence are preferentially selected, explaining the similarity in performance to BayesTyper in this regime, which has the inherent limitation of excluding relatively many variants (in fact, more than 72% of SVs in repetitive regions).

Comment 1.2: I thought all of the mapping-based methods are doing surprisingly poor in the benchmarks, as these metrics are not much in line with previous publications. For example, GATK and Platypus, have been repeatedly shown to have at least 99.5+% and 98+% sensitivity and

precision with high coverage data (30x+) in non-repeat regions for SNPs and indels, respectively. Both methods are mostly doing poorly in the benchmark of the paper and it is not clear why. My guess is that either the benchmark truth set has many incorrect calls, or that these methods perform suboptimally when “force calling” a set of variants and would actually perform better if they are allowed to discover their own set of variants. In case of the latter, the authors should also run these tools in their recommended way (with variant discovery). I realize this would mean that the dataset no longer contains the exact same variants, but it should not be an issue if you use published benchmark tools such as “rtg vcfeval” and Illumina’s [hap.py](https://github.com/Illumina/hap.py) for comparisons. These tools are designed to handle well datasets that have different variant representations. I think the authors should also add a comparison to an external public truth set using one of those benchmark tools. The truth set could, for example, either be Genome in a Bottle (GIAB, <https://www.nist.gov/programs-projects/genome-bottle>) or syndip (<https://www.nature.com/articles/s41592-018-0054-7/>). These truth set are an extremely useful resource for validating genotyping methods with real data. Genome in a Bottle already has a truth set for NA12878/HG001, which are already part of this study. The syndip truth set VCF is created using the same tools as the authors use for creating the input VCF, so it should also include the STR/VNTR regions that the manuscript highlights.

Response: We thank the reviewer for suggesting “rtg vcfeval”, which we now utilize to compute precision/recall/F-score, which we agree improves comparability to previous studies. We propose and present two different versions of precision and recall, once the “usual” evaluation with respect to a full truth set and once an “adjusted” version where we remove the variants from the truth set that are undetectable in a re-genotyping setting because they are unique in the evaluation sample (and hence absent in the input set of variants to be genotyped). We now also run Platypus and GATK in discovery mode with standard parameters and additionally compare these results to the GIAB small variant benchmark. The results are displayed in new supplementary figures S18 and S19 (pasted below). To specifically answer the reviewer’s queries on GATK and Platypus, we collected the corresponding statistics here, where an arrow points from the evaluated call set to the “ground truth” call set and is annotated with the respective (adjusted) precision/recall metric:

We make the following observations:

1. When comparing a standard discovery call set to the GIAB small variant benchmark (version 3.3.2), we observe a recall of 98.3% for GATK and 90.8% for Platypus (at 30x coverage). So these

numbers are somewhat worse than the ones mentioned by the reviewer. The discrepancy might stem from the fact that we use a recent version of the GIAB benchmark, which might incorporate more “difficult” regions/variants than previous versions as a result of the curation efforts of the GIAB team.

2. According to the numbers shown in the above figure, comparing the genotyping results to the GIAB ground truth leads to very similar estimates for adjusted precision/recall as compared to our assembly-based truth set. This supports the validity of using our assembly based call sets as truth sets in the leave-one-out analysis.

3. When using unadjusted precision/recall, then (expectedly), discovery mode has better recall than re-genotyping mode because it can successfully find variants absent from the input set to be genotyped (see Figure S18, right). Adjusted recall/precision adequately corrects for this.

4. The comparison of discovery mode and genotyping mode of GATK/Platypus results in a mixed picture. As shown in Figure S17, Platypus does markedly worse on complex regions when run in discovery mode. GATK does slightly better in discovery mode for complex repetitive regions, otherwise does better in re-genotyping mode. For the GIAB benchmark (Figure S18), both tools do slightly better in discovery mode with the exception of Platypus, where the recall is worse.

In conclusion, we have now provided substantially more statistics. In this way, we offer readers a very nuanced analysis. While many figures are presented in the Supplement for space constraints, we have added F-scores to the main Figure 3 (also pasted below). Especially for SVs, PanGenie clearly outperforms all other tools in terms of F-score, with a particularly drastic difference in repeat regions, supporting our previous conclusions.

Figure S17. variant discovery vs. re-genotyping for NA12878. In addition to re-genotyping given variants, GATK and Platypus were run in discovery mode to detect and genotype their own SNPs and indels (< 50bp). Results were evaluated inside of STR/VNTR regions and in non-repetitive regions. Adjusted F-scores were computed for coverage level 30x. We separately evaluate results for all variants falling into biallelic and complex regions of the genome as defined by the bubble structures in the pangenome graph.

Figure S18. Comparison to GIAB small variants for NA12878. The GIAB small variants benchmark set¹ was used as ground truth for evaluating the results of our "leave-one-out" experiment for SNPs and indels (< 50bp). We computed the adjusted precision and recall (left), as well as the un-adjusted versions (right) including variants unique to NA12878 and thus not genotypable by a re-genotyping approach. GATK and Platypus were additionally run in detection mode.

Figure 3. Leave-one-out experiment. Weighted genotype concordance at different coverages for sample NA12878 and F-scores for coverage 30× in non-repetitive (top) and STR/VNTR regions (bottom). We ran PanGenie, BayesTyper, Paragraph, Platypus, GATK, GraphTyper and Giraffe in order to re-genotype all callset variants. Besides not applying any filter on the reported genotype qualities ("all"), we additionally report genotyping statistics for PanGenie when using "high-gq" filtering (genotype quality ≥ 200). Insertions and deletions include all respective variants in biallelic regions of the genome, while *complex* contains all variant alleles falling into regions with complex bubbles in the pangenome graph representation.

Comment 1.3: In the PanGenie graph, the haplotype information is stored and is used to infer genotype calls. PanGenie has the fewest missing genotype calls out of all the benchmarked methods, and my understanding is that it is because they use the haplotype information for

inferring a possible genotype call, even when there are no unique k-mers to tag variant alleles. While it is not novel to infer genotypes based on haplotypes in population genetics, I have not seen this used as part of a genotyping method before. A few methods use the allele frequency as a prior but I personally like the authors' method much better, however, I think the reference panel here is way too small to be effective. Reference panels usually require several hundreds and often thousands of haplotypes for an accurate imputation of the genotype calls. I suspect that many of the genotype calls that have no unique k-mers that inferred genotype calls are widely inaccurate since the call will be determined only from a handful of known haplotypes. The authors seem to be aware of the lack of haplotypes, because they add six haplotype paths into their graph by typing them with PanGenie. However, this does not add any new haplotype path for variants that have no unique k-mers, since they are solely inferred using the other five haplotypes.

Response: We have created a new set of high-quality haplotype-resolved assemblies for 14 diverse samples (including 3 trios), using an updated version of our PGAS pipeline (Porubsky et al., Nature Biotechnology, 2021) that now incorporates HifiAsm as an assembler. We now call variants from these 22 assembled haplotypes (from 11 unrelated samples) as a reference panel, more than doubling its size compared to the previous version of this manuscript (five samples). While 22 haplotypes are a small number compared to panels of many thousands used for imputation in GWAS studies, the results show that this proves already very effective. We attribute this to the joint use of k-mer and haplotype information, where genotyping becomes possible when weak evidence from both sources (k-mers and haplotypes) is combined. Note that from 22 diverse reference haplotypes, one can expect to include the majority of all common alleles / local haplotype tracts. That being said, we do anticipate that the accuracy of PanGenie will improve further when the number of reference haplotypes increases. At present, the Human Pangenome Reference Consortium (HPRC) is producing sequencing data and corresponding genome assemblies, aiming to reach haplotype-level assemblies for 350 diverse samples over the next years. As part of the HPRC, we will apply PanGenie and hope to demonstrate such gains in accuracy going forward.

To investigate the accuracy of genotyping variants without unique k-mers, we plotted the corresponding weighted genotype concordance as a function of allele frequency (AF). As expected, we see that variants without unique k-mers (UK=0, shown in blue below) perform worse than those with k-mer support. Also expectedly, the influence of AF on the UK=0 variants is visibly stronger, reflecting that imputation becomes more difficult at low AFs. All in all, we note that the performance for UK=0 variants is still quite good. With larger reference panels, the performance for low AF variants would very likely improve further.

SNPs+Indels:

Biallelic SVs:

Comment 1.4: This also makes the “leave-one-out” experiment not very realistic. Instead of excluding an individual from the extended panel, it would be more realistic to leave out the individual before the other extended panel was created, because his/her haplotype has been used for inferring other calls in the dataset.

Response: With the increased number of assemblies, we do not create an “extended panel” anymore by genotyping additional samples. We have now designed the leave-one-out experiment such that we leave out the respective sample completely before constructing the panel used for genotyping. In this way, the variants contained in the panel as well as their representation in the graph/callset is completely independent of the left out sample.

Comment 1.5: As a side note: I believe that genotyping with genome inference would shine in genotyping studies in which there are thousands of publicly available haplotypes. For example, the phased set of 1000 genomes variants, or genotyping known haplotypes of HLA, KIR, immunoglobulin, and other difficult-to-call genes that have hundreds or thousands of known haplotypes publicly available. Of course, I do not insist that the authors would add any such analysis to the manuscript.

Response: We thank the reviewer for these suggestions and agree that challenging but medically relevant regions are particularly interesting. While there are specialized databases, we reasoned that many challenging regions are now correctly reconstructed in phased genome assemblies and we therefore wanted to explore how well we can genotype a challenging region using our assembly-based reference panel before resorting to external databases. We considered the MHC region and first evaluated the quality of our input assemblies in terms of their agreement to publicly available HLA genotypes for 1000 Genomes Samples, finding perfect agreement. We then applied PanGenie to three diverse test samples and observed excellent performance in most considered

genes, where HLA-DRB1 and C4 are performing the worst. These results are now included in the paper as follows.

New text in Results:

*“To evaluate the accuracy of all 14 haplotype-resolved assemblies in the HLA region, we used HLA*ASM to determine assembly HLA types (Supplementary Table S6). HLA*ASM successfully processed 27 out of 28 input assemblies and identified perfect (edit distance 0) HLA G group matches for all classical HLA loci (HLA-A, -B, -C, -DQA1 -DQB1, -DRB1) in all processed input assemblies with one exception (HLA-DRB1 in NA19238), which was resolved by manual curation. To verify the accuracy of the assembly HLA types, we integrated publicly available HLA genotype data for 1000 Genomes samples for HLA-A, -B, -C, -DQB1, and -DRB1, intersected these with the assembly-implied HLA types, and found perfect agreement in all evaluated cases (9 samples and 85 individual genotype comparisons, Supplementary Table S6).*

We additionally evaluated PanGenie's genotyping performance in the HLA region based on a "leave-one-out" experiment for samples HG00731, NA12878 and NA24385 and observed high levels of weighted genotype concordances across commonly studied HLA genes. While the average weighted genotype concordance across all three samples was lowest for genes HLA-DRB1 and C4 (58% and 79%, respectively in biallelic regions), it was between 98-100% for HLA-C, HLA-DPA1, HLA-DPB1, HLA-DRA in biallelic regions, and between 93-100% for all variants in complex regions (Supplementary Figure S19).”

Figure S19a:

Comment 1.6: The paper also describes genotyping of random 100 individuals from the 1000 Genomes project to show that the methods can scale to larger cohorts. The structural variants of this set is compared against public gnomad SVs, however, I found the results very surprising because the overlap between common variants (AF \geq 5%) in gnomad and PanGenie is very low (around 23% of PanGenie variants, I did not find in the manuscript the fraction of gnomad SVs which were not found with PanGenie but I would think based on Figure 4 that it is also quite low). The two datasets also have very different allele frequency spectrum (Figure 4a and 4c). The authors interpret the results by suggesting that gnomad might be missing up to 77% of common SVs in their study. However, in my experience the gnomad set is of very high quality, particularly for common SVs so I do not find this very convincing. I believe a much more likely explanation is that PanGenie is overestimating SV frequencies due to the low number of haplotypes used for genome inference.

Response: GnomAD is a carefully done study and a great SV resource. But it is based on short reads, which have been consistently shown to miss the majority of SVs in human genomes. Our results are consistent with these findings. We want to refer specifically to an overview study by Zhao et al. (AJHG, 2021) led by Michael Talkowski who also led the GnomadSV study (Collins et al., Nature, 2020). Zhao et al. provide a comprehensive overview on the number of SVs reported per sample by different studies (Figure 1A):

Therefore, in short-read based studies, typically <10,000 SVs are reported, while it is more than 20,000 in long-read based studies. Note that another major study by Abel et al. (Nature, 2020) on 17,795 genomes found as few as 4,014 SVs per genome. Zhao et al. go on and characterize the fraction of SVs discovered by long-reads that are also discoverable by short reads (and vice versa), shown in their Figure 1B:

The top bar shows that fewer than 35% of SVs discovered by long reads are detectable from short reads, with particularly low fractions of recovered SVs for insertions/duplications. All these observations are consistent with our finding that GnomAD misses a large fraction of SVs, among which there are many common ones.

When re-running the comparison of allele frequencies (AFs) estimated from haplotype assemblies (now n=22) to the AFs estimated from genotyping short-read samples, we again observe excellent agreement. Therefore, we see no evidence that our AF estimates could be inflated. Updated Figure 4b:

In conclusion, it may be still underappreciated by the genomics community how much structural variation is missed in typical short-read studies, even when using an ensemble of SV calling algorithms on large cohorts. We emphasize that providing access to such common variants in short read datasets is a major novelty of PanGenie. The robustness of the additional common SVs now accessible is further corroborated by adding Mendelian consistency analyses, see next comment below.

Comment 1.7: The only quality metric used here is testing for HWE abnormalities but those tests would not detect such issues. These results emphasize even further the need to benchmark against external truth sets that the authors could use to validate their SV genotyping (GIAB SV or syndip, for example).

Response: Besides the comparisons to the GIAB truth sets (see response to Comment 1.2, Figure S18), we now added additional measures to assess the quality and define a subsets of SVs passing most stringent quality criteria, termed “strict set” (note that what was called “strict” in the previous iteration of the paper is now called “high-gq” to avoid confusion between the two). One important new evaluation is based on Mendelian consistency evaluated across 100 mother-father-child trio, which reveals that a majority of SVs show excellent consistency rates (median of 0.95 and 0.96 for SV insertions and deletions, respectively):

Figure S20. Mendelian Consistency. Distribution of mendelian consistencies computed for each variant across all trios with at least two different genotypes. Our definition of mendelian consistency only takes trios with at least two different genotypes into consideration. That is, we exclude trios with all 0/0, 0/1 or 1/1 genotypes.

We defined a total of five different filter criteria, described in Methods as follows:

“We randomly selected 100 trios (20 of each superpopulation: AFR,AMR,EAS,EUR,SAS) that are part of the 1000 Genomes Project and genotyped all our variant calls across these 300 samples. We used the pangenome graph containing all eleven assembly samples as an input for PanGenie. Our callset might contain variants that are difficult to genotype correctly. Our goal is to identify a high quality subset of variants that we can reliably genotype. For this purpose, we define different filters based on the predicted genotypes that we will list below. One metric used for defining filters is the mendelian consistency. We computed the mendelian consistency for each variant by counting the number of trios for which the predicted genotypes are consistent with Mendelian laws. We only consider trios with at least two different genotypes, that is, we exclude a trio if all

three genotypes are 0/0, 0/1 or 1/1. This results in a more strict definition of mendelian consistency. For the unfiltered variant set, the mean mendelian consistency for SNPs was 0.98, for small variants between 0.93-0.95, for midsize variants between 0.90-0.93 and for large variants we observed numbers between 0.88-0.89 (Figure S20). In addition to genotyping all 300 trio samples we also genotyped all eleven panel samples using the full input panel. Genotyping samples that are also in the input graph can help us to find cases where panel haplotypes and reads disagree and thus is another useful filter criterion. We define filters as follows:

- **ac0-fail:** a variant fails this filter, if it was genotyped with allele frequency 0.0 across all samples.
- **mendel-fail:** a variant fails this filter if the fraction of mendelian consistent trios was below 90%. Our definition of mendelian consistency excludes all trios with all 0/0, all 0/1 or all 1/1 genotypes and only considers such with at least two different genotypes.
- **gq-fail:** a variant fails this filter if it was genotyped with a genotype quality below 200 in less than 5 samples.
- **self-fail:** in addition to the 100 trios, we also genotyped the 11 panel samples. A variant fails this filter, if the genotype concordance across all panel samples was below 90%.
 - **non-ref-fail:** the variant was genotyped as 0/0 across all panel samples.”

The majority of calls pass all these filters, as we now show in Figure S21:

Figure S21. Filters. We show all combinations of filters that we have applied to our genotyped variant callset and the respective number of variants in each subset. The black dots indicate that the respective filter failed.

Aside from the “strict” set passing all filters, we introduce a lenient set as an intermediate of lower but still good accurate calls:

“Similar to the approach introduced previously, we employ Mendelian consistency of the genotyped trios and the genotype quality reported by PanGenie to compute an integrated score for genotyping reliability of each variant. To this end, we defined different filters for a positive set with the most reliable (termed “strict” set) and a negative set with the most unreliable variants. Using a machine learning approach trained on these two subsets, we compute scores for all remaining variants reflecting how confident we are about their genotyping and used those to derive a “lenient” set of variants containing 78% and 83% of all insertion SVs and deletion SVs, respectively (Supplementary material). To confirm that the lenient set still offers very good genotyping performance, we analyzed allele frequencies and heterozygosities observed from the predicted genotypes for all variants in the lenient set and observed a relationship close to what is expected from Hardy-Weinberg equilibrium (Figure 4a, Methods). In non-repetitive

genomic regions as well as inside of repeats, between 88% and 92% of variants of different types in our lenient set showed no significant deviation when testing for HWE.”

Comment 1.8: In summary, I found the inclusion of haplotype information an interesting approach and I believe it could be promising for many applications where a large number of haplotypes is known. However, from the above observations, I have many concerns regarding the validation of the genotyping accuracy in the authors’ experiments. The authors should more properly benchmark their method by using published tools and compare against external truth sets to validate the accuracy of their method. Additionally, comparing the datasets to external datasets created with orthogonal methods would increase the confidence in the accuracy even further.

Response: We thank the reviewer for her/his enthusiasm about the future utility of PanGenie and hope to have addressed all concerns about the validity of the findings through the completely re-worked and substantially extended evaluation described above. To further relate our findings to one of the latest GIAB benchmarks, we analyzed the overlap of our strict and lenient sets to the benchmark of challenging medically important SVs (Wagner et al., bioRxiv, 2021):

“We compared our variant calls to the Genome in a Bottle set of medically relevant SVs. Our unfiltered callset contained 209 of all 250 medically relevant SVs. We further analyzed which fraction of these made it into our strict and lenient sets. We observed that 174 medically relevant SVs were contained in our lenient set, of which 119 were part of our strictly filtered set. We show the score distribution for these variants as well as allele frequencies and heterozygosities observed across all 200 unrelated samples for the lenient set in Supplementary Figure S24”

Minor comments

Comment 1.9: The manuscript compares against k-mer based pangenome method but none of the mapping-based pangenome methods, which are referenced in the introduction (vg, graph typer, GRAF). I would suggest comparing to at least one of those methods.

Response: We thank the reviewer for these suggestions. We added the two open source tools VG Giraffe (Siren et al., bioRxiv 2020) and GraphTyper. However, we did not add the proprietary GRAF tool, which we did not expect to perform better than the other tools based on previous experiences. Figure 3 shows that the two added tools do not change the conclusions and do not offer better performance than the other competing tools in repetitive regions. Regarding runtime, we can report that both methods are slower than PanGenie and especially the read-to-graph alignment step of Giraffe was prohibitively slow (which is why we only ran it for one sample and one coverage level). Refer to Table S5, which we also paste below in our answer to Comment 1.11.

Comment 1.10: Figure 4a labels both plots with “pangenie”. While reading, I assumed the lower plot was for gnomad SV since the allele frequencies reach lower numbers in that plot.

Response: We now show the labels of the top vs bottom part of Figure 4a more prominently, to avoid this confusion for future readers.

Comment 1.11: When discussing runtimes, I only found the total runtimes shown in Supplementary Table S4. It would be good to add some further stratification of it, particularly the time for running BWA and the k-mer counting. Since if the input variants were updated, you could subtract those runtimes for estimating the time for re-genotyping.

Response: In addition to the total runtimes required for producing genotypes from raw sequencing reads, we now also provide the runtimes needed only for the genotyping step (excluding the time needed for alignment or k-mer counting). All runtimes can be found in Supplementary Table S5:

coverage	method	NA12878				NA24385			
		time total	time genotyping	memory total	memory genotyping	time total	time genotyping	memory total	memory genotyping
5	PanGenie	21:06:10	19:42:05	84.8	36.4	31:44:24	29:30:54	84.6	36.2
	BayesTyper	27:23:15	26:22:21	39.3	39.3	36:31:30	35:20:37	39.2	39.2
	Platypus	18:12:42	1:20:10	18.2	0.2	20:39:51	1:31:42	8.7	0.1
	GATK ¹	34:41:06	17:24:26	18.2	0.4	35:24:17	15:53:15	8.7	0.4
	Paragraph ²	39:49:37	22:57:04	18.2	10.1	40:51:58	21:43:48	11.1	11.1
	GraphTyper ³	22:06:44	5:14:12	18.2	0.2	23:12:02	4:03:52	8.7	0.2
10	PanGenie	21:36:59	19:27:31	84.8	36.4	33:07:31	29:29:26	84.7	36.2
	BayesTyper	38:42:03	37:20:08	40.7	40.7	36:05:15	34:16:52	40.7	40.7
	Platypus	35:20:29	1:42:35	18.6	0.4	42:57:08	1:57:21	8.8	0.3
	GATK ¹	59:42:39	25:21:58	18.6	0.4	67:21:06	25:36:00	8.8	0.5
	Paragraph ²	66:02:14	32:24:20	18.6	13.2	86:19:41	45:19:54	12.2	12.2
	GraphTyper ³	42:52:25	9:14:31	18.6	0.3	49:30:28	8:30:41	8.8	0.2
20	PanGenie	23:46:08	19:39:33	84.8	36.4	24:24:09	19:41:24	84.7	36.3
	BayesTyper	32:03:53	29:59:40	41.0	41.0	44:49:37	41:59:38	41.1	41.1
	Platypus	68:38:45	2:11:46	28.4	0.7	81:28:44	2:42:48	8.8	0.5
	GATK ¹	107:04:36	39:18:12	28.4	0.5	120:51:45	40:43:18	8.8	0.8
	ParaGraph ²	137:18:30	70:51:31	28.4	14.3	139:56:03	61:10:07	12.9	12.9
	GraphTyper ³	84:34:29	18:07:30	28.4	0.5	92:58:00	14:12:04	8.8	0.3
30	PanGenie	24:58:54	19:31:51	84.8	36.4	26:48:22	19:41:23	84.7	36.3
	BayesTyper	32:24:13	29:34:54	41.1	41.1	48:30:38	44:34:30	44.4	44.4
	Platypus	99:12:01	1:59:29	39.1	1.0	123:09:20	3:02:53	8.8	0.9
	GATK ¹	143:57:46	44:54:12	39.1	0.5	176:26:20	54:21:41	8.8	0.9
	Paragraph ²	210:28:50	113:16:17	39.1	14.7	256:00:10	135:53:43	13.3	13.3
	GraphTyper ³	123:03:06	25:50:33	39.1	0.7	141:57:38	21:51:11	8.8	0.5
	Giraffe ³	3043:47:18	11:10:38	188.7	45.2				

¹ GATK was run on SNPs, small and midsize variants only.

² Paragraph was run on midsize and large variants only.

³ GraphTyper and Giraffe were run on large variants only.

Table S5. Resources. Runtime (in CPU hhh:mm:ss) and peak memory usage (in GB) of the different genotyping methods at different coverages. For all methods, we show the total resources needed for producing genotypes from raw, unaligned sequencing reads (“total”), as well as the resources needed only for the genotyping step (“genotyping”). Thus, for Platypus, GATK, Paragraph and GraphTyper the latter excludes the time needed to generate alignments against the reference genome. For Giraffe, it excludes the time for graph construction with vg, indexing and alignment. For k-mer based k-mer based approaches (PanGenie and BayesTyper), it excludes the k-mer counting step. All tools were run on a HPC-cluster predominantly consisting of Intel E5-2697v2 (2 × 12 cores and 128 GB of RAM) and Intel Xeon Gold 6136 (2 × 12 cores and 192 GB of RAM) nodes.

Comment 1.12: In addition to runtimes, memory requirements should be added.

Response: We added the peak memory usages (in GB) for all tools to Supplementary Table S5.

Comment 1.13: I thank the authors for the good work on the topic. I found the paper to be well written and easy to follow. I hope you find my comments to be constructive.

Response: We thank the reviewer for the positive evaluation and the many thoughtful and constructive comments, which have been very valuable for improving our manuscript.

Reviewer #2 :

Comment 2.1: In their publication, Ebler et al. present a novel algorithm PanGenie for genotyping of populations based on short reads.

PanGenie builds a pangenome graph with ~two haplotypes from phased SNPs, and then takes short reads from many individuals, counts k-mers in these reads, and looks up k-mers in the haplotype graph to genotype individuals fast. PanGenie outperforms regular short-read based genotyping pipelines because k-mer counting and lookup is faster than short-read alignment. Faster genotyping is needed now as human sequencing projects sequence millions of individuals, with read-alignment being one of the bottlenecks.

The authors show that PanGenie is more accurate in genotyping alleles in repetitive regions, which I feel is a bit like an apples-to-oranges comparison. PanGenie's input is a haplotype graph built by aligning chromosome-scale assemblies with the human reference (in the paper called 'contigs') using minimap2 via the authors' vcf-merging pipeline. The input for the other pipelines (BayesTyper etc.) are BWA-made soft-trimmed short-read alignments.

Response: We agree that the different tools have different modes of operation, and have differences in their ability to leverage different data sources. In fact, the ability to use haplotype-resolved assemblies is a major novelty in PanGenie and we are not aware of other tools able to do that while genotyping. The common denominator of the tools/workflows we selected is in their common input of the variants to be genotyped and in their output: We include different workflows/tools to achieve genotyping of an input set of variants. While indeed not all tools use the same data resources, we believe that such a comparison is justified due to the importance of this task to genomics researchers who want to use the best approach to accomplish genotyping of a new sample.

Also, we want to point out that BayesTyper does not rely on read alignments and, as PanGenie, is given the raw, unaligned short reads. For this revision, we have also added two additional tools to the comparison, GraphTyper and VG Giraffe. The latter one considers the haplotype paths

during its read mapping stage and hence makes use of the same data as PanGenie (albeit in a different way).

Comment 2.2: It is no surprise then that repetitive regions look better in PanGenie, as large chromosome-scale alignments easily bridge repeats while soft-trimmed short-read alignments struggle. This should at least be discussed. I wonder how it would look like if the input to PanGenie were BWA-made short-read-based SNPs, it would probably look much worse. The quality of the input SNPs obviously has a huge impact on PanGenie's performance.

Response: We thank the reviewer for these comments. We agree that the performance of PanGenie stems from the quality of the input assemblies. Such panels of assemblies now become more and more available as a resource to the community. That is, they can be downloaded by users who can use them to analyze their short read genomes. A main goal of our paper is to demonstrate how such assemblies can enable especially the genotyping of SVs typically missed by short reads (also see our response to Comment 1.6 above). By using a variant set generated by short-read analyses, this purpose would be defeated and we therefore decided against including such an analysis. The following sentences are included in the Discussion:

“Traditionally, especially longer variants are difficult to genotype based on short-reads only, since such variants are often located in repetitive or duplicated regions of the genome. Their short length makes it difficult to unambiguously map the reads in these regions affecting a genotyping process that relies on these alignments. K-mer based approaches additionally lack connectivity information contained in the reads. PanGenie overcomes these limitations of short reads, as it incorporates long-range haplotype information inherent to the pangenome reference panel it uses.”

Comment 2.3: The tool is presented as a generic tool when in fact it has only been assessed on human genomes and is unlikely to work with some other species. Plants for example have very different genomes to humans, often with polyploidy and low heterozygosity. This limitation should be clarified in the text.

Response: This is a good point. We have added a corresponding sentence to the discussion (pasted below). We would have liked to include a corresponding use case for a non-human species as a vignette in the paper, but felt that comparable high-quality phased assembly data with matched short reads are not yet publicly available for other diploid species. Therefore, we decided to keep such experiments for future work. We agree with the reviewer that different rates of heterozygosity and different LD structures can affect the performance. While we do not foresee any limitations applying PanGenie, these potential differences need to be quantified in future evaluations.

In its present implementation, PanGenie does not support polyploids. We agree that this would open up exciting applications in plant genomics. The mathematical models could be readily

extended to polyploids, but implementing this in a fast way will require substantial engineering effort.

“While we have only tested it on human data so far, PanGenie can be applied to any diploid genome once corresponding panels of high quality assemblies become available for other species.”

Comment 2.4: While some of the shortcomings of the method are discussed, I see a major advantage of read mapping is the ability to detect previously unseen variants, inversions and translocations. I wonder if these limitations will limit the future implementation of this method over read mapping.

Response: We agree that discovery of rare/unknown variants is not addressed by genotyping. Indeed the major added value of PanGenie is for common variants, where GnomAD (Collins et al., Nature, 2020), as a recent state-of-the-art study, only detected 23,202 common SVs (AF \geq 5%) and we make additional 36,315 common SV alleles accessible to short reads. With growing sizes of assembly-based reference panels, more SVs of lower AFs will be included and can be genotyped using PanGenie. That being said, we do see interesting avenues for future research on workflows that combine the “best of both worlds” and add a step for rare variant detection after PanGenie, as noted in the Discussion:

“Our model assumes that the unknown haplotypes of the sample to be genotyped are mosaics of the given panel haplotypes. Therefore, it currently cannot be used in order to genotype rare variants that are only present in the sample, but in none of the other haplotypes. Here, we believe that there are exciting opportunities to define downstream workflows that only discover variation that our approach has not captured because it was not present in the reference panel. That is, one could filter the reads for yet “unexplained” k-mers and use those for the discovery of rare variants.”

Comment 2.5: While the approach is interesting, it isn’t totally novel. Shajii et al., 2016 and Sun et al. 2018 should be cited because at its base, PanGenie uses the same idea – genotyping known SNPs is faster when you look up known k-mers instead of wasting time aligning reads (<https://pubmed.ncbi.nlm.nih.gov/27587672/> and <https://academic.oup.com/bioinformatics/article/35/3/415/5056043>). A major question is, given the limitations, would this method become widely adopted, and if so, how would any bias from using different haplotype graphs be accounted for when comparing studies?

Response: We thank the reviewer for bringing these two additional references to our attention. We do agree that using k-mers for genotyping is not a novel idea and we had already cited four papers on this aspect before (Iqbal et al., 2012; Dilthey et al., 2015; Dolle et al., 2017; and Sibbesen et al., 2018). We now added Shajii et al. (2016) and Sun et al. (2018) as well. The

novelty of PanGenie lies in combining k-mer based techniques with population genetic models of haplotypes in an integrated algorithm.

From the Introduction:

“A much faster alternative is to genotype known variants based on k-mers, short sequences of a fixed length k, in the raw sequencing reads. Counts of reference- and allele-specific k-mers allow fast and accurate genotyping of various types of genetic variation^{27–32} by bypassing the time consuming alignment step.”

Comment 2.6: Table S4 contains the runtimes for all compared software. I’d like to have added on what kind of machine these steps were run – number of CPUs, memory etc. I think for completeness’ and fairness sake, this table should also include the runtime of the vcf-merging pipeline. After all, that step does most of the heavy lifting, PanGenie itself ‘just’ checks k-mers for the called variants.

Response: Variant calling was done in a previous step and the resulting callset was given to **all** genotyping tools. Thus, the time needed for variant calling is exactly the same for all tools. Therefore, it was not included here. In practice, such reference panels are provided as a resource and users can just download them. The biggest computational hurdle in creating them is not the variant calling and graph construction, but doing the genome assembly. Thus, we anticipate that these steps are done by larger efforts creating reference resources for the community. We thank the reviewer for pointing out that we missed including CPU specifications, which we have now added. A column for memory consumption was also added.

Comment 2.7: The last line in the supplementary notes before the References part breaks off early – ‘tools can be f’

Response: Fixed.

Comment 2.8: Line 343, in the definition of genotype concordance, how is ‘wrong predictions’ defined here? Is it false positives, so a genotype is called where one shouldn’t be called, or is it false negatives, no genotype is called where one should be called, or is it both together, or something else entirely, i.e., genotype B is called where A should be called? Listing in a sentence which one it is probably helpful. If it is false negatives then ‘genotype concordance’ is the same as ‘recall’.

Response: With the substantially extended evaluation (see response to Comment E.2), we added additional metrics and now include in depth explanations of the different metrics, see section “Evaluation Metrics” in Methods and Figure S3:

Figure S3. Metrics used to evaluate genotyping results and how they define errors.

Comment 2.9: Line 218, ‘we separately mapped contigs of each haplotype’, it should say the average or N50 size of these contigs here, they’re probably huge

Response: We added a Supplementary Table (S1) with N50s of the new assemblies and refer to it in the corresponding sentence (both pasted below).

“For each sample, we separately mapped contigs of each haplotype (Supplementary Table S1) to the reference genome (GRCh38).”

Sample	Haplotype	ctg. N50	# contigs	v13 / hifiasm length (bp)	largest contig (bp)
HO00512	hl	34,301,582	1,777	3,188,143,530	101,334,844
HG00512	h2	34,874,650	1,449	3,163,234,783	112,440,880
HG00513	hl	45,364,295	1,500	3,136,967,952	133,503,586
HG00513	h2	45,319,314	1,227	3,113,449,797	139,638,253
HG00514	hl	17,395,387	1,983	3,141,391,447	94,121,208
HG00514	h2	18,722,440	1,650	3,121,917,894	72,949,789
HG00731	hl	35,342,509	2,218	3,179,873,135	130,295,053
HG00731	h2	31,517,176	1,791	3,145,395,507	131,214,380
HG00732	hl	22,794,071	1,084	3,161,371,445	70,183,886
HG00732	h2	18,785,010	849	3,128,313,934	85,401,373
HG00733	hl	32,098,695	1,711	3,141,624,750	81,981,905
HG00733	h2	39,889,742	1,327	3,127,483,091	110,942,518
HG02818	hl	14,735,342	1,645	3,148,193,201	62,389,557
HG02818	h2	13,891,585	1,346	3,131,545,528	53,330,289
HG03125	hl	19,740,003	1,453	3,144,381,367	78,085,250
HG03125	h2	16,030,141	1,233	3,121,673,094	69,854,835
HG03486	hl	13,742,469	1,400	3,172,408,948	63,278,388
HG03486	h2	15,627,668	1,269	3,152,826,116	58,336,852
NA12878	hl	33,400,276	3,631	3,129,308,283	104,882,848
NA12878	h2	27,880,200	3,014	3,108,352,699	110,737,365
NA19238	hl	15,612,125	2,954	3,126,238,496	71,400,915
NA19238	h2	15,239,724	2,418	3,107,339,654	72,441,052
NA19239	hl	19,056,746	2,217	3,198,825,166	84,398,966
NA19239	h2	16,698,371	1,806	3,171,600,679	95,184,935
NA19240	hl	29,153,232	1,978	3,153,890,228	104,206,385

NA19240	h2	32,117,261	1,588	3,136,752,685	95,070,688
NA24385	hl	23,950,673	1,649	3,173,344,587	98,025,482
NA24385	h2	28,576,363	1,306	3,156,300,763	111,314,854
mean		25,423,466	1,767	3,145,791,027	92,748,083

Table Si. Assembly statistics. Shown are N50s, the number of contigs, the length of the assembly (bp) as well as the size of the largest contig for all our 14 haplotype-resolved assemblies.

Reviewer #3:

Comment 3.1: The manuscript from authors Jana Ebler and colleagues reports a graph-based method that combines pan-genome representations from haplotype-resolved assemblies with k-mer based genotyping.

The method is well designed and algorithmically sound. The utilization of a Hidden Markov Model to link reconstructed genotypes is a great application of the technology in this context. Laudably, the authors carry out detailed and careful comparisons to alternative genotyping methods, and demonstrate that their graph-based method excels at genotyping larger variants (which is exactly what is expected from this method). They also convincingly show that the method scales well for population studies (i.e. with the number of genomes to be genotyped).

Response: We thank the reviewer for this positive assessment of our work.

I have two related major concerns:

Comment 3.2: Concern 1: The authors restricted the construction of the pan-genome graph to 87.5% of the genome, where they were able to identify genomic variation across the included 10 haplotypes from the 5 reference individuals. This likely excludes the majority of more complex regions, where the lion's share of the types of longer and more complex variation resides for which genotyping improvements are the most sorely needed. This may severely restrict the usefulness of the method.

Response: We thank the reviewer for bringing up this important point. Indeed, we are aiming to enable genotyping of complex loci. For the version of PanGenie used in preparation of our initial submission, only loci could be handled where all haplotype assemblies had an alignment. Now we improved our strategy to handle difficult loci: We include all regions where at least 80% of the assemblies align. PanGenie is now able to handle such “missing alleles” in a reference panel. We also include chrX. In summary, these measures lead to the inclusion of 2.8Gbp of sequence in our reference panel (previously 2.5Gbp), corresponding to 91.8% of the included chromosomes (previously 87.5%). Of the remaining regions still inaccessible, 48.3% are gaps in GRCh38 and

24.0% are centromeres. Due to the extended reference panel in our new evaluation (11 samples in contrast to 5 before, see response to Comment E.2 above) and this more inclusive strategy to incorporate difficult loci, the number of SV alleles included has grown from 55,908 to 150,428 during the revision. When comparing results for repetitive loci of the previous version to our current evaluation (both pasted below), we see that results of PanGenie have remained stable. In contrast, the results of some of the competing tools were slightly worse on the new evaluation; for instance, BayesTyper's genotype concordance went down for insertions and complex events, which might be due to the inclusion of more "difficult" SVs. (We note, however, that also the definition of the concordance metric changed slightly: To avoid a bias towards the ability to genotype absent 0/0 variants, we give equal weight to the concordance on all three genotypes 0/0, 0/1, and 1/1, we use "weighted genotype concordance", see corresponding discussion in response to Comment E.2).

INITIAL SUBMISSION:

REVISED VERSION:

Comment 3.3: Concern 2: It is unclear how the method will scale with the number of reference haplotypes included in the pan-genome reference graph. With an increasing number of haplotypes, the number of regions where imperfect copies of medium-sized and larger variants overlap in complex ways also increases. It is unclear how PanGenie's graph construction method that is conceptually clean for "well-behaved" and spatially distinct bi-allelic variations will perform in such scenarios. Will many of these areas be excluded (see concern 1 above)?

Response: The concern that too many regions would be excluded as the number of reference haplotype grows is addressed above in our answer to the previous comment. To give readers a view on the extent of multi-allelic variation as a function of allele length we include the following Supplementary Figure 1c:

We also added a paragraph describing the (asymptotic) scaling behavior to the supplement (pasted below). PanGenie handles the panel size we use in the present evaluation (22 haplotypes) with ease, but the basic model scales super-linearly in the number of reference haplotypes. In the meantime, we have applied PanGenie also in the frame of the main study of the Human Genome Structural Variation Consortium (HGSVC), published as Ebert et al. (Science, 2021), where we applied it to a reference panel of 64 haplotypes and genotyped 3,202 samples. To this end, we have added a mechanism to run iteratively on subsets of the input haplotypes and then combine the resulting posterior probabilities. This heuristic effectively leads to a scaling behavior linear in the number of haplotypes and will be practical for hundreds of reference assemblies. Looking ahead, we see different opportunities for further engineering of PanGenie in the spirit of pruning techniques now used in statistical phasing packages (e.g. Shapelt), which could ultimately lead to scalability to extremely large panels. This heuristic panel splitting however, is only used for input panels of >30 haplotypes and therefore not used in the present study.

Supplementary text:

“PanGenie is based on a Hidden Markov Model which, for each variant position, defines one state for each pair of haplotypes of the input panel. Given n variants to be genotyped and m panel haplotypes (which equals twice the number of samples), there will be $O(m^2 \cdot n)$ states. Applying the Forward-Backward algorithm to the HMM thus corresponds to a runtime quadratic in the number of states. If the number of panel haplotypes grows, the algorithm will get slow. To tackle this problem, we have implemented a subsampling step, which repeatedly subsamples sets of haplotypes from the full panel and genotypes all variants in each subset. Genotype predictions resulting from each of these subsets are later combined to obtain the final genotype likelihoods. Assume we split the set of m input haplotypes in l subsets each of a fixed size k . PanGenie’s genotyping step is now run separately on each of the l sets in time $O(k^4 \cdot n)$. This will result in a total runtime linear in the number of subsets, i.e. $O(l \cdot k^4 \cdot n)$. PanGenie automatically switches to this subsampling mode if the number of input haplotypes exceeds 30. For all experiments in this paper, we ran PanGenie without subsampling using the full HMM.”

Comment 3.4: Overall, a really nice method, but I fear that method will not be able to achieve its full potential impact for genotype improvement without strengthening extending the pan-genome graph construction method to include more than the "best-behaved" parts of the human genome.

Response: We thank the reviewer for this largely positive assessment and refer to our above answers (to Comments 3.2 and 3.3) for how we have addressed the remaining concerns.

Reviewer #4 :

Comment 4.1: The paper describes a new method for genotype calling from sequence data, that uses haplotype-resolved sequences to build a reference graph. Reads are then compared to the graph and an HMM is used to estimate a likely pair of sequences. The method seems novel, and the comparison to other methods is convincing.

Response: We are grateful for this positive assessment.

Comment 4.2: However the method itself is poorly described. The paper does not make it clear how variants are called. Especially variants that are not represented in the pangenome graph.

I think the method has 4 main steps?

1. The pangenome graph is constructed. I think this involves only unique kmers to each region?
2. kmers in the graph are counted in the reads of a new sample.
3. The HMM resolves a pair of paths through the graph.
4. How then are variants that are not represented in graph called? The methods section doesn't mention the word deletion once. Tell us how they are called in the Methods. I think you need to

make this a lot clearer. Setting out a series of steps, and having figures that better describe examples would help.

Response: To improve clarity, we now provide two new supplementary figures (pasted below) that illustrate variant calling, pangenome graph construction and the leave one out evaluation. These figures are now referenced at the appropriate places in Methods. The Methods part has also been revised to reflect the substantially extended new evaluation and we carefully describe all involved steps.

PanGenie cannot discover variants. Our method is a re-genotyping method and here we use it to genotype variation detected from haplotype-resolved assemblies of a set of known samples in a new target sample (which is **not** present in the pangenome graph). Thus, like the other genotyping methods (BayesTyper, Paragraph, GraphTyper, etc), PanGenie can only genotype variants present in the pangenome graph. This limitation and potential future synergies with downstream methods is mentioned in the Discussion:

“Our model assumes that the unknown haplotypes of the sample to be genotyped are mosaics of the given panel haplotypes. Therefore, it currently cannot be used in order to genotype rare variants that are only present in the sample, but in none of the other haplotypes. Here, we believe that there are exciting opportunities to define downstream workflows that only discover variation that our approach has not captured because it was not present in the reference panel. That is, one could filter the reads for yet “unexplained” k-mers and use those for the discovery of rare variants.”

Figure S1. Variant calling and graph construction. **a**) Shown are haplotype-resolved assemblies for three samples and corresponding variant calls made relative to a reference genome. On the right, we show how these variants are represented in a VCF file (simplified). The VCF file is biallelic and contains one record per (distinct) variant allele detected across the assemblies. **b**) Shown is the pangenome representation of the variants detected in panel a). Variants are represented as bubble structures. Sets of overlapping variants are merged into a single multi-allelic bubble (see first and last bubble for examples). Each haplotype can be represented as a path through the graph. We represent the pangenome in terms of a VCF file containing a record for each bubble and alleles corresponding to the branches of the bubble (right). We keep track of which callset variants each branch of the bubble was constructed from of as illustrated in the VCF representation. In this way, we can later convert genotypes derived for a bubble back to genotypes for each individual variant inside of a bubble. Note that our VCFs contain the actual allele sequences in their "ALT" column, we replaced them by their IDs in this figure for simplicity. **c**) For our pangenome graph constructed from eleven samples, we show the number of branches in a bubble as a function of its length which we define by the length of the longest path through the bubble (in bp).

Figure S2. Leave one out experiment. We illustrate the leave-one-out experiment using three samples. Variants are called for all samples based on haplotype-resolved assemblies. For evaluation, we construct a callset containing all variants called in samples 1 and 3, and a truth set containing all variants called in the left out sample (sample 2). The former set of variants is used for genotyping, the latter for evaluation. When running PanGenie, BayesTyper and Platypus, we first convert the variant calls into a pangenome graph representation (stored as VCF) and genotyped the corresponding bubbles (A). We keep track of which bubbles consist of which variant alleles so that genotypes can later be converted back to the original variant representation. For the other tools tested (GATK, Platypus, GraphTyper, Giraffe), we directly used the callset variants as input, without creating the graph (B). The genotypes predicted by each tool are then compared to the variants detected in the left out sample for evaluation. Variants unique to the left out sample cannot be genotyped correctly by any re-genotyping approach (marked in red). We exclude such variants when computing weighted genotype concordances and adjusted precision/recall/F-score metrics.

Comment 4.3: In figure 1 : The dashed red and blue lines - do these colors correspond to the solid colored lines (grey, green, blue and orange) in the left hand panels? It seems to me that the red dashed line might correspond to the solid green line?

Response: Thanks for pointing this out. The dashed lines were indeed not properly explained. Their purpose was just to indicate the two alleles the sample likely carries for this bubble region, but we realized that the figure is more clear without them. They are now removed.

Comment 4.4: I think the figure could be better arranged/annotated to make it clearer that the reads (top left box) are from the sample to be genotyped.

Response: Thanks. We added corresponding labels to make this clear.

Comment 4.5: It's not clear where you get the list of 3 possible alleles from in the top right panel. Also the kmer CAGG in the middle of the blue dashed line - does the correspond to the middle kmer (orange circle) on the blue and orange lines of the first bubble in the bottom left panel. It could be clearer how you go about calling a deletion in this example. Maybe you could have an example without a deletion as the main figure, then a supplementary figure where you illustrate what happens when there is a deletion.

Response: Indeed, the number of k-mers shown on top (in the zoomed in inset) was different from the zoomed out figure below. Thanks for spotting this. We have synchronized this. Also, we added Supplementary Figure S1 (see above) with an extended example.

Using orange for circles and lines in the bottom left panel isn't ideal. Could you annotate in the figure "bubble 1" , "bubble 2" etc

Response: We changed the color of that haplotype path and annotated the bubbles as suggested.

Comment 4.6: In the caption you say "The second bubble is poorly covered by k-mers, however, linkage to adjacent variants can be used to infer the two local haplotype paths." In the text (lines 53-54) you say "The second variant is poorly covered by k-mers but the count distributions along the alleles of the first variant indicate that the unknown genome is composed of the green and blue haplotypes." I think you mean second bubble rather than second variant?

Response: Thanks for spotting this. We now talk about the "second bubble".

Comment 4.7: On lines 70-74 you describe "our callsets" and it would be better if you could define that more clearly.

Response: These parts of the paper have been rewritten to both improve clarity and reflect the revised experimental setup.

Comment 4.7: Does the method work equally well on WGS and Exome sequencing?

Response: If "WGS" refers to "whole genome sequencing", then this is what is used throughout the evaluation: Samples from short-read WGS are genotyped with respect to a reference panel of haplotype-resolved assemblies. The method is indeed designed for WGS data as it tries to do a "holistic" inference of the genotypes along each chromosome. Hence, it is not intended for exome data, which only yields isolated coverage of the captured regions.

Decision Letter, first revision:

6th Sep 2021

Dear Tobias,

Your Technical Report, "Pangenome-based Genome Inference" has now been seen by 4 referees. You will see from their comments below that while they find your work of interest, some important points are raised. We are interested in the possibility of publishing your study in Nature Genetics, but would like to consider your response to these concerns in the form of a revised manuscript before we make a final decision on publication.

In brief, three reviewers are satisfied with your revision and are now supportive of publication.

Reviewer #1, however, is not. They note that the new GIAB small variant truth set comparison suggests that PanGenie is missing a large proportion of calls that would have been made by other, widely used tools. They also suggest that this truth set benchmarking should be extended to the full SV set.

We agree with Reviewer #1 that this performance gap does appear concerning; it would be important to explain to the reader what exactly is happening with these variants, such that PanGenie is not able to call them. We also concur with them that an expanded SV truth set benchmark would also add to the confidence in the advance offered by PanGenie.

To guide the scope of the revisions, the editors discuss the referee reports in detail within the team, including with the chief editor, with a view to identifying key priorities that should be addressed in revision and sometimes overruling referee requests that are deemed beyond the scope of the current study. We hope that you will find the prioritized set of referee points to be useful when revising your study. Please do not hesitate to get in touch if you would like to discuss these issues further.

We therefore invite you to revise your manuscript taking into account all reviewer and editor comments. Please highlight all changes in the manuscript text file. At this stage we will need you to upload a copy of the manuscript in MS Word .docx or similar editable format.

*1) Include a “Response to referees” document detailing, point-by-point, how you addressed each referee comment. If no action was taken to address a point, you must provide a compelling argument. This response will be sent back to the referees along with the revised manuscript.

*2) If you have not done so already please begin to revise your manuscript so that it conforms to our Technical Report format instructions, available

[here](http://www.nature.com/ng/authors/article_types/index.html).

*3) Include a revised version of any required Reporting Summary:

[REDACTED]

We hope to receive your revised manuscript within four to eight weeks. If you cannot send it within this time, please let us know.

Nature Genetics is committed to improving transparency in authorship. As part of our efforts in this direction, we are now requesting that all authors identified as ‘corresponding author’ on published papers create and link their Open Researcher and Contributor Identifier (ORCID) with their account on the Manuscript Tracking System (MTS), prior to acceptance. ORCID helps the scientific community achieve unambiguous attribution of all scholarly contributions. You can create and link your ORCID from the home page of the MTS by clicking on ‘Modify my Springer Nature account’. For more information please visit please visit www.springernature.com/orcid.

Sincerely,

Michael Fletcher, PhD
Associate Editor, Nature Genetics

ORCID: 0000-0003-1589-7087

Reviewers' Comments:

Reviewer #1:

Remarks to the Author:

In this revision, the authors made several improvements to the PanGenie manuscript which addresses most of the reviewers' concerns. I have the following points for the new additions.

The manuscript now has a comparison of small variants to the Genome in a Bottle truth set, as was suggested in the last round of reviews. However, the same successes of the PanGenie pipeline in the "leave-one-out" experiment are not observed in the GIAB truth set comparison. Looking at the 30x coverage results, the PanGenie pipelines' F1-scores are now among the lowest ones in the benchmark due to low recall. The PanGenie "high-gq" pipeline appears to have the lowest F1-score of all tested methods at around 0.90-0.91 (based on my eyeballing on the figure since I did not find the exact values), considerably lower than the mapping-based methods. The PanGenie "all" pipeline has a somewhat higher F1-score but it still looks to be lower than the mapping-based methods.

Can the authors provide any insights why they think PanGenie is missing 10-17% of those highly confident variant calls whereas GATK is missing ~1-2%? Even after removing variants from the truth set which are not in the input set, PanGenie is missing 4-10% of the remaining variants. The GIAB truth set is, to my knowledge, the highest quality truth set available for small variants so these results should warrant attention.

Also, I do not think the authors have fully addressed my point on validating their SV genotyping against an external truth set (i.e. GIAB SV or syndip), so I want to reiterate that point. These data sets are the best resource available for comparing different SV genotyping pipelines so I think it is reasonable to ask for such a comparison. For the small variants, we see quite a different picture between the "leave-one-out" and the GIAB benchmarks and I am concerned that a similar story will unfold if the SV genotypes are evaluated with orthogonal data. Thus, please consider benchmarking PanGenie SV genotyping against the

other methods using one of the suggested external SV truth sets.

(Minor) I found the definition of the "gq-fail" filter confusing:

gq-fail: a variant fails this filter if it was genotyped with a genotype quality below 200 in less than 5 samples.

Wouldn't it be a good variant if there are fewer samples with low genotype quality?

Again I would like to thank the authors for their work on the topic.

Reviewer #2:

Remarks to the Author:

Many thanks for the opportunity to read the revised version, I can confirm that my comments have been suitably addressed and that I have no further suggestions.

Reviewer #3:

Remarks to the Author:

As far as I am concerned, my reviewer comments were addressed satisfactorily.

Reviewer #4:

Remarks to the Author:

The authors have addressed all my comments well. Congratulations on a nice paper.

Author Rebuttal, first revision:

Reviewer #1

Comment 1.1: The manuscript now has a comparison of small variants to the Genome in a Bottle truth set, as was suggested in the last round of reviews. However, the same successes of the PanGenie pipeline in the "leave-one-out" experiment are not observed in the GIAB truth set comparison. Looking at the 30x coverage results, the PanGenie pipelines' F1-scores are now among the lowest ones in the benchmark due to low recall. The PanGenie "high-gq" pipeline appears to have the lowest F1-score of all tested methods at around 0.90-0.91 (based on my eyeballing on the figure since I did not find the exact values), considerably lower than the mapping-based methods. The PanGenie "all" pipeline has a somewhat higher F1-score but it still looks to be lower than the mapping-based methods.

Response: We thank the reviewer for looking carefully at these comparisons. We now provide further explanations in order to clarify that the results from our leave-one-out evaluations with assembly-based truth sets are in fact consistent with GIAB-based evaluations. Specifically:

1. We realized that our definition of "untypable" variants might have been slightly misleading. It led to counting variants as FN that were included in the GIAB benchmark, but missing in the assembly based call set (see illustration as part of our answer to Comment 1.2); that applied to 85,476 out of 3,561,178 small variants (2.4%) in the GIAB benchmark. No genotyping tool could ever detect such a variant, such that all recall values were smaller by the corresponding amount. We show the results for SNPs+indels for both experiments below (left two panels copied from previous Supplementary Figures S17 and S18; for GIAB we show two versions, using the old (middle panel) and the new definition of untypable variants (right panel)). It becomes apparent that the adjusted recall of all re-typing tools increases as those additional "untypable" variants are not considered.
2. When comparing the evaluation using the assembly-based truth set (left) to GIAB (right), it can be seen that PanGenie's precision is slightly lower for this variant class compared to the other tools in *both* experiments. That is, the qualitative picture painted by these analyses for the precision of re-genotyping tools is very comparable.
3. While all re-typing tools reach a similar recall when using the GIAB set as ground truth, PanGenie's recall is higher compared to the other tools when the assembly-based truth set is used, which leads to the higher F-score. A likely explanation for this is that our assembly-based callset includes regions that are not well accessible by short-read alignments, while the GIAB high confidence regions mainly cover regions that short-read callers can handle well (note that GATK was one of the callers used to construct this benchmark set). Our non-repetitive and non-complex regions cover around 323Mbp of sequence missing from the GIAB high confidence regions. Thus, it is not surprising that the recall of the mapping-based callers drops when using the assembly-based calls as ground truth.

Overall, the established tools (e.g. GATK) deliver excellent performance for small variants, especially in non-repetitive regions. In our view the two evaluations (assembly-based vs. GIAB; left vs. right panels below) display similar behaviors for the re-genotyping tools (solid lines) and specifically the use of the assembly-based benchmark has not provided an unfair advantage to PanGenie. While the qualitative behavior is similar, there are some differences, likely mostly reflecting the larger set of genomic regions included in the assembly-based benchmark. We would like to emphasize that the main focus of this paper -- and the main advance of PanGenie -- is on longer indels and SVs, where we have demonstrated that PanGenie clearly outperforms the other tools tested. We expect PanGenie's performance to increase further, for both small and large variants, as larger reference panels become available.

3

Comment 1.2: Can the authors provide any insights why they think PanGenie is missing 1017% of those highly confident variant calls whereas GATK is missing ~1-2%? Even after removing variants from the truth set which are not in the input set, PanGenie is missing 4-10% of the remaining variants. The GIAB truth set is, to my knowledge, the highest quality truth set available for small variants so these results should warrant attention.

Response: The 10-17% quoted by the reviewer pertain to (unadjusted) recall and hence include variants unique to the benchmark sample and absent from the input set to be genotyped. Beyond that, even the “adjusted recall” included variants part of the GIAB benchmark, but absent in the assembly call set (see answer to Comment 1.1 above). The relatively low recall of all re-genotyping methods can be explained based on how we defined the set of *untypable* variants. Consider panel a in the figure below. The GIAB small variant benchmark contains variant calls for NA12878 (small circle). Our assembly-based callset contains calls for all our 11 panel samples (including NA12878, large circle). A subset of these calls are unique to NA12878, i.e. they were only seen in NA12878 and in none of the remaining 10 samples. For our genotyping experiments, we provide all re-typing tools (GATK, Platypus, BayesTyper, Paragraph, GraphTyper and PanGenie) with the calls detected across these 10 samples, which means, all variants unique to NA12878 cannot be genotyped. Therefore, we previously removed the set of unique variants from our truth set for evaluation (area shown in gray) when computing “adjusted recall”. However, for the GIAB comparisons, this definition of *untypable* did **not** include the orange set: these are variants that are in the GIAB small variant set, but are not contained in our assembly-based callset. These variants cannot be genotyped either when using re-typing tools, as they are missing from the input variants and therefore, will be considered false negative calls during evaluation. In other words, the recall of the re-typers is limited by the recall of the input callset. This explains why all re-typing tools show a decreased recall, while the discovery tools (GATK and Platypus in discovery mode) are not affected due to their ability to detect variants themselves.

We apologize for the confusion that this might have created and have now updated our definition of *untypable variants* to also include variants contained in external truth sets that were missing from the input variants (panel b in figure below). Using this new definition, the curves of all re-typers are shifted to the right, reaching recalls similar to the discovery tools (Supplementary Figure S8 (left), see second figure below for a comparison of new/old version).

old definiton of "untypables"

new definiton of "untypables"

Definition of "untypable" variants previously and now. Previously, the orange variants (contained in the external truth set but not contained in our filtered assembly-based calls) were considered to be false negatives for all re-typing tools, as those tools can only type what they are given as input. The new definition now corrects for this.

New version

Old version

*New (left) and old (right) version of **Supplementary Figure S18** (“without untypables”). The new version uses the updated definition of untypable variants.*

We further analyzed why our assembly-based callset misses some of the GIAB variants. Comparing both callsets results in a precision of 99.3% and a recall of 97.6%. This means around 2.4% of the GIAB small variant calls are missing from our calls (= orange part in panel a). This is mainly because we strictly filtered our calls prior to using them as ground truth for our genotyping evaluations. As explained in the Methods section, we remove all variants from our callset for which there was a mendelian error in at least one of the trios and positions covered by less than 20% of the panel haplotypes. The reason for this rather strict filtering was that we aimed to derive a high quality subset of variants with reliable genotypes (rather than maximizing recall) that we can use as a ground truth for evaluating genotyping accuracies of the different tools on a given set of variants. We want to stress again that the focus of this paper is genotyping, not variant calling.

Comment 1.3: Also, I do not think the authors have fully addressed my point on validating their SV genotyping against an external truth set (i.e. GIAB SV or syndip), so I want to reiterate that point. These data sets are the best resource available for comparing different SV genotyping pipelines so I think it is reasonable to ask for such a comparison. For the small variants, we see quite a different picture between the "leave-one-out" and the GIAB benchmarks and I am concerned that a similar story will unfold if the SV genotypes are evaluated with orthogonal data. Thus, please consider benchmarking PanGenie SV genotyping against the other methods using one of the suggested external SV truth sets.

Response: We now additionally use the syndip SV calls as an external ground truth for evaluating our genotypes. We performed the same “leave-one-out” experiment as before (using the provided reads for syndip; with the exception of Giraffe which requires an excessive amount of compute time), providing our assembly-based variants as input to all genotyping tools. We then evaluated the results by comparing the genotypes to the syndip SV set, in the same way as we did when using our assembly-based calls of the left out sample as ground truth. We generated plots equivalent to main Figure 3 and pasted them below: on the left, we show the results for NA12878 presented in main Figure 3, on the right we show the new results for syndip evaluated using the syndip benchmark set.

Overall, for both truth sets (assembly-based, syndip) the performance plots show very similar behaviors. For syndip, the absolute results tend to be somewhat worse for all tools, especially for insertions in repeat regions, which likely is related to our callset being more strictly filtered (and

thus excluding some of the very difficult cases). Overall, PanGenie outperforms the other tools also using the syndip benchmark, leading to the same conclusions reached in the paper.

*Leave-one-out results using the assembly-based callset (left, taken from **main Figure 3**) and the syndip SV set (right) as ground truth for genotyping evaluation.*

We also considered using the GIAB SV truth set as another external benchmark set but it is only available for reference sequence version GRCh37. All our callsets and analyses were generated using the newer version GRCh38, this makes a comparison to GIAB SV quite difficult. From our experience, attempting to liftover SVs would discard many events and shift breakpoints for many others leading to a lower-quality analysis. While it's possible to regenerate the whole evaluation on GRCh37, it's intensely time consuming. For these reasons and because this comparison would add little over analyses we have already done, we don't believe it's worth pursuing a GIAB SV comparison.

Comment 1.4:

(Minor) I found the definition of the "gq-fail" filter confusing:

gq-fail: a variant fails this filter if it was genotyped with a genotype quality below 200 in less than 5 samples.

Wouldn't it be a good variant if there are fewer samples with low genotype quality?

Response: This is indeed a typo, "less than" should be "more than". We thank the reviewer for pointing this out.

Decision Letter, second revision:

8th Nov 2021

Dear Tobias,

Thank you for submitting your revised manuscript "Pangenome-based Genome Inference" (NG-TR56979R1). It has now been seen by the original referees and their comments are below. The reviewers find that the paper has improved in revision, and therefore we'll be happy in principle to publish it in Nature Genetics, pending minor revisions to satisfy the referees' final requests and to comply with our editorial and formatting guidelines.

Please email us a copy of the file in an editable format (Microsoft Word or LaTeX)-- we can not proceed with PDFs at this stage.

Sincerely,

Michael Fletcher, PhD
Associate Editor, Nature Genetics

ORCID: 0000-0003-1589-7087

Reviewer #1 (Remarks to the Author):

Thank you for adequately responding to my remaining concerns.

Final Decision Letter:

3rd Mar 2022

Dear Tobias,

I am delighted to say that your manuscript "Pangenome-based genome inference allows efficient and accurate genotyping across a wide spectrum of variant classes" has been accepted for publication in an upcoming issue of Nature Genetics.

Your paper will be published online after we receive your corrections and will appear in print in the next available issue. You can find out your date of online publication by contacting the Nature Press Office (press@nature.com) after sending your e-proof corrections. Now is the time to inform your Public Relations or Press Office about your paper, as they might be interested in promoting its publication. This will allow them time to prepare an accurate and satisfactory press release. Include your manuscript tracking number (NG-TR56979R2) and the name of the journal, which they will need when they contact

our Press Office.

Please note that *Nature Genetics* is a Transformative Journal (TJ). Authors may publish their research with us through the traditional subscription access route or make their paper immediately open access through payment of an article-processing charge (APC). Authors will not be required to make a final decision about access to their article until it has been accepted. [Find out more about Transformative Journals](https://www.springernature.com/gp/open-research/transformative-journals)

Authors may need to take specific actions to achieve [compliance with funder and institutional open access mandates](https://www.springernature.com/gp/open-research/funding/policy-compliance-faqs). If your research is supported by a funder that requires immediate open access (e.g. according to [Plan S principles](https://www.springernature.com/gp/open-research/plan-s-compliance)) then you should select the gold OA route, and we will direct you to the compliant route where possible. For authors selecting the subscription publication route, the journal's standard licensing terms will need to be accepted, including [self-archiving-and-license-to-publish](https://www.nature.com/nature-portfolio/editorial-policies/self-archiving-and-license-to-publish). Those licensing terms will supersede any other terms that the author or any third party may assert apply to any version of the manuscript.

Please note that Nature Research offers an immediate open access option only for papers that were first submitted after 1 January, 2021.

If you have posted a preprint on any preprint server, please ensure that the preprint details are updated with a publication reference, including the DOI and a URL to the published version of the article on the

journal website.

If you have not already done so, we invite you to upload the step-by-step protocols used in this manuscript to the Protocols Exchange, part of our on-line web resource, natureprotocols.com. If you complete the upload by the time you receive your manuscript proofs, we can insert links in your article that lead directly to the protocol details. Your protocol will be made freely available upon publication of your paper. By participating in natureprotocols.com, you are enabling researchers to more readily reproduce or adapt the methodology you use. [Natureprotocols.com](https://natureprotocols.com) is fully searchable, providing your protocols and paper with increased utility and visibility. Please submit your protocol to <https://protocolexchange.researchsquare.com/>. After entering your [nature.com](https://www.nature.com) username and password you will need to enter your manuscript number (NG-TR56979R2). Further information can be found at <https://www.nature.com/nature-portfolio/editorial-policies/reporting-standards#protocols>

Sincerely,

Michael Fletcher, PhD
Associate Editor, Nature Genetics

ORCID: 0000-0003-1589-7087